# BERTScore: Evaluating Text Generation with BERT

**Tianyi Zhang**[*†‡◇], **Varsha Kishore**[*‡], **Felix Wu**[*‡], **Kilian Q. Weinberger**[†‡◇], and **Yoav Artzi**[‡§]

[‡]Department of Computer Science and [§]Cornell Tech, Cornell University
{vk352, fw245, kilian}@cornell.edu    {yoav}@cs.cornell.edu

[◇]ASAPP Inc.
tzhang@asapp.com

## Abstract

We propose BERTScore, an automatic evaluation metric for text generation. Analogously to common metrics, BERTScore computes a similarity score for each token in the candidate sentence with each token in the reference sentence. However, instead of exact matches, we compute token similarity using contextual embeddings. We evaluate using the outputs of 363 machine translation and image captioning systems. BERTScore correlates better with human judgments and provides stronger model selection performance than existing metrics. Finally, we use an adversarial paraphrase detection task to show that BERTScore is more robust to challenging examples when compared to existing metrics.

## 1 Introduction

Automatic evaluation of natural language generation, for example in machine translation and caption generation, requires comparing candidate sentences to annotated references. The goal is to evaluate semantic equivalence. However, commonly used methods rely on surface-form similarity only. For example, BLEU (Papineni et al., 2002), the most common machine translation metric, simply counts $n$-gram overlap between the candidate and the reference. While this provides a simple and general measure, it fails to account for meaning-preserving lexical and compositional diversity.

In this paper, we introduce BERTScore, a language generation evaluation metric based on pretrained BERT contextual embeddings (Devlin et al., 2019). BERTScore computes the similarity of two sentences as a sum of cosine similarities between their tokens' embeddings.

BERTScore addresses two common pitfalls in $n$-gram-based metrics (Banerjee & Lavie, 2005). First, such methods often fail to robustly match paraphrases. For example, given the reference *people like foreign cars*, BLEU and METEOR (Banerjee & Lavie, 2005) incorrectly give a higher score to *people like visiting places abroad* compared to *consumers prefer imported cars*. This leads to performance underestimation when semantically-correct phrases are penalized because they differ from the surface form of the reference. In contrast to string matching (e.g., in BLEU) or matching heuristics (e.g., in METEOR), we compute similarity using contextualized token embeddings, which have been shown to be effective for paraphrase detection (Devlin et al., 2019). Second, $n$-gram models fail to capture distant dependencies and penalize semantically-critical ordering changes (Isozaki et al., 2010). For example, given a small window of size two, BLEU will only mildly penalize swapping of cause and effect clauses (e.g. *A because B* instead of *B because A*), especially when the arguments A and B are long phrases. In contrast, contextualized embeddings are trained to effectively capture distant dependencies and ordering.

We experiment with BERTScore on machine translation and image captioning tasks using the outputs of 363 systems by correlating BERTScore and related metrics to available human judgments. Our experiments demonstrate that BERTScore correlates highly with human evaluations. In machine translation, BERTScore shows stronger system-level and segment-level correlations with human judgments than existing metrics on multiple common benchmarks and demonstrates

---

[*]Equal contribution. [†] Work done at Cornell.

strong model selection performance compared to BLEU. We also show that BERTSCORE is well-correlated with human annotators for image captioning, surpassing SPICE, a popular task-specific metric (Anderson et al., 2016). Finally, we test the robustness of BERTSCORE on the adversarial paraphrase dataset PAWS (Zhang et al., 2019), and show that it is more robust to adversarial examples than other metrics. The code for BERTSCORE is available at `https://github.com/Tiiiger/bert_score`.

## 2 PROBLEM STATEMENT AND PRIOR METRICS

Natural language text generation is commonly evaluated using annotated reference sentences. Given a reference sentence $x$ tokenized to $k$ tokens $\langle x_1, \dots, x_k \rangle$ and a candidate $\hat{x}$ tokenized to $l$ tokens $\langle \hat{x}_1, \dots, \hat{x}_l \rangle$, a generation evaluation metric is a function $f(x, \hat{x}) \in \mathbb{R}$. Better metrics have a higher correlation with human judgments. Existing metrics can be broadly categorized into using $n$-gram matching, edit distance, embedding matching, or learned functions.

### 2.1 $n$-GRAM MATCHING APPROACHES

The most commonly used metrics for generation count the number of $n$-grams that occur in the reference $x$ and candidate $\hat{x}$. The higher the $n$ is, the more the metric is able to capture word order, but it also becomes more restrictive and constrained to the exact form of the reference.

Formally, let $S_x^n$ and $S_{\hat{x}}^n$ be the lists of token $n$-grams ($n \in \mathbb{Z}_+$) in the reference $x$ and candidate $\hat{x}$ sentences. The number of matched $n$-grams is $\sum_{w \in S_{\hat{x}}^n} \mathbb{I}[w \in S_x^n]$, where $\mathbb{I}[\cdot]$ is an indicator function. The exact match precision (Exact-$P_n$) and recall (Exact-$R_n$) scores are:

$$\text{Exact-P}_n = \frac{\sum_{w \in S_{\hat{x}}^n} \mathbb{I}[w \in S_x^n]}{|S_{\hat{x}}^n|} \quad \text{and} \quad \text{Exact-R}_n = \frac{\sum_{w \in S_x^n} \mathbb{I}[w \in S_{\hat{x}}^n]}{|S_x^n|}.$$

Several popular metrics build upon one or both of these exact matching scores.

**BLEU** The most widely used metric in machine translation is BLEU (Papineni et al., 2002), which includes three modifications to Exact-$P_n$. First, each $n$-gram in the reference can be matched at most once. Second, the number of exact matches is accumulated for all reference-candidate pairs in the corpus and divided by the total number of $n$-grams in all candidate sentences. Finally, very short candidates are discouraged using a brevity penalty. Typically, BLEU is computed for multiple values of $n$ (e.g. $n = 1, 2, 3, 4$) and the scores are averaged geometrically. A smoothed variant, SENT-BLEU (Koehn et al., 2007) is computed at the sentence level. In contrast to BLEU, BERTSCORE is not restricted to maximum $n$-gram length, but instead relies on contextualized embeddings that are able to capture dependencies of potentially unbounded length.

**METEOR** METEOR (Banerjee & Lavie, 2005) computes Exact-$P_1$ and Exact-$R_1$ while allowing backing-off from exact unigram matching to matching word stems, synonyms, and paraphrases. For example, *running* may match *run* if no exact match is possible. Non-exact matching uses an external stemmer, a synonym lexicon, and a paraphrase table. METEOR 1.5 (Denkowski & Lavie, 2014) weighs content and function words differently, and also applies importance weighting to different matching types. The more recent METEOR++ 2.0 (Guo & Hu, 2019) further incorporates a learned external paraphrase resource. Because METEOR requires external resources, only five languages are supported with the full feature set, and eleven are partially supported. Similar to METEOR, BERTSCORE allows relaxed matches, but relies on BERT embeddings that are trained on large amounts of raw text and are currently available for 104 languages. BERTSCORE also supports importance weighting, which we estimate with simple corpus statistics.

**Other Related Metrics** NIST (Doddington, 2002) is a revised version of BLEU that weighs each $n$-gram differently and uses an alternative brevity penalty. $\Delta$BLEU (Galley et al., 2015) modifies multi-reference BLEU by including human annotated negative reference sentences. CHRF (Popović, 2015) compares character $n$-grams in the reference and candidate sentences. CHRF++ (Popović, 2017) extends CHRF to include word bigram matching. ROUGE (Lin, 2004) is a commonly used metric for summarization evaluation. ROUGE-$n$ (Lin, 2004) computes Exact-$R_n$ (usually $n = 1, 2$), while ROUGE-$L$ is a variant of Exact-$R_1$ with the numerator replaced by the length of the longest common subsequence. CIDEr (Vedantam et al., 2015) is an image captioning metric that computes

cosine similarity between tf–idf weighted $n$-grams. We adopt a similar approach to weigh tokens differently. Finally, Chaganty et al. (2018) and Hashimoto et al. (2019) combine automatic metrics with human judgments for text generation evaluation.

## 2.2 EDIT-DISTANCE-BASED METRICS

Several methods use word edit distance or word error rate (Levenshtein, 1966), which quantify similarity using the number of edit operations required to get from the candidate to the reference. TER (Snover et al., 2006) normalizes edit distance by the number of reference words, and ITER (Panja & Naskar, 2018) adds stem matching and better normalization. PER (Tillmann et al., 1997) computes position independent error rate, CDER (Leusch et al., 2006) models block reordering as an edit operation. CHARACTER (Wang et al., 2016) and EED (Stanchev et al., 2019) operate on the character level and achieve higher correlation with human judgements on some languages.

## 2.3 EMBEDDING-BASED METRICS

Word embeddings (Mikolov et al., 2013; Pennington et al., 2014; Grave et al., 2018; Nguyen et al., 2017; Athiwaratkun et al., 2018) are learned dense token representations. MEANT 2.0 (Lo, 2017) uses word embeddings and shallow semantic parses to compute lexical and structural similarity. YISI-1 (Lo et al., 2018) is similar to MEANT 2.0, but makes the use of semantic parses optional. Both methods use a relatively simple similarity computation, which inspires our approach, including using greedy matching (Corley & Mihalcea, 2005) and experimenting with a similar importance weighting to YISI-1. However, we use contextual embeddings, which capture the specific use of a token in a sentence, and potentially capture sequence information. We do not use external tools to generate linguistic structures, which makes our approach relatively simple and portable to new languages. Instead of greedy matching, WMD (Kusner et al., 2015), WMD$_O$ (Chow et al., 2019), and SMS (Clark et al., 2019) propose to use optimal matching based on earth mover's distance (Rubner et al., 1998). The tradeoff[1] between greedy and optimal matching was studied by Rus & Lintean (2012). Sharma et al. (2018) compute similarity with sentence-level representations. In contrast, our token-level computation allows us to weigh tokens differently according to their importance.

## 2.4 LEARNED METRICS

Various metrics are trained to optimize correlation with human judgments. BEER (Stanojević & Sima'an, 2014) uses a regression model based on character $n$-grams and word bigrams. BLEND (Ma et al., 2017) uses regression to combine 29 existing metrics. RUSE (Shimanaka et al., 2018) combines three pre-trained sentence embedding models. All these methods require costly human judgments as supervision for each dataset, and risk poor generalization to new domains, even within a known language and task (Chaganty et al., 2018). Cui et al. (2018) and Lowe et al. (2017) train a neural model to predict if the input text is human-generated. This approach also has the risk of being optimized to existing data and generalizing poorly to new data. In contrast, the model underlying BERTSCORE is not optimized for any specific evaluation task.

## 3 BERTSCORE

Given a reference sentence $x = \langle x_1, \dots, x_k \rangle$ and a candidate sentence $\hat{x} = \langle \hat{x}_1, \dots, \hat{x}_l \rangle$, we use contextual embeddings to represent the tokens, and compute matching using cosine similarity, optionally weighted with inverse document frequency scores. Figure 1 illustrates the computation.

**Token Representation** We use contextual embeddings to represent the tokens in the input sentences $x$ and $\hat{x}$. In contrast to prior word embeddings (Mikolov et al., 2013; Pennington et al., 2014), contextual embeddings, such as BERT (Devlin et al., 2019) and ELMO (Peters et al., 2018), can generate different vector representations for the same word in different sentences depending on the surrounding words, which form the context of the target word. The models used to generate these embeddings are most commonly trained using various language modeling objectives, such as masked word prediction (Devlin et al., 2019).

---

[1]We provide an ablation study of this design choice in Appendix C.

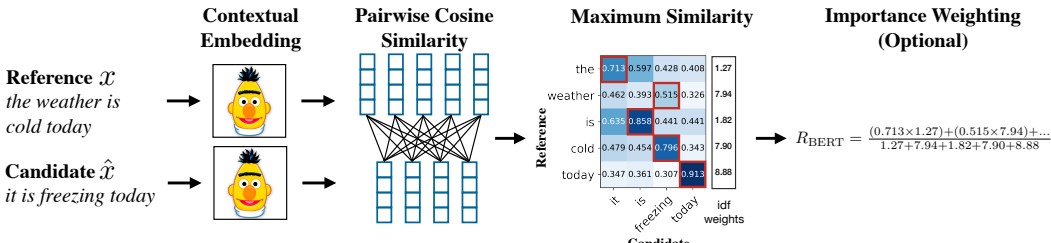

Figure 1: Illustration of the computation of the recall metric $R_{\text{BERT}}$. Given the reference $x$ and candidate $\hat{x}$, we compute BERT embeddings and pairwise cosine similarity. We highlight the greedy matching in red, and include the optional $\text{idf}$ importance weighting.

We experiment with different models (Section 4), using the tokenizer provided with each model. Given a tokenized reference sentence $x = \langle x_1, \ldots, x_k \rangle$, the embedding model generates a sequence of vectors $\langle \mathbf{x}_1, \ldots, \mathbf{x}_k \rangle$. Similarly, the tokenized candidate $\hat{x} = \langle \hat{x}_1, \ldots, \hat{x}_m \rangle$ is mapped to $\langle \hat{\mathbf{x}}_1, \ldots, \hat{\mathbf{x}}_l \rangle$. The main model we use is BERT, which tokenizes the input text into a sequence of word pieces (Wu et al., 2016), where unknown words are split into several commonly observed sequences of characters. The representation for each word piece is computed with a Transformer encoder (Vaswani et al., 2017) by repeatedly applying self-attention and nonlinear transformations in an alternating fashion. BERT embeddings have been shown to benefit various NLP tasks (Devlin et al., 2019; Liu, 2019; Huang et al., 2019; Yang et al., 2019a).

**Similarity Measure** The vector representation allows for a soft measure of similarity instead of exact-string (Papineni et al., 2002) or heuristic (Banerjee & Lavie, 2005) matching. The cosine similarity of a reference token $x_i$ and a candidate token $\hat{x}_j$ is $\frac{\mathbf{x}_i^\top \hat{\mathbf{x}}_j}{\|\mathbf{x}_i\|\|\hat{\mathbf{x}}_j\|}$. We use pre-normalized vectors, which reduces this calculation to the inner product $\mathbf{x}_i^\top \hat{\mathbf{x}}_j$. While this measure considers tokens in isolation, the contextual embeddings contain information from the rest of the sentence.

**BERTSCORE** The complete score matches each token in $x$ to a token in $\hat{x}$ to compute recall, and each token in $\hat{x}$ to a token in $x$ to compute precision. We use greedy matching to maximize the matching similarity score,[2] where each token is matched to the most similar token in the other sentence. We combine precision and recall to compute an F1 measure. For a reference $x$ and candidate $\hat{x}$, the recall, precision, and F1 scores are:

$$R_{\text{BERT}} = \frac{1}{|x|} \sum_{x_i \in x} \max_{\hat{x}_j \in \hat{x}} \mathbf{x}_i^\top \hat{\mathbf{x}}_j \ , \quad P_{\text{BERT}} = \frac{1}{|\hat{x}|} \sum_{\hat{x}_j \in \hat{x}} \max_{x_i \in x} \mathbf{x}_i^\top \hat{\mathbf{x}}_j \ , \quad F_{\text{BERT}} = 2 \frac{P_{\text{BERT}} \cdot R_{\text{BERT}}}{P_{\text{BERT}} + R_{\text{BERT}}} \ .$$

**Importance Weighting** Previous work on similarity measures demonstrated that rare words can be more indicative for sentence similarity than common words (Banerjee & Lavie, 2005; Vedantam et al., 2015). BERTSCORE enables us to easily incorporate importance weighting. We experiment with inverse document frequency ($\text{idf}$) scores computed from the test corpus. Given $M$ reference sentences $\{x^{(i)}\}_{i=1}^M$, the $\text{idf}$ score of a word-piece token $w$ is

$$\text{idf}(w) = -\log \frac{1}{M} \sum_{i=1}^M \mathbb{I}[w \in x^{(i)}] \ ,$$

where $\mathbb{I}[\cdot]$ is an indicator function. We do not use the full tf-idf measure because we process single sentences, where the term frequency ($\text{tf}$) is likely 1. For example, recall with $\text{idf}$ weighting is

$$R_{\text{BERT}} = \frac{\sum_{x_i \in x} \text{idf}(x_i) \max_{\hat{x}_j \in \hat{x}} \mathbf{x}_i^\top \hat{\mathbf{x}}_j}{\sum_{x_i \in x} \text{idf}(x_i)} \ .$$

Because we use reference sentences to compute $\text{idf}$, the $\text{idf}$ scores remain the same for all systems evaluated on a specific test set. We apply plus-one smoothing to handle unknown word pieces.

---

[2]We compare greedy matching with optimal assignment in Appendix C.

**Baseline Rescaling**   Because we use pre-normalized vectors, our computed scores have the same numerical range of cosine similarity (between $-1$ and $1$). However, in practice we observe scores in a more limited range, potentially because of the learned geometry of contextual embeddings. While this characteristic does not impact BERTSCORE's capability to rank text generation systems, it makes the actual score less readable. We address this by rescaling BERTSCORE with respect to its empirical lower bound $b$ as a baseline. We compute $b$ using Common Crawl monolingual datasets.[3] For each language and contextual embedding model, we create 1M candidate-reference pairs by grouping two random sentences. Because of the random pairing and the corpus diversity, each pair has very low lexical and semantic overlapping.[4] We compute $b$ by averaging BERTSCORE computed on these sentence pairs. Equipped with baseline $b$, we rescale BERTSCORE linearly. For example, the rescaled value $\hat{R}_{\text{BERT}}$ of $R_{\text{BERT}}$ is:

$$\hat{R}_{\text{BERT}} = \frac{R_{\text{BERT}} - b}{1 - b} \ .$$

After this operation $\hat{R}_{\text{BERT}}$ is typically between 0 and 1. We apply the same rescaling procedure for $P_{\text{BERT}}$ and $F_{\text{BERT}}$. This method does not affect the ranking ability and human correlation of BERTSCORE, and is intended solely to increase the score readability.

## 4   EXPERIMENTAL SETUP

We evaluate our approach on machine translation and image captioning.

**Contextual Embedding Models**   We evaluate twelve pre-trained contextual embedding models, including variants of BERT (Devlin et al., 2019), RoBERTa (Liu et al., 2019b), XLNet (Yang et al., 2019b), and XLM (Lample & Conneau, 2019). We present the best-performing models in Section 5. We use the 24-layer RoBERTa$_{\text{large}}$ model[5] for English tasks, 12-layer BERT$_{\text{chinese}}$ model for Chinese tasks, and the 12-layer cased multilingual BERT$_{\text{multi}}$ model for other languages.[6] We show the performance of all other models in Appendix F. Contextual embedding models generate embedding representations at every layer in the encoder network. Past work has shown that intermediate layers produce more effective representations for semantic tasks (Liu et al., 2019a). We use the WMT16 dataset  (Bojar et al., 2016) as a validation set to select the best layer of each model (Appendix B).

**Machine Translation**   Our main evaluation corpus is the WMT18 metric evaluation dataset (Ma et al., 2018), which contains predictions of 149 translation systems across 14 language pairs, gold references, and two types of human judgment scores. Segment-level human judgments assign a score to each reference-candidate pair. System-level human judgments associate each system with a single score based on all pairs in the test set. WMT18 includes translations from English to Czech, German, Estonian, Finnish, Russian, and Turkish, and from the same set of languages to English. We follow the WMT18 standard practice and use absolute Pearson correlation $|\rho|$ and Kendall rank correlation $\tau$ to evaluate metric quality, and compute significance with the Williams test (Williams, 1959) for $|\rho|$ and bootstrap re-sampling for $\tau$ as suggested by Graham & Baldwin (2014). We compute system-level scores by averaging BERTSCORE for every reference-candidate pair. We also experiment with hybrid systems by randomly sampling one candidate sentence from one of the available systems for each reference sentence (Graham & Liu, 2016). This enables system-level experiments with a higher number of systems. Human judgments of each hybrid system are created by averaging the WMT18 segment-level human judgments for the corresponding sentences in the sampled data. We compare BERTSCOREs to one canonical metric for each category introduced in Section 2, and include the comparison with all other participating metrics from WMT18 in Appendix F.

In addition to the standard evaluation, we design model selection experiments. We use 10K hybrid systems super-sampled from WMT18. We randomly select 100 out of 10K hybrid systems, and rank them using the automatic metrics. We repeat this process 100K times. We report the percentage of the metric ranking agreeing with the human ranking on the best system (Hits@1). In Tables 23-28,

---

[3]https://commoncrawl.org/

[4]BLEU computed on these pairs is around zero.

[5]We use the tokenizer provided with each model. For all Hugging Face models that use the GPT-2 tokenizer, at the time of our experiments, the tokenizer adds a space to the beginning of each sentence.

[6]All the models used are from https://github.com/huggingface/pytorch-transformers.

| Metric | en↔cs (5/5) | en↔de (16/16) | en↔et (14/14) | en↔fi (9/12) | en↔ru (8/9) | en↔tr (5/8) | en↔zh (14/14) |
|---|---|---|---|---|---|---|---|
| BLEU | .970/**.995** | .971/**.981** | **.986**/.975 | .973/.962 | .979/**.983** | **.657**/.826 | .978/.947 |
| ITER | .975/.915 | .990/**.984** | .975/**.981** | **.996**/.973 | .937/.975 | **.861**/.865 | .980/ – |
| RUSE | .981/ – | .997/ – | **.990**/ – | .991/ – | **.988**/ – | **.853**/ – | .981/ – |
| YiSi-1 | .950/**.987** | .992/**.985** | .979/**.979** | .973/.940 | **.991**/.992 | .958/**.976** | .951/**.963** |
| $P_{\text{BERT}}$ | .980/**.994** | **.998**/.988 | **.990**/.981 | .995/.957 | .982/**.990** | .791/**.935** | .981/.954 |
| $R_{\text{BERT}}$ | **.998**/.997 | .997/**.990** | .986/**.980** | **.997**/.980 | .995/.989 | .054/.879 | **.990**/.976 |
| $F_{\text{BERT}}$ | **.990**/.997 | **.999**/.989 | .990/**.982** | **.998**/.972 | .990/.990 | .499/.908 | **.988**/.967 |
| $F_{\text{BERT}}$ (idf) | .985/**.995** | **.999**/.990 | **.992**/.981 | .992/**.972** | **.991**/.991 | .826/**.941** | **.989**/.973 |

Table 1: Absolute Pearson correlations with system-level human judgments on WMT18. For each language pair, the left number is the to-English correlation, and the right is the from-English. We bold correlations of metrics not significantly outperformed by any other metric under Williams Test for that language pair and direction. The numbers in parenthesis are the number of systems used for each language pair and direction.

| Metric | en↔cs | en↔de | en↔et | en↔fi | en↔ru | en↔tr | en↔zh |
|---|---|---|---|---|---|---|---|
| BLEU | .956/**.993** | .969/**.977** | **.981**/.971 | .962/.958 | .972/.977 | .586/.796 | .968/.941 |
| ITER | .966/.865 | .990/.978 | .975/**.982** | .989/.966 | .943/.965 | .742/.872 | .978/ – |
| RUSE | .974/ – | .996/ – | .988/ – | **.983**/ – | .982/ – | .780/ – | .973/ – |
| YiSi-1 | .942/.985 | .991/.983 | .976/.976 | .964/.938 | **.985**/.989 | **.881**/**.942** | .943/.957 |
| $P_{\text{BERT}}$ | .965/.989 | .995/.983 | **.990**/**.970** | .976/.951 | .976/.988 | .846/.936 | .975/.950 |
| $R_{\text{BERT}}$ | **.989**/**.995** | .997/**.991** | .982/**.979** | .989/**.977** | **.988**/.989 | .540/**.872** | **.981**/**.980** |
| $F_{\text{BERT}}$ | .978/**.993** | .998/.988 | .989/.978 | .983/.969 | .985/.989 | .760/.910 | **.981**/.969 |
| $F_{\text{BERT}}$ (idf) | .982/.995 | **.998**/.988 | **.988**/.979 | **.989**/.969 | .983/.987 | .453/.877 | .980/.963 |

Table 2: Absolute Pearson correlations with system-level human judgments on WMT18. We use 10K hybrid super-sampled systems for each language pair and direction. For each language pair, the left number is the to-English correlation, and the right is the from-English. Bolding criteria is the same as in Table 1.

we include two additional measures to the model selection study: (a) the mean reciprocal rank of the top metric-rated system according to the human ranking, and (b) the difference between the human score of the top human-rated system and that of the top metric-rated system.

Additionally, we report the same study on the WMT17 (Bojar et al., 2017) and the WMT16 (Bojar et al., 2016) datasets in Appendix F.[7] This adds 202 systems to our evaluation.

**Image Captioning** We use the human judgments of twelve submission entries from the COCO 2015 Captioning Challenge. Each participating system generates a caption for each image in the COCO validation set (Lin et al., 2014), and each image has approximately five reference captions. Following Cui et al. (2018), we compute the Pearson correlation with two system-level metrics: the percentage of captions that are evaluated as better or equal to human captions (M1) and the percentage of captions that are indistinguishable from human captions (M2). We compute BERTSCORE with multiple references by scoring the candidate with each available reference and returning the highest score. We compare with eight task-agnostic metrics: BLEU (Papineni et al., 2002), METEOR (Banerjee & Lavie, 2005), ROUGE-L (Lin, 2004), CIDER (Vedantam et al., 2015), BEER (Stanojević & Sima'an, 2014), EED (Stanchev et al., 2019), CHRF++ (Popović, 2017), and CHARACTER (Wang et al., 2016). We also compare with two task-specific metrics: SPICE (Anderson et al., 2016) and LEIC (Cui et al., 2018). SPICE is computed using the similarity of scene graphs parsed from the reference and candidate captions. LEIC is trained to predict if a caption is written by a human given the image.

---

[7] For WMT16, we only conduct segment-level experiments on to-English pairs due to errors in the dataset.

| Metric | en↔cs | en↔de | en↔et | en↔fi | en↔ru | en↔tr | en↔zh |
|---|---|---|---|---|---|---|---|
| BLEU | .134/.151 | .803/.610 | .756/.618 | .461/.088 | .228/.519 | .095/.029 | .658/.515 |
| ITER | .154/.000 | .814/.692 | .742/.733 | .475/.111 | .234/.532 | .102/.030 | .673/ – |
| RUSE | **.214**/ – | .823/ – | **.785**/ – | .487/ – | .248/ – | .109/ – | .670/ – |
| YiSi-1 | .159/.178 | .809/.671 | .749/.671 | .467/**.230** | .248/.544 | .108/**.398** | .613/.594 |
| $P_{BERT}$ | .173/.180 | .706/.663 | .764/**.771** | .498/.078 | .255/**.545** | .140/.372 | .661/.551 |
| $R_{BERT}$ | .163/**.184** | .804/**.730** | .770/.722 | .494/.148 | .260/.542 | .005/.030 | .677/**.657** |
| $F_{BERT}$ | .175/.184 | **.824**/.703 | .769/.763 | .501/.082 | .262/.544 | **.142**/.031 | .673/.629 |
| $F_{BERT}$ (idf) | .179/.178 | **.824**/.722 | .760/.764 | **.503**/.082 | **.265**/.539 | .004/.030 | **.678**/.595 |

Table 3: Model selection accuracies (Hits@1) on WMT18 hybrid systems. We report the average of 100K samples and the 0.95 confidence intervals are below $10^{-3}$. We bold the highest numbers for each language pair and direction.

| Metric | en↔cs (5k/5k) | en↔de (78k/ 20k) | en↔et (57k/32k) | en↔fi (16k/10k) | en↔ru (10k/22k) | en↔tr (9k/1k) | en↔zh (33k/29k) |
|---|---|---|---|---|---|---|---|
| BLEU | .233/.389 | .415/.620 | .285/.414 | .154/.355 | .228/.330 | .145/.261 | .178/.311 |
| ITER | .198/.333 | .396/.610 | .235/.392 | .128/.311 | .139/.291 | -.029/.236 | .144/ – |
| RUSE | .347/ – | .498/ – | .368/ – | .273/ – | .311/ – | .259/ – | .218/ – |
| YiSi-1 | .319/.496 | .488/.691 | .351/.546 | .231/.504 | .300/.407 | .234/.418 | .211/.323 |
| $P_{BERT}$ | .387/.541 | .541/.715 | .389/.549 | .283/.486 | .345/.414 | .280/.328 | .248/.337 |
| $R_{BERT}$ | .388/**.570** | .546/**.728** | .391/**.594** | **.304**/**.565** | .343/.420 | .290/**.411** | .255/**.367** |
| $F_{BERT}$ | .404/.562 | **.550**/**.728** | **.397**/.586 | .296/.546 | **.353**/.423 | .292/.399 | **.264**/.364 |
| $F_{BERT}$ (idf) | **.408**/.553 | **.550**/.721 | .395/585 | .293/.537 | .346/**.425** | **.296**/.406 | .260/.366 |

Table 4: Kendall correlations with segment-level human judgments on WMT18. For each language pair, the left number is the to-English correlation, and the right is the from-English. We bold correlations of metrics not significantly outperformed by any other metric under bootstrap sampling for that language pair and direction. The numbers in parenthesis are the number of candidate-reference sentence pairs for each language pair and direction.

## 5 RESULTS

**Machine Translation** Tables 1–3 show system-level correlation to human judgements, correlations on hybrid systems, and model selection performance. We observe that BERTSCORE is consistently a top performer. In to-English results, RUSE (Shimanaka et al., 2018) shows competitive performance. However, RUSE is a supervised method trained on WMT16 and WMT15 human judgment data. In cases where RUSE models were not made available, such as for our from-English experiments, it is not possible to use RUSE without additional data and training. Table 4 shows segment-level correlations. We see that BERTSCORE exhibits significantly higher performance compared to the other metrics. The large improvement over BLEU stands out, making BERTSCORE particularly suitable to analyze specific examples, where SENTBLEU is less reliable. In Appendix A, we provide qualitative examples to illustrate the segment-level performance difference between SENTBLEU and BERTSCORE. At the segment-level, BERTSCORE even significantly outperforms RUSE. Overall, we find that applying importance weighting using idf at times provides small benefit, but in other cases does not help. Understanding better when such importance weighting is likely to help is an important direction for future work, and likely depends on the domain of the text and the available test data. We continue without idf weighting for the rest of our experiments. While recall $R_{BERT}$, precision $P_{BERT}$, and F1 $F_{BERT}$ alternate as the best measure in different setting, F1 $F_{BERT}$ performs reliably well across all the different settings. Our overall recommendation is therefore to use F1. We present additional results using the full set of 351 systems and evaluation metrics in Tables 12–28 in the appendix, including for experiments with idf importance weighting, different contextual embedding models, and model selection.

**Image Captioning** Table 5 shows correlation results for the COCO Captioning Challenge. BERTSCORE outperforms all task-agnostic baselines by large margins. Image captioning presents a challenging evaluation scenario, and metrics based on strict $n$-gram matching, including BLEU and ROUGE, show weak correlations with human judgments. idf importance weighting shows signifi-

| Metric | M1 | M2 |
|---|---|---|
| BLEU | -0.019* | -0.005* |
| METEOR | 0.606* | 0.594* |
| ROUGE-L | 0.090* | 0.096* |
| CIDER | 0.438* | 0.440* |
| SPICE | 0.759* | 0.750* |
| LEIC | **0.939*** | **0.949*** |
| BEER | 0.491 | 0.562 |
| EED | 0.545 | 0.599 |
| CHRF++ | 0.702 | 0.729 |
| CHARACTER | 0.800 | 0.801 |
| $P_{BERT}$ | -0.105 | -0.041 |
| $R_{BERT}$ | 0.888 | 0.863 |
| $F_{BERT}$ | 0.322 | 0.350 |
| $R_{BERT}$ (idf) | **0.917** | **0.889** |

Table 5: Pearson correlation on the 2015 COCO Captioning Challenge. The M1 and M2 measures are described in Section 4. LEIC uses images as additional inputs. Numbers with * are cited from Cui et al. (2018). We bold the highest correlations of task-specific and task-agnostic metrics.

| Type | Method | QQP | PAWS$_{QQP}$ |
|---|---|---|---|
| Trained on QQP (supervised) | DecAtt | 0.939* | 0.263 |
| | DIIN | 0.952* | 0.324 |
| | BERT | **0.963*** | **0.351** |
| Trained on QQP + PAWS$_{QQP}$ (supervised) | DecAtt | - | 0.511 |
| | DIIN | - | 0.778 |
| | BERT | - | **0.831** |
| Metric (Not trained on QQP or PAWS$_{QQP}$) | BLEU | 0.707 | 0.527 |
| | METEOR | 0.755 | 0.532 |
| | ROUGE-L | 0.740 | 0.536 |
| | CHRF++ | 0.577 | 0.608 |
| | BEER | 0.741 | 0.564 |
| | EED | 0.743 | 0.611 |
| | CHARACTER | 0.698 | 0.650 |
| | $P_{BERT}$ | 0.757 | 0.687 |
| | $R_{BERT}$ | 0.744 | 0.685 |
| | $F_{BERT}$ | 0.761 | 0.685 |
| | $F_{BERT}$ (idf) | **0.777** | **0.693** |

Table 6: Area under ROC curve (AUC) on QQP and PAWS$_{QQP}$ datasets. The scores of trained DeCATT (Parikh et al., 2016), DIIN (Gong et al., 2018), and fine-tuned BERT are reported by Zhang et al. (2019). Numbers with * are scores on the held-out test set of QQP. We bold the highest correlations of task-specific and task-agnostic metrics.

cant benefit for this task, suggesting people attribute higher importance to content words. Finally, LEIC (Cui et al., 2018), a trained metric that takes images as additional inputs and is optimized specifically for the COCO data and this set of systems, outperforms all other methods.

**Speed** Despite the use of a large pre-trained model, computing BERTSCORE is relatively fast. We are able to process 192.5 candidate-reference pairs/second using a GTX-1080Ti GPU. The complete WMT18 en-de test set, which includes 2,998 sentences, takes 15.6sec to process, compared to 5.4sec with SacreBLEU (Post, 2018), a common BLEU implementation. Given the sizes of commonly used test and validation sets, the increase in processing time is relatively marginal, and BERTSCORE is a good fit for using during validation (e.g., for stopping) and testing, especially when compared to the time costs of other development stages.

## 6 ROBUSTNESS ANALYSIS

We test the robustness of BERTSCORE using adversarial paraphrase classification. We use the Quora Question Pair corpus (QQP; Iyer et al., 2017) and the adversarial paraphrases from the Paraphrase Adversaries from Word Scrambling dataset (PAWS; Zhang et al., 2019). Both datasets contain pairs of sentences labeled to indicate whether they are paraphrases or not. Positive examples in QQP are real duplicate questions, while negative examples are related, but different questions. Sentence pairs in PAWS are generated through word swapping. For example, in PAWS, *Flights from New York to Florida* may be changed to *Flights from Florida to New York* and a good classifier should identify that these two sentences are not paraphrases. PAWS includes two parts: PAWS$_{QQP}$, which is based on the QQP data, and PAWS$_{Wiki}$. We use the PAWS$_{QQP}$ development set which contains 667 sentences. For the automatic metrics, we use no paraphrase detection training data. We expect that pairs with higher scores are more likely to be paraphrases. To evaluate the automatic metrics on QQA, we use the first 5,000 sentences in the training set instead of the the test set because the test labels are not available. We treat the first sentence as the reference and the second sentence as the candidate.

Table 6 reports the area under ROC curve (AUC) for existing models and automatic metrics. We observe that supervised classifiers trained on QQP perform worse than random guess on PAWS$_{QQP}$, which shows these models predict the adversarial examples are more likely to be paraphrases. When

adversarial examples are provided in training, state-of-the-art models like DIIN (Gong et al., 2018) and fine-tuned BERT are able to identify the adversarial examples but their performance still decreases significantly from their performance on QQP. Most metrics have decent performance on QQP, but show a significant performance drop on $PAWS_{QQP}$, almost down to chance performance. This suggests these metrics fail to to distinguish the harder adversarial examples. In contrast, the performance of BERTSCORE drops only slightly, showing more robustness than the other metrics.

## 7  DISCUSSION

We propose BERTSCORE, a new metric for evaluating generated text against gold standard references. BERTSCORE is purposely designed to be simple, task agnostic, and easy to use. Our analysis illustrates how BERTSCORE resolves some of the limitations of commonly used metrics, especially on challenging adversarial examples. We conduct extensive experiments with various configuration choices for BERTSCORE, including the contextual embedding model used and the use of importance weighting. Overall, our extensive experiments, including the ones in the appendix, show that BERTSCORE achieves better correlation than common metrics, and is effective for model selection. However, there is no one configuration of BERTSCORE that clearly outperforms all others. While the differences between the top configurations are often small, it is important for the user to be aware of the different trade-offs, and consider the domain and languages when selecting the exact configuration to use. In general, for machine translation evaluation, we suggest using $F_{BERT}$, which we find the most reliable. For evaluating text generation in English, we recommend using the 24-layer RoBERTa$_{large}$ model to compute BERTSCORE. For non-English language, the multilingual BERT$_{multi}$ is a suitable choice although BERTSCORE computed with this model has less stable performance on low-resource languages. We report the optimal hyperparameter for all models we experimented with in Appendix B

Briefly following our initial preprint publication, Zhao et al. (2019) published a concurrently developed method related to ours, but with a focus on integrating contextual word embeddings with earth mover's distance (EMD; Rubner et al., 1998) rather than our simple matching process. They also propose various improvements compared to our use of contextualized embeddings. We study these improvements in Appendix C and show that integrating them into BERTSCORE makes it equivalent or better than the EMD-based approach. Largely though, the effect of the different improvements on BERTSCORE is more modest compared to their method. Shortly after our initial publication, YiSi-1 was updated to use BERT embeddings, showing improved performance (Lo, 2019). This further corroborates our findings. Other recent related work includes training a model on top of BERT to maximize the correlation with human judgments (Mathur et al., 2019) and evaluating generation with a BERT model fine-tuned on paraphrasing (Yoshimura et al., 2019). More recent work shows the potential of using BERTSCORE for training a summarization system (Li et al., 2019) and for domain-specific evaluation using SciBERT (Beltagy et al., 2019) to evaluate abstractive text summarization (Gabriel et al., 2019).

In future work, we look forward to designing new task-specific metrics that use BERTSCORE as a subroutine and accommodate task-specific needs, similar to how Wieting et al. (2019) suggests to use semantic similarity for machine translation training. Because BERTSCORE is fully differentiable, it also can be incorporated into a training procedure to compute a learning loss that reduces the mismatch between optimization and evaluation objectives.

## ACKNOWLEDGEMENT

This research is supported in part by grants from the National Science Foundation (III-1618134, III-1526012, IIS1149882, IIS-1724282, TRIPODS-1740822, CAREER-1750499), the Office of Naval Research DOD (N00014-17-1-2175), and the Bill and Melinda Gates Foundation, SAP, Zillow, Workday, and Facebook Research. We thank Graham Neubig and David Grangier for for their insightful comments. We thank the Cornell NLP community including but not limited to Claire Cardie, Tianze Shi, Alexandra Schofield, Gregory Yauney, and Rishi Bommasani. We thank Yin Cui and Guandao Yang for their help with the COCO 2015 dataset.

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

| Case | No. | Reference and Candidate Pairs | Human | $F_{\text{BERT}}$ | BLEU |
|---|---|---|---|---|---|
| $F_{\text{BERT}} >$ BLEU | 1. | $x$: At the same time Kingfisher is closing 60 B&Q outlets across the country
$\hat{x}$: At the same time, Kingfisher will close 60 B & Q stores nationwide | 38 | 125 | 530 |
| | 2. | $x$: Hewlett-Packard to cut up to 30,000 jobs
$\hat{x}$: Hewlett-Packard will reduce jobs up to 30.000 | 119 | 39 | 441 |
| | 3. | $x$: According to opinion in Hungary, Serbia is "a safe third country".
$\hat{x}$: According to Hungarian view, Serbia is a "safe third country." | 23 | 96 | 465 |
| | 4. | $x$: Experts believe November's Black Friday could be holding back spending.
$\hat{x}$: Experts believe that the Black Friday in November has put the brakes on spending | 73 | 147 | 492 |
| | 5. | $x$: And it's from this perspective that I will watch him die.
$\hat{x}$: And from this perspective, I will see him die. | 37 | 111 | 414 |
| BLEU $> F_{\text{BERT}}$ | 6. | $x$: In their view the human dignity of the man had been violated.
$\hat{x}$: Look at the human dignity of the man injured. | 500 | 470 | 115 |
| | 8. | $x$: For example when he steered a shot from Ideye over the crossbar in the 56th minute.
$\hat{x}$: So, for example, when he steered a shot of Ideye over the latte (56th). | 516 | 524 | 185 |
| | 7. | $x$: A good prank is funny, but takes moments to reverse.
$\hat{x}$: A good prank is funny, but it takes only moments before he becomes a boomerang. | 495 | 424 | 152 |
| | 9. | $x$: I will put the pressure on them and onus on them to make a decision.
$\hat{x}$: I will exert the pressure on it and her urge to make a decision. | 507 | 471 | 220 |
| | 10. | $x$: Transport for London is not amused by this flyposting "vandalism."
$\hat{x}$: Transport for London is the Plaka animal "vandalism" is not funny. | 527 | 527 | 246 |
| $F_{\text{BERT}} >$ Human | 11. | $x$: One big obstacle to access to the jobs market is the lack of knowledge of the German language.
$\hat{x}$: A major hurdle for access to the labour market are a lack of knowledge of English. | 558 | 131 | 313 |
| | 12. | $x$: On Monday night Hungary closed its 175 km long border with Serbia.
$\hat{x}$: Hungary had in the night of Tuesday closed its 175 km long border with Serbia. | 413 | 135 | 55 |
| | 13. | $x$: They got nothing, but they were allowed to keep the clothes.
$\hat{x}$: You got nothing, but could keep the clothes. | 428 | 174 | 318 |
| | 14. | $x$: A majority of Republicans don't see Trump's temperament as a problem.
$\hat{x}$: A majority of Republicans see Trump's temperament is not a problem. | 290 | 34 | 134 |
| | 15. | $x$:His car was still running in the driveway.
$\hat{x}$: His car was still in the driveway. | 299 | 49 | 71 |
| Human $> F_{\text{BERT}}$ | 16. | $x$: Currently the majority of staff are men.
$\hat{x}$: At the moment the men predominate among the staff. | 77 | 525 | 553 |
| | 17. | $x$: There are, indeed, multiple variables at play.
$\hat{x}$: In fact, several variables play a role. | 30 | 446 | 552 |
| | 18. | $x$: One was a man of about 5ft 11in tall.
$\hat{x}$: One of the men was about 1,80 metres in size. | 124 | 551 | 528 |
| | 19. | $x$: All that stuff sure does take a toll.
$\hat{x}$: All of this certainly exacts its toll. | 90 | 454 | 547 |
| | 20. | $x$: Wage gains have shown signs of picking up.
$\hat{x}$: Increases of wages showed signs of a recovery. | 140 | 464 | 514 |

Table 7: Examples sentences where similarity ranks assigned by Human, $F_{\text{BERT}}$, and BLEU differ significantly on WMT16 German-to-English evaluation task. $x$: gold reference, $\hat{x}$: candidate outputs of MT systems. Rankings assigned by Human, $F_{\text{BERT}}$, and BLEU are shown in the right three columns. The sentences are ranked by the similarity, *i.e.* rank 1 is the most similar pair assigned by a score. An ideal metric should rank similar to humans.

## A    QUALITATIVE ANALYSIS

We study BERTSCORE and SENTBLEU using WMT16 German-to-English (Bojar et al., 2016). We rank all 560 candidate-reference pairs by human score, BERTSCORE, or SENTBLEU from most similar to least similar. Ideally, the ranking assigned by BERTSCORE and SENTBLEU should be similar to the ranking assigned by the human score.

Table 7 first shows examples where BERTSCORE and SENTBLEU scores disagree about the ranking for a candidate-reference pair by a large number. We observe that BERTSCORE is effectively able to capture synonyms and changes in word order. For example, the reference and candidate sentences in pair 3 are almost identical except that the candidate replaces *opinion in Hungary* with *Hungarian view* and switches the order of the quotation mark (*"*) and *a*. While BERTSCORE ranks the pair relatively high, SENTBLEU judges the pair as dissimilar, because it cannot match synonyms and is sensitive to the small word order changes. Pair 5 shows a set of changes that preserve the semantic meaning: replacing *to cut* with *will reduce* and swapping the order of *30,000* and *jobs*. BERTSCORE ranks the candidate translation similar to the human judgment, whereas SENTBLEU ranks it much lower. We also see that SENTBLEU potentially over-rewards $n$-gram overlap, even when phrases are used very differently. In pair 6, both the candidate and the reference contain *the human dignity of the man*. Yet the two sentences convey very different meaning. BERTSCORE agrees with the human judgment and ranks the pair low. In contrast, SENTBLEU considers the pair as relatively similar because of the significant word overlap.

Figure 2: BERTSCORE visualization. The cosine similarity of each word matching in $P_{\text{BERT}}$ are color-coded.

The bottom half of Table 7 shows examples where BERTSCORE and human judgments disagree about the ranking. We observe that BERTSCORE finds it difficult to detect factual errors. For example, BERTSCORE assigns high similarity to pair 11 when the translation replaces *German language* with *English* and pair 12 where the translation incorrectly outputs *Tuesday* when it is supposed to generate *Monday*. BERTSCORE also fails to identify that *5ft 11in* is equivalent with *1.80 metres* in pair 18. As a result, BERTSCORE assigns low similarity to the eighth pair in Table 7. SENTBLEU also suffers from these limitations.

Figure 2 visualizes the BERTSCORE matching of two pairs of candidate and reference sentences. The figure illustrates how $F_{\text{BERT}}$ matches synonymous phrases, such as *imported cars* and *foreign cars*. We also see that $F_{\text{BERT}}$ effectively matches words even given a high ordering distortion, for example the token *people* in the figure.

## B   REPRESENTATION CHOICE

As suggested by previous works (Peters et al., 2018; Reimers & Gurevych, 2019), selecting a good layer or a good combination of layers from the BERT model is important. In designing BERTSCORE, we use WMT16 segment-level human judgment data as a development set to facilitate our representation choice. For Chinese models, we tune with the WMT17 "en-zh" data because the language pair "en-zh" is not available in WMT16. In Figure 3, we plot the change of human correlation of $F_{\text{BERT}}$ over different layers of BERT, RoBERTa, XLNet and XLM models. Based on results from different models, we identify a common trend that $F_{\text{BERT}}$ computed with the intermediate representations tends to work better. We tune the number of layer to use for a range of publicly available models.[8] Table 8 shows the results of our hyperparameter search.

| Model | Total Number of Layers | Best Layer |
|---|---|---|
| bert-base-uncased | 12 | 9 |
| bert-large-uncased | 24 | 18 |
| bert-base-cased-finetuned-mrpc | 12 | 9 |
| bert-base-multilingual-cased | 12 | 9 |
| bert-base-chinese | 12 | 8 |
| roberta-base | 12 | 10 |
| roberta-large | 24 | 17 |
| roberta-large-mnli | 24 | 19 |
| xlnet-base-cased | 12 | 5 |
| xlnet-large-cased | 24 | 7 |
| xlm-mlm-en-2048 | 12 | 7 |
| xlm-mlm-100-1280 | 16 | 11 |

Table 8: Recommended layer of representation to use for BERTSCORE. The layers are chosen based on a held-out validation set (WMT16).

---

[8]https://huggingface.co/pytorch-transformers/pretrained_models.html

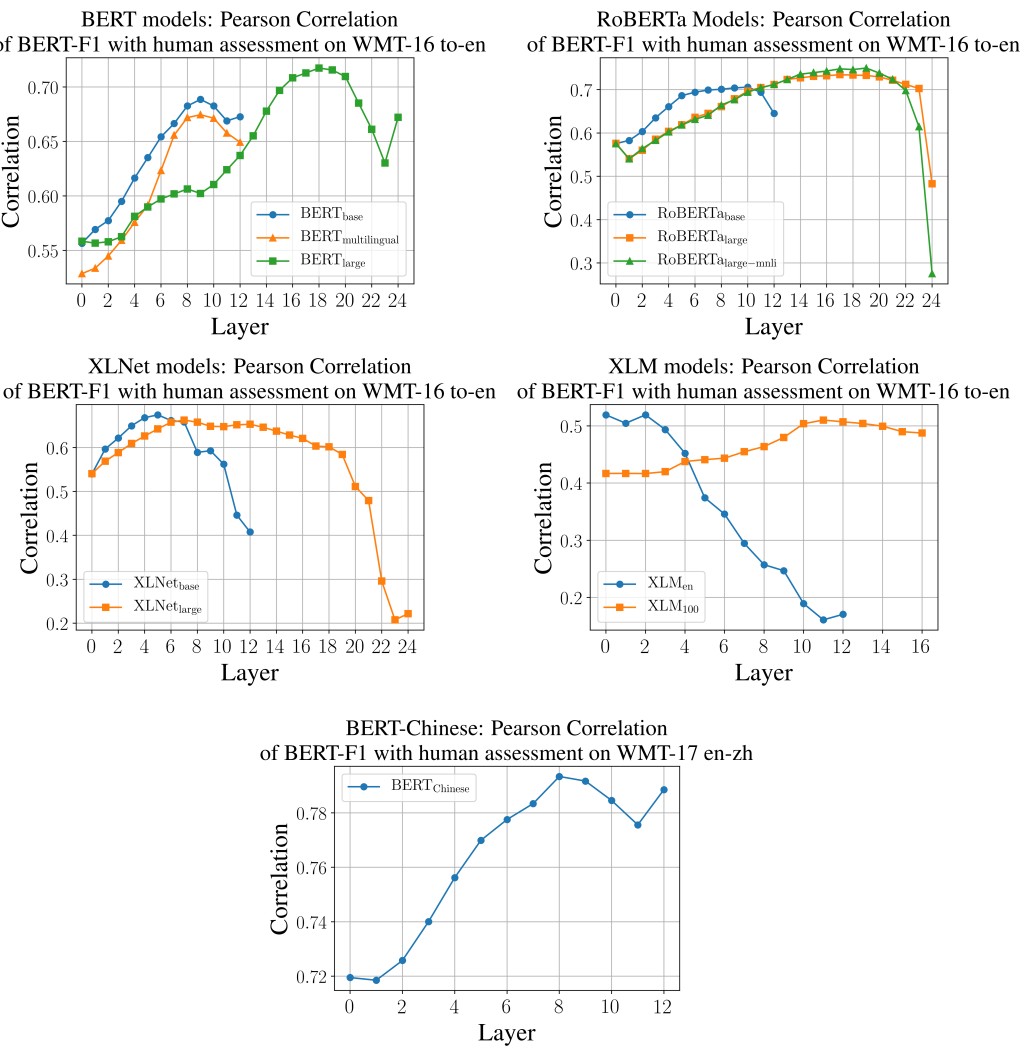

Figure 3: Pearson correlation of $F_{\text{BERT}}$ computed with different models, across different layers, with segment-level human judgments on the WMT16 to-English machine translation task. The WMT17 English-Chinese data is used for the BERT Chinese model. Layer 0 corresponds to using BPE embeddings. Consistently, correlation drops significantly in the final layers.

## C   ABLATION STUDY OF MOVERSCORE

Word Mover's Distance (WMD; Kusner et al., 2015) is a semantic similarity metric that relies on word embeddings and optimal transport. MOVERSCORE (Zhao et al., 2019) combines contextual embeddings and WMD for text generation evaluation. In contrast, BERTSCORE adopts a greedy approach to aggregate token-level information. In addition to using WMD for generation evaluation, Zhao et al. (2019) also introduce various other improvements. We do a detailed ablation study to understand the benefit of each improvement, and to investigate whether it can be applied to BERTSCORE. We use a 12-layer uncased BERT model on the WMT17 to-English segment-level data, the same setting as Zhao et al. (2019).

We identify several differences between MOVERSCORE and BERTSCORE by analyzing the released source code. We isolate each difference, and mark it with a bracketed tag for our ablation study:

1. `[MNLI]` Use a BERT model fine-tuned on MNLI (Williams et al., 2018).

2. `[PMEANS]` Apply power means (Rückl et al., 2018) to aggregate the information of different layers.[9]

3. `[IDF-L]` For reference sentences, instead of computing the idf scores on the 560 sentences in the segment-level data (`[IDF-S]`), compute the idf scores on the 3,005 sentences in the system-level data.

4. `[SEP]` For candidate sentences, recompute the idf scores on the candidate sentences. The weighting of reference tokens are kept the same as in `[IDF-S]`

5. `[RM]` Exclude punctuation marks and sub-word tokens except the first sub-word in each word from the matching.

We follow the setup of Zhao et al. (2019) and use their released fine-tuned BERT model to conduct the experiments. Table 9 shows the results of our ablation study. We report correlations for the two variants of WMD Zhao et al. (2019) study: unigrams (WMD1) and bigrams (WMD2). Our $F_{\text{BERT}}$ corresponds to the vanilla setting and the importance weighted variant corresponds to the `[IDF-S]` setting. The complete MOVERSCORE metric corresponds to `[IDF-S]`+`[SEP]`+`[PMEANS]`+`[MNLI]`+`[RM]`. We make several observations. First, for all language pairs except fi-en and lv-en, we can replicate the reported performance. For these two language pairs, Zhao et al. (2019) did not release their implementations at the time of publication.[10] Second, we confirm the effectiveness of `[PMEANS]` and `[MNLI]`. In Appendix F, we study more pre-trained models and further corroborate this conclusion. However, the contribution of other techniques, including `[RM]` and `[SEP]`, seems less stable. Third, replacing greedy matching with WMD does not lead to consistent improvement. In fact, oftentimes BERTSCORE is the better metric when given the same setup. In general, for any given language pair, BERTSCORE is always among the best performing ones. Given the current results, it is not clear tht WMD is better than greedy matching for text generation evaluation.

---

[9] Zhao et al. (2019) uses the embeddings from the last five layers from BERT and L2-normalizes the embedding vectors at each layer before computing the P-MEANs and L2-normalizing the concatenated P-MEANS.

[10] A public comment on the project page indicates that some of the techniques are not applied for these two language pairs (https://github.com/AIPHES/emnlp19-moverscore/issues/1).

| Ablation | Metric | cs-en | de-en | fi-en | lv-en | ru-en | tr-en | zh-en |
|---|---|---|---|---|---|---|---|---|
| Vanilla | WMD1 | 0.628 | 0.655 | 0.795 | 0.692 | 0.701 | 0.715 | 0.699 |
| | WMD2 | 0.638 | 0.661 | 0.797 | 0.695 | 0.700 | 0.728 | 0.714 |
| | $F_{\text{BERT}}$ | 0.659 | 0.680 | 0.817 | 0.702 | 0.719 | 0.727 | 0.717 |
| IDF-S | WMD1 | 0.636 | 0.662 | 0.824 | 0.709 | 0.716 | 0.728 | 0.713 |
| | WMD2 | 0.643 | 0.662 | 0.821 | 0.708 | 0.712 | 0.732 | 0.715 |
| | $F_{\text{BERT}}$ | 0.657 | 0.681 | 0.823 | 0.713 | 0.725 | 0.718 | 0.711 |
| IDF-L | WMD1 | 0.633 | 0.659 | 0.825 | 0.708 | 0.716 | 0.727 | 0.715 |
| | WMD2 | 0.641 | 0.661 | 0.822 | 0.708 | 0.713 | 0.730 | 0.716 |
| | $F_{\text{BERT}}$ | 0.655 | 0.682 | 0.823 | 0.713 | 0.726 | 0.718 | 0.712 |
| IDF-L + SEP | WMD1 | 0.651 | 0.660 | 0.819 | 0.703 | 0.714 | 0.724 | 0.715 |
| | WMD2 | 0.659 | 0.662 | 0.816 | 0.702 | 0.712 | 0.729 | 0.715 |
| | $F_{\text{BERT}}$ | 0.664 | 0.681 | 0.818 | 0.709 | 0.724 | 0.716 | 0.710 |
| IDF-L + SEP + RM | WMD1 | 0.651 | 0.686 | 0.803 | 0.681 | **0.730** | 0.730 | 0.720 |
| | WMD2 | 0.664 | 0.687 | 0.797 | 0.679 | **0.728** | 0.735 | 0.718 |
| | $F_{\text{BERT}}$ | 0.659 | 0.695 | 0.800 | 0.683 | **0.734** | 0.722 | 0.712 |
| IDF-L + SEP + PMEANS | WMD1 | 0.658 | 0.663 | 0.820 | 0.707 | 0.717 | 0.725 | 0.712 |
| | WMD2 | 0.667 | 0.665 | 0.817 | 0.707 | 0.717 | 0.727 | 0.712 |
| | $F_{\text{BERT}}$ | **0.671** | 0.682 | 0.819 | 0.708 | 0.725 | 0.715 | 0.704 |
| IDF-L + SEP + MNLI | WMD1 | 0.659 | 0.679 | 0.822 | 0.732 | 0.718 | 0.746 | 0.725 |
| | WMD2 | 0.664 | 0.682 | 0.819 | 0.731 | 0.715 | 0.748 | 0.722 |
| | $F_{\text{BERT}}$ | 0.668 | 0.701 | 0.825 | **0.737** | 0.727 | 0.744 | 0.725 |
| IDF-L + SEP + PMEANS + MNLI | WMD1 | 0.672 | 0.686 | **0.831** | **0.738** | 0.725 | 0.753 | **0.737** |
| | WMD2 | **0.677** | 0.690 | 0.828 | **0.736** | 0.722 | **0.755** | 0.735 |
| | $F_{\text{BERT}}$ | **0.682** | 0.707 | **0.836** | **0.741** | 0.732 | 0.751 | **0.736** |
| IDF-L + SEP + PMEANS + MNLI + RM | WMD1 | 0.670 | 0.708 | 0.821 | 0.717 | **0.738** | 0.762 | **0.744** |
| | WMD2 | **0.679** | **0.709** | 0.814 | 0.716 | **0.736** | **0.762** | 0.738 |
| | $F_{\text{BERT}}$ | **0.676** | **0.717** | 0.824 | 0.719 | **0.740** | **0.757** | **0.738** |

Table 9: Ablation Study of MOVERSCORE and BERTSCORE using Pearson correlations on the WMT17 to-English segment-level data. Correlations that are not outperformed by others for that language pair under Williams Test are bolded. We observe that using WMD does not consistently improve BERTSCORE.

| Type | Metric | Meaning | Grammar | Combined |
|---|---|---|---|---|
| BERTSCORE | $P_{\text{BERT}}$ | 0.36 | 0.47 | 0.46 |
| | $R_{\text{BERT}}$ | 0.64 | 0.29 | 0.52 |
| | $F_{\text{BERT}}$ | 0.58 | 0.41 | 0.56 |
| Common metrics | BLEU | 0.46 | 0.13 | 0.33 |
| | METEOR | 0.53 | 0.11 | 0.36 |
| | ROUGE-L | 0.51 | 0.16 | 0.38 |
| | SARI | 0.50 | 0.15 | 0.37 |
| Best metrics according to Toutanova et al. (2016) | SKIP-2+RECALL+MULT-PROB | 0.59 | N/A | 0.51 |
| | PARSE-2+RECALL+MULT-MAX | N/A | 0.35 | 0.52 |
| | PARSE-2+RECALL+MULT-PROB | 0.57 | 0.35 | 0.52 |

Table 10: Pearson correlations with human judgments on the MSR Abstractive Text Compression Dataset.

# D   ADDITIONAL EXPERIMENTS ON ABSTRACTIVE TEXT COMPRESSION

We use the human judgments provided from the MSR Abstractive Text Compression Dataset (Toutanova et al., 2016) to illustrate the applicability of BERTSCORE to abstractive text compression evaluation. The data includes three types of human scores: (a) meaning: how well a compressed text preserve the meaning of the original text; (b) grammar: how grammatically correct a compressed text is; and (c) combined: the average of the meaning and the grammar scores. We follow the experimental setup of Toutanova et al. (2016) and report Pearson correlation between BERTSCORE and the three types of human scores. Table 10 shows that $R_{\text{BERT}}$ has the highest correlation with human meaning judgments, and $P_{\text{BERT}}$ correlates highly with human grammar judgments. $F_{\text{BERT}}$ provides a balance between the two aspects.

| Task | Model | BLEU | $\hat{P}_{\text{BERT}}$ | $\hat{R}_{\text{BERT}}$ | $\hat{F}_{\text{BERT}}$ | $P_{\text{BERT}}$ | $R_{\text{BERT}}$ | $F_{\text{BERT}}$ |
|---|---|---|---|---|---|---|---|---|
| WMT14 En-De | ConvS2S (Gehring et al., 2017) | 0.266 | 0.6099 | 0.6055 | 0.6075 | 0.8499 | 0.8482 | 0.8488 |
| | Transformer-big** (Ott et al., 2018) | **0.298** | **0.6587** | **0.6528** | **0.6558** | **0.8687** | **0.8664** | **0.8674** |
| | DynamicConv*** (Wu et al., 2019) | 0.297 | 0.6526 | 0.6464 | 0.6495 | 0.8664 | 0.8640 | 0.8650 |
| WMT14 En-Fr | ConvS2S (Gehring et al., 2017) | 0.408 | 0.6998 | 0.6821 | 0.6908 | 0.8876 | 0.8810 | 0.8841 |
| | Transformer-big (Ott et al., 2018) | **0.432** | 0.7148 | 0.6978 | 0.7061 | 0.8932 | 0.8869 | 0.8899 |
| | DynamicConv (Wu et al., 2019) | **0.432** | **0.7156** | **0.6989** | **0.7071** | **0.8936** | **0.8873** | **0.8902** |
| IWSLT14 De-En | Transformer-iwslt[+] (Ott et al., 2019) | 0.350 | 0.6749 | 0.6590 | 0.6672 | 0.9452 | 0.9425 | 0.9438 |
| | LightConv (Wu et al., 2019) | 0.348 | 0.6737 | 0.6542 | 0.6642 | 0.9450 | 0.9417 | 0.9433 |
| | DynamicConv (Wu et al., 2019) | 0.352 | **0.6770** | **0.6586** | **0.6681** | **0.9456** | **0.9425** | **0.9440** |

Table 11: BLEU scores and BERTSCOREs of publicly available pre-trained MT models in fairseq (Ott et al., 2019). We show both rescaled scores marked with ˆ and raw BERTSCORE scores. *: trained on unconfirmed WMT data version, **: trained on WMT16 + ParaCrawl, ***: trained on WMT16, [+]: trained by us using fairseq.

## E    BERTSCORE OF RECENT MT MODELS

Table 11 shows the BLEU scores and the BERTSCOREs of pre-trained machine translation models on WMT14 English-to-German, WMT14 English-to-French, IWSLT14 German-to-English task. We used publicly available pre-trained models from fairseq (Ott et al., 2019).[11]  Because a pre-trained Transformer model on IWSLT is not released, we trained our own using the fairseq library. We use multilingual cased BERT$_{\text{base}}$[12] for English-to-German and English-to-French pairs, and English uncased BERT$_{\text{base}}$[13] for German-to-English pairs. Interestingly, the gap between a DynamicConv (Wu et al., 2019) trained on only WMT16 and a Transformer (Ott et al., 2018) trained on WMT16 and ParaCrawl[14] (about 30× more training data) becomes larger when evaluated with BERTSCORE rather than BLEU.

---

[11] Code and pre-trained model available at https://github.com/pytorch/fairseq.

[12] Hash code: `bert-base-multilingual-cased_L9_version=0.2.0`

[13] Hash code: `roberta-large_L17_version=0.2.0`

[14] http://paracrawl.eu/download.html

## F  ADDITIONAL RESULTS

In this section, we present additional experimental results:

1. Segment-level and system-level correlation studies on three years of WMT metric evaluation task (WMT16–18)
2. Model selection study on WMT18 10K hybrid systems
3. System-level correlation study on 2015 COCO captioning challenge
4. Robustness study on PAWS-QQP.

Following BERT (Devlin et al., 2019), a variety of Transformer-based (Vaswani et al., 2017) pre-trained contextual embeddings have been proposed and released. We conduct additional experiments with four types of pre-trained embeddings: BERT, XLM (Lample & Conneau, 2019), XLNet (Yang et al., 2019b), and RoBERTa (Liu et al., 2019b). XLM (Cross-lingual Language Model) is a Transformer pre-trained on the translation language modeling of predicting masked tokens from a pair of sentence in two different languages and masked language modeling tasks using multi-lingual training data. Yang et al. (2019b) modify the Transformer architecture and pre-train it on a permutation language modeling task resulting in some improvement on top of the original BERT when fine-tuned on several downstream tasks. Liu et al. (2019b) introduce RoBERTa (Robustly optimized BERT approach) and demonstrate that an optimized BERT model is comparable to or sometimes outperforms an XLNet on downstream tasks.

We perform a comprehensive study with the following pre-trained contextual embedding models:[15]

- BERT models: `bert-base-uncased`, `bert-large-uncased`, `bert-based-chinese`, `bert-base-multilingual-cased`, and `bert-base-cased-mrpc`
- RoBERTa models: `roberta-base`, `roberta-large`, and `roberta-large-mnli`
- XLNet models: `xlnet-base-cased` and `xlnet-base-large`
- XLM models: `xlm-mlm-en-2048` and `xlm-mlm-100-1280`

### F.1  WMT CORRELATION STUDY

**Experimental setup**  Because of missing data in the released WMT16 dataset (Bojar et al., 2016), we are only able to experiment with to-English segment-level data, which contains the outputs of 50 different systems on 6 language pairs. We use this data as the validation set for hyperparameter tuning (Appendix B). Table 12 shows the Pearson correlations of all participating metrics and BERTSCOREs computed with different pre-trained models. Significance testing for this dataset does not include the baseline metrics because the released dataset does not contain the original outputs from the baseline metrics. We conduct significance testing between BERTSCORE results only.

The WMT17 dataset (Bojar et al., 2017) contains outputs of 152 different translations on 14 language pairs. We experiment on the segment-level and system-level data on both to-English and from-English language pairs. We exclude fi-en data from the segment-level experiment due to an error in the released data. We compare our results to all participating metrics and perform standard significance testing as done by Bojar et al. (2017). Tables 13–16 show the results.

The WMT18 dataset (Ma et al., 2018) contains outputs of 159 translation systems on 14 language pairs. In addition to the results in Tables 1–4, we complement the study with the correlations of all participating metrics in WMT18 and results from using different contextual models for BERTSCORE.

**Results**  Table 12–22 collectively showcase the effectiveness of BERTSCORE in correlating with human judgments. The improvement of BERTSCORE is more pronounced on the segment-level than on the system-level. We also see that more optimized or larger BERT models can produce better contextual representations (e.g., comparing $F_{\text{RoBERTa–Large}}$ and $F_{\text{BERT–Large}}$). In contrast, the smaller XLNet performs better than a large one. Based on the evidence in Figure 8 and Tables 12–22, we

---

[15]Denoted by names specified at https://huggingface.co/pytorch-transformers/pretrained_models.html.

hypothesize that the permutation language task, though leading to a good set of model weights for fine-tuning on downstream tasks, does not necessarily produce informative pre-trained embeddings for generation evaluation. We also observe that fine-tuning pre-trained models on a related task, such as natural language inference (Williams et al., 2018), can lead to better human correlation in evaluating text generation. Therefore, for evaluating English sentences, we recommend computing BERTSCORE with a 24-layer RoBERTa model fine-tuned on the MNLI dataset. For evaluating Non-English sentences, both the multilingual BERT model and the XLM model trained on 100 languages are suitable candidates. We also recommend using domain- or language-specific contextual embeddings when possible, such as using BERT Chinese models for evaluating Chinese tasks. In general, we advise users to consider the target domain and languages when selecting the exact configuration to use.

## F.2 Model Selection Study

**Experimental setup** Similar to Section 4, we use the 10K hybrid systems super-sampled from WMT18. We randomly select 100 out of 10K hybrid systems, rank them using automatic metrics, and repeat this process 100K times. We add to the results in the main paper (Table 3) performance of all participating metrics in WMT18 and results from using different contextual embedding models for BERTSCORE. We reuse the hybrid configuration and metric outputs released in WMT18. In addition to the Hits@1 measure, we evaluate the metrics using (a) mean reciprocal rank (MRR) of the top metric-rated system in human rankings, and (b) the absolute human score difference (Diff) between the top metric- and human-rated systems. Hits@1 captures a metric's ability to select the best system. The other two measures quantify the amount of error a metric makes in the selection process. Tables 23–28 show the results from these experiments.

**Results** The additional results further support our conclusion from Table 3: BERTSCORE demonstrates better model selection performance. We also observe that the supervised metric RUSE displays strong model selection ability.

## F.3 Image Captioning on COCO

We follow the experimental setup described in Section 4. Table 29 shows the correlations of several pre-trained contextual embeddings. We observe that precision-based methods such as BLEU and $P_{\text{BERT}}$ are weakly correlated with human judgments on image captioning tasks. We hypothesize that this is because human judges prefer captions that capture the main objects in a picture for image captioning. In general, $R_{\text{BERT}}$ has a high correlation, even surpassing the task-specific metric SPICE Anderson et al. (2016). While the fine-tuned RoBERTa-Large model does not result in the highest correlation, it is one of the best metrics.

## F.4 Robustness Analysis on PAWS-QQP

We present the full results of the robustness study described in Section 6 in Table 30. In general, we observe that BERTSCORE is more robust than other commonly used metrics. BERTSCORE computed with the 24-layer RoBERTa model performs the best. Fine-tuning RoBERTa-Large on MNLI (Williams et al., 2018) can significantly improve the robustness against adversarial sentences. However, a fine-tuned BERT on MRPC (Microsoft Research Paraphrasing Corpus) (Dolan & Brockett, 2005) performs worse than its counterpart.

| Setting | Metric | cs-en 560 | de-en 560 | fi-en 560 | ro-en 560 | ru-en 560 | tr-en 560 |
|---|---|---|---|---|---|---|---|
| Unsupervised | DPMF$_{\text{COMB}}$ | 0.713 | 0.584 | 0.598 | 0.627 | 0.615 | 0.663 |
| | METRICS-F | 0.696 | 0.601 | 0.557 | 0.662 | 0.618 | 0.649 |
| | COBALT-F. | 0.671 | 0.591 | 0.554 | 0.639 | 0.618 | 0.627 |
| | UPF-COBA. | 0.652 | 0.550 | 0.490 | 0.616 | 0.556 | 0.626 |
| | MPEDA | 0.644 | 0.538 | 0.513 | 0.587 | 0.545 | 0.616 |
| | CHRF2 | 0.658 | 0.457 | 0.469 | 0.581 | 0.534 | 0.556 |
| | CHRF3 | 0.660 | 0.455 | 0.472 | 0.582 | 0.535 | 0.555 |
| | CHRF1 | 0.644 | 0.454 | 0.452 | 0.570 | 0.522 | 0.551 |
| | UoW-REVAL | 0.577 | 0.528 | 0.471 | 0.547 | 0.528 | 0.531 |
| | WORDF3 | 0.599 | 0.447 | 0.473 | 0.525 | 0.504 | 0.536 |
| | WORDF2 | 0.596 | 0.445 | 0.471 | 0.522 | 0.503 | 0.537 |
| | WORDF1 | 0.585 | 0.435 | 0.464 | 0.508 | 0.497 | 0.535 |
| | SENTBLEU | 0.557 | 0.448 | 0.484 | 0.499 | 0.502 | 0.532 |
| | DTED | 0.394 | 0.254 | 0.361 | 0.329 | 0.375 | 0.267 |
| Supervised | BEER | 0.661 | 0.462 | 0.471 | 0.551 | 0.533 | 0.545 |
| Pre-Trained | $P_{\text{BERT–Base}}$ | 0.729 | 0.617 | 0.719 | 0.651 | 0.684 | 0.678 |
| | $R_{\text{BERT–Base}}$ | 0.741 | 0.639 | 0.616 | 0.693 | 0.660 | 0.660 |
| | $F_{\text{BERT–Base}}$ | 0.747 | 0.640 | 0.661 | 0.723 | 0.672 | 0.688 |
| | $P_{\text{BERT–Base}}$ (no idf) | 0.723 | 0.638 | 0.662 | 0.700 | 0.633 | 0.696 |
| | $R_{\text{BERT–Base}}$ (no idf) | 0.745 | 0.656 | 0.638 | 0.697 | 0.653 | 0.674 |
| | $F_{\text{BERT–Base}}$ (no idf) | 0.747 | 0.663 | 0.666 | 0.714 | 0.662 | 0.703 |
| | $P_{\text{BERT–Base–MRPC}}$ | 0.697 | 0.618 | 0.614 | 0.676 | 0.62 | 0.695 |
| | $R_{\text{BERT–Base–MRPC}}$ | 0.723 | 0.636 | 0.587 | 0.667 | 0.648 | 0.664 |
| | $F_{\text{BERT–Base–MRPC}}$ | 0.725 | 0.644 | 0.617 | 0.691 | 0.654 | 0.702 |
| | $P_{\text{BERT–Base–MRPC}}$ (idf) | 0.713 | 0.613 | 0.630 | 0.693 | 0.635 | 0.691 |
| | $R_{\text{BERT–Base–MRPC}}$ (idf) | 0.727 | 0.631 | 0.573 | 0.666 | 0.642 | 0.662 |
| | $F_{\text{BERT–Base–MRPC}}$ (idf) | 0.735 | 0.637 | 0.620 | 0.700 | 0.658 | 0.697 |
| | $P_{\text{BERT–Large}}$ | 0.756 | 0.671 | 0.701 | 0.723 | 0.678 | 0.706 |
| | $R_{\text{BERT–Large}}$ | 0.768 | 0.684 | 0.677 | 0.720 | 0.686 | 0.699 |
| | $F_{\text{BERT–Large}}$ | 0.774 | 0.693 | 0.705 | 0.736 | 0.701 | 0.717 |
| | $P_{\text{BERT–Large}}$ (idf) | 0.758 | 0.653 | 0.704 | 0.734 | 0.685 | 0.705 |
| | $R_{\text{BERT–Large}}$ (idf) | 0.771 | 0.680 | 0.661 | 0.718 | 0.687 | 0.692 |
| | $F_{\text{BERT–Large}}$ (idf) | 0.774 | 0.678 | 0.700 | **0.740** | 0.701 | 0.711 |
| | $P_{\text{RoBERTa–Base}}$ | 0.738 | 0.642 | 0.671 | 0.712 | 0.669 | 0.671 |
| | $R_{\text{RoBERTa–Base}}$ | 0.745 | 0.669 | 0.645 | 0.698 | 0.682 | 0.653 |
| | $F_{\text{RoBERTa–Base}}$ | 0.761 | 0.674 | 0.686 | 0.732 | 0.697 | 0.689 |
| | $P_{\text{RoBERTa–Base}}$ (idf) | 0.751 | 0.626 | 0.678 | 0.723 | 0.685 | 0.668 |
| | $R_{\text{RoBERTa–Base}}$ (idf) | 0.744 | 0.652 | 0.638 | 0.699 | 0.685 | 0.657 |
| | $F_{\text{RoBERTa–Base}}$ (idf) | 0.767 | 0.653 | 0.688 | 0.737 | 0.705 | 0.685 |
| | $P_{\text{RoBERTa–Large}}$ | 0.757 | 0.702 | 0.709 | 0.735 | 0.721 | 0.676 |
| | $R_{\text{RoBERTa–Large}}$ | 0.765 | 0.713 | 0.686 | 0.718 | 0.714 | 0.676 |
| | $F_{\text{RoBERTa–Large}}$ | 0.780 | **0.724** | **0.728** | **0.753** | **0.738** | 0.709 |
| | $P_{\text{RoBERTa–Large}}$ (idf) | 0.771 | 0.682 | 0.705 | 0.727 | 0.714 | 0.681 |
| | $R_{\text{RoBERTa–Large}}$ (idf) | 0.762 | 0.695 | 0.683 | 0.711 | 0.708 | 0.678 |
| | $F_{\text{RoBERTa–Large}}$ (idf) | 0.786 | 0.704 | **0.727** | **0.747** | **0.732** | 0.711 |
| | $P_{\text{RoBERTa–Large–MNLI}}$ | 0.777 | 0.718 | **0.733** | 0.744 | 0.729 | 0.747 |
| | $R_{\text{RoBERTa–Large–MNLI}}$ | 0.790 | **0.731** | 0.702 | 0.741 | 0.727 | 0.732 |
| | $F_{\text{RoBERTa–Large–MNLI}}$ | 0.795 | **0.736** | **0.733** | **0.757** | **0.744** | **0.756** |
| | $P_{\text{RoBERTa–Large–MNLI}}$ (idf) | 0.794 | 0.695 | 0.731 | 0.752 | 0.732 | 0.747 |
| | $R_{\text{RoBERTa–Large–MNLI}}$ (idf) | 0.792 | 0.706 | 0.694 | 0.737 | 0.724 | 0.733 |
| | $F_{\text{RoBERTa–Large–MNLI}}$ (idf) | **0.804** | 0.710 | **0.729** | **0.760** | **0.742** | **0.754** |
| | $P_{\text{XLNet–Base}}$ | 0.708 | 0.612 | 0.639 | 0.650 | 0.606 | 0.690 |
| | $R_{\text{XLNet–Base}}$ | 0.728 | 0.630 | 0.617 | 0.645 | 0.621 | 0.675 |
| | $F_{\text{XLNet–Base}}$ | 0.727 | 0.631 | 0.640 | 0.659 | 0.626 | 0.695 |
| | $P_{\text{XLNet–Base}}$ (idf) | 0.726 | 0.618 | 0.655 | 0.678 | 0.629 | 0.700 |
| | $R_{\text{XLNet–Base}}$ (idf) | 0.734 | 0.633 | 0.618 | 0.66 | 0.635 | 0.682 |
| | $F_{\text{XLNet–Base}}$ (idf) | 0.739 | 0.633 | 0.649 | 0.681 | 0.643 | 0.702 |
| | $P_{\text{XL-NET–LARGE}}$ | 0.710 | 0.577 | 0.643 | 0.647 | 0.616 | 0.684 |
| | $R_{\text{XL-NET–LARGE}}$ | 0.732 | 0.600 | 0.610 | 0.636 | 0.627 | 0.668 |
| | $F_{\text{XL-NET–LARGE}}$ | 0.733 | 0.600 | 0.643 | 0.655 | 0.637 | 0.691 |
| | $P_{\text{XL-NET–LARGE}}$ (idf) | 0.728 | 0.574 | 0.652 | 0.669 | 0.633 | 0.681 |
| | $R_{\text{XL-NET–LARGE}}$ (idf) | 0.735 | 0.592 | 0.597 | 0.642 | 0.629 | 0.662 |
| | $F_{\text{XL-NET–LARGE}}$ (idf) | 0.742 | 0.592 | 0.643 | 0.670 | 0.645 | 0.685 |
| | $P_{\text{XLM–En}}$ | 0.688 | 0.569 | 0.613 | 0.645 | 0.583 | 0.659 |
| | $R_{\text{XLM–En}}$ | 0.715 | 0.603 | 0.577 | 0.645 | 0.609 | 0.644 |
| | $F_{\text{XLM–En}}$ | 0.713 | 0.597 | 0.610 | 0.657 | 0.610 | 0.668 |
| | $P_{\text{XLM–En}}$ (idf) | 0.728 | 0.576 | 0.649 | 0.681 | 0.604 | 0.683 |
| | $R_{\text{XLM–En}}$ (idf) | 0.730 | 0.597 | 0.591 | 0.659 | 0.622 | 0.669 |
| | $F_{\text{XLM–En}}$ (idf) | 0.739 | 0.594 | 0.636 | 0.682 | 0.626 | 0.691 |

Table 12: Pearson correlations with segment-level human judgments on WMT16 to-English translations. Correlations of metrics not significantly outperformed by any other for that language pair are highlighted in bold. For each language pair, we specify the number of examples.

| Setting | Metric | cs-en 560 | de-en 560 | fi-en 560 | lv-en 560 | ru-en 560 | tr-en 560 | zh-en 560 |
|---|---|---|---|---|---|---|---|---|
| Unsupervised | CHRF | 0.514 | 0.531 | 0.671 | 0.525 | 0.599 | 0.607 | 0.591 |
| | CHRF++ | 0.523 | 0.534 | 0.678 | 0.520 | 0.588 | 0.614 | 0.593 |
| | MEANT 2.0 | 0.578 | 0.565 | 0.687 | 0.586 | 0.607 | 0.596 | 0.639 |
| | MEANT 2.0-NOSRL | 0.566 | 0.564 | 0.682 | 0.573 | 0.591 | 0.582 | 0.630 |
| | SENTBLEU | 0.435 | 0.432 | 0.571 | 0.393 | 0.484 | 0.538 | 0.512 |
| | TREEAGGREG | 0.486 | 0.526 | 0.638 | 0.446 | 0.555 | 0.571 | 0.535 |
| | UHH_TSKM | 0.507 | 0.479 | 0.600 | 0.394 | 0.465 | 0.478 | 0.477 |
| Supervised | AUTODA | 0.499 | 0.543 | 0.673 | 0.533 | 0.584 | 0.625 | 0.583 |
| | BEER | 0.511 | 0.530 | 0.681 | 0.515 | 0.577 | 0.600 | 0.582 |
| | BLEND | 0.594 | 0.571 | 0.733 | 0.577 | 0.622 | 0.671 | 0.661 |
| | BLEU2VEC | 0.439 | 0.429 | 0.590 | 0.386 | 0.489 | 0.529 | 0.526 |
| | NGRAM2VEC | 0.436 | 0.435 | 0.582 | 0.383 | 0.490 | 0.538 | 0.520 |
| Pre-Trained | $P_{\text{BERT–Base}}$ | 0.625 | 0.659 | 0.808 | 0.688 | 0.698 | 0.713 | 0.675 |
| | $R_{\text{BERT–Base}}$ | 0.653 | 0.645 | 0.782 | 0.662 | 0.716 | 0.716 | 0.715 |
| | $F_{\text{BERT–Base}}$ | 0.654 | 0.671 | 0.811 | 0.692 | 0.707 | 0.731 | 0.714 |
| | $P_{\text{BERT–Base}}$ (idf) | 0.626 | 0.668 | 0.819 | 0.708 | 0.719 | 0.702 | 0.667 |
| | $R_{\text{BERT–Base}}$ (idf) | 0.652 | 0.658 | 0.789 | 0.678 | 0.696 | 0.703 | 0.712 |
| | $F_{\text{BERT–Base}}$ (idf) | 0.657 | 0.680 | **0.823** | 0.712 | 0.725 | 0.718 | 0.711 |
| | $P_{\text{BERT–Base–MRPC}}$ | 0.599 | 0.630 | 0.788 | 0.657 | 0.659 | 0.710 | 0.681 |
| | $R_{\text{BERT–Base–MRPC}}$ | 0.613 | 0.620 | 0.754 | 0.616 | 0.650 | 0.685 | 0.705 |
| | $F_{\text{BERT–Base–MRPC}}$ | 0.627 | 0.647 | 0.792 | 0.656 | 0.676 | 0.717 | 0.712 |
| | $P_{\text{BERT–Base–MRPC}}$ (idf) | 0.609 | 0.630 | 0.801 | 0.680 | 0.676 | 0.712 | 0.682 |
| | $R_{\text{BERT–Base–MRPC}}$ (idf) | 0.611 | 0.628 | 0.759 | 0.633 | 0.665 | 0.687 | 0.703 |
| | $F_{\text{BERT–Base–MRPC}}$ (idf) | 0.633 | 0.649 | 0.803 | 0.678 | 0.690 | 0.719 | 0.713 |
| | $P_{\text{BERT–Large}}$ | 0.638 | 0.685 | 0.816 | 0.717 | 0.719 | 0.746 | 0.693 |
| | $R_{\text{BERT–Large}}$ | 0.661 | 0.676 | 0.782 | 0.693 | 0.705 | 0.744 | 0.730 |
| | $F_{\text{BERT–Large}}$ | 0.666 | 0.701 | 0.814 | 0.723 | 0.730 | **0.760** | 0.731 |
| | $P_{\text{BERT–Large}}$ (idf) | 0.644 | 0.692 | **0.827** | 0.728 | 0.729 | 0.734 | 0.689 |
| | $R_{\text{BERT–Large}}$ (idf) | 0.665 | 0.686 | 0.796 | 0.712 | 0.729 | 0.733 | 0.730 |
| | $F_{\text{BERT–Large}}$ (idf) | 0.671 | 0.707 | **0.829** | 0.738 | 0.745 | 0.746 | 0.729 |
| | $P_{\text{RoBERTa–Base}}$ | 0.639 | 0.663 | 0.801 | 0.689 | 0.688 | 0.700 | 0.704 |
| | $R_{\text{RoBERTa–Base}}$ | 0.648 | 0.652 | 0.768 | 0.651 | 0.669 | 0.684 | 0.734 |
| | $F_{\text{RoBERTa–Base}}$ | 0.675 | 0.683 | 0.818 | 0.693 | 0.707 | 0.718 | 0.740 |
| | $P_{\text{RoBERTa–Base}}$ (idf) | 0.629 | 0.655 | 0.804 | 0.702 | 0.711 | 0.707 | 0.700 |
| | $R_{\text{RoBERTa–Base}}$ (idf) | 0.652 | 0.646 | 0.773 | 0.667 | 0.676 | 0.689 | 0.734 |
| | $F_{\text{RoBERTa–Base}}$ (idf) | 0.673 | 0.673 | 0.823 | 0.708 | 0.719 | 0.721 | 0.739 |
| | $P_{\text{RoBERTa–Large}}$ | 0.658 | 0.724 | 0.811 | 0.743 | 0.727 | 0.720 | 0.744 |
| | $R_{\text{RoBERTa–Large}}$ | 0.685 | 0.714 | 0.778 | 0.711 | 0.718 | 0.713 | 0.759 |
| | $F_{\text{RoBERTa–Large}}$ | **0.710** | **0.745** | **0.833** | 0.756 | 0.746 | 0.751 | **0.775** |
| | $P_{\text{RoBERTa–Large}}$ (idf) | 0.644 | 0.721 | 0.815 | 0.740 | 0.734 | 0.736 | 0.734 |
| | $R_{\text{RoBERTa–Large}}$ (idf) | 0.683 | 0.705 | 0.783 | 0.718 | 0.720 | 0.726 | 0.751 |
| | $F_{\text{RoBERTa–Large}}$ (idf) | **0.703** | **0.737** | **0.838** | **0.761** | 0.752 | **0.764** | 0.767 |
| | $P_{\text{RoBERTa–Large–MNLI}}$ | 0.694 | 0.736 | 0.822 | 0.764 | 0.741 | 0.754 | 0.737 |
| | $R_{\text{RoBERTa–Large–MNLI}}$ | 0.706 | 0.725 | 0.785 | 0.732 | 0.741 | 0.750 | **0.760** |
| | $F_{\text{RoBERTa–Large–MNLI}}$ | **0.722** | **0.747** | 0.822 | 0.758 | 0.767 | 0.767 | **0.765** |
| | $P_{\text{RoBERTa–Large–MNLI}}$ (idf) | 0.686 | 0.733 | **0.836** | **0.772** | 0.760 | 0.767 | 0.738 |
| | $R_{\text{RoBERTa–Large–MNLI}}$ (idf) | 0.697 | 0.717 | 0.796 | 0.741 | 0.753 | 0.757 | **0.762** |
| | $F_{\text{RoBERTa–Large–MNLI}}$ (idf) | **0.714** | **0.740** | **0.835** | **0.774** | **0.773** | **0.776** | **0.767** |
| | $P_{\text{XLNET–Base}}$ | 0.595 | 0.579 | 0.779 | 0.632 | 0.626 | 0.688 | 0.646 |
| | $R_{\text{XLNET–Base}}$ | 0.603 | 0.560 | 0.746 | 0.617 | 0.624 | 0.689 | 0.677 |
| | $F_{\text{XLNET–Base}}$ | 0.610 | 0.580 | 0.775 | 0.636 | 0.639 | 0.700 | 0.675 |
| | $P_{\text{XLNET–Base}}$ (idf) | 0.616 | 0.603 | 0.795 | 0.665 | 0.659 | 0.693 | 0.649 |
| | $R_{\text{XLNET–Base}}$ (idf) | 0.614 | 0.583 | 0.765 | 0.640 | 0.648 | 0.697 | 0.688 |
| | $F_{\text{XLNET–Base}}$ (idf) | 0.627 | 0.603 | 0.795 | 0.663 | 0.665 | 0.707 | 0.684 |
| | $P_{\text{XLNET–Large}}$ | 0.620 | 0.622 | 0.796 | 0.648 | 0.648 | 0.694 | 0.660 |
| | $R_{\text{XLNET–Large}}$ | 0.622 | 0.601 | 0.758 | 0.628 | 0.645 | 0.684 | 0.701 |
| | $F_{\text{XLNET–Large}}$ | 0.635 | 0.627 | 0.794 | 0.654 | 0.664 | 0.705 | 0.698 |
| | $P_{\text{XLNET–Large}}$ (idf) | 0.635 | 0.633 | 0.808 | 0.673 | 0.672 | 0.688 | 0.649 |
| | $R_{\text{XLNET–Large}}$ (idf) | 0.626 | 0.611 | 0.770 | 0.646 | 0.661 | 0.682 | 0.700 |
| | $F_{\text{XLNET–Large}}$ (idf) | 0.646 | 0.636 | 0.809 | 0.675 | 0.682 | 0.700 | 0.695 |
| | $P_{\text{XLM–En}}$ | 0.565 | 0.594 | 0.769 | 0.631 | 0.649 | 0.672 | 0.643 |
| | $R_{\text{XLM–En}}$ | 0.592 | 0.586 | 0.734 | 0.618 | 0.647 | 0.673 | 0.686 |
| | $F_{\text{XLM–En}}$ | 0.595 | 0.605 | 0.768 | 0.641 | 0.664 | 0.686 | 0.683 |
| | $P_{\text{XLM–En}}$ (idf) | 0.599 | 0.618 | 0.795 | 0.670 | 0.686 | 0.690 | 0.657 |
| | $R_{\text{XLM–En}}$ (idf) | 0.624 | 0.605 | 0.768 | 0.652 | 0.680 | 0.684 | 0.698 |
| | $F_{\text{XLM–En}}$ (idf) | 0.630 | 0.624 | 0.798 | 0.676 | 0.698 | 0.698 | 0.694 |

Table 13: Absolute Pearson correlations with segment-level human judgments on WMT17 to-English translations. Correlations of metrics not significantly outperformed by any other for that language pair are highlighted in bold. For each language pair, we specify the number of examples.

| Setting | Metric | en-cs 32K $\tau$ | en-de 3K $\tau$ | en-fi 3K $\tau$ | en-lv 3K $\tau$ | en-ru 560 $|r|$ | en-tr 247 $\tau$ | en-zh 560 $|r|$ |
|---|---|---|---|---|---|---|---|---|
| Unsupervised | AUTODA | 0.041 | 0.099 | 0.204 | 0.130 | 0.511 | 0.409 | 0.609 |
| | AUTODA-TECTO | 0.336 | - | - | - | - | - | - |
| | CHRF | 0.376 | 0.336 | 0.503 | 0.420 | 0.605 | 0.466 | 0.608 |
| | CHRF+ | 0.377 | 0.325 | 0.514 | 0.421 | 0.609 | 0.474 | - |
| | CHRF++ | 0.368 | 0.328 | 0.484 | 0.417 | 0.604 | 0.466 | 0.602 |
| | MEANT 2.0 | - | 0.350 | - | - | - | - | 0.727 |
| | MEANT 2.0-NOSRL | 0.395 | 0.324 | 0.565 | 0.425 | 0.636 | 0.482 | 0.705 |
| | SENTBLEU | 0.274 | 0.269 | 0.446 | 0.259 | 0.468 | 0.377 | 0.642 |
| | TREEAGGREG | 0.361 | 0.305 | 0.509 | 0.383 | 0.535 | 0.441 | 0.566 |
| Supervised | BEER | 0.398 | 0.336 | 0.557 | 0.420 | 0.569 | 0.490 | 0.622 |
| | BLEND | - | - | - | - | 0.578 | - | - |
| | BLEU2VEC | 0.305 | 0.313 | 0.503 | 0.315 | 0.472 | 0.425 | - |
| | NGRAM2VEC | - | - | 0.486 | 0.317 | - | - | - |
| Pre-Trained | $P_{\text{BERT–Multi}}$ | 0.412 | 0.364 | 0.561 | 0.435 | 0.606 | 0.579 | 0.759 |
| | $R_{\text{BERT–Multi}}$ | 0.443 | 0.430 | 0.587 | **0.480** | **0.663** | 0.571 | **0.804** |
| | $F_{\text{BERT–Multi}}$ | 0.440 | 0.404 | 0.587 | 0.466 | **0.653** | 0.587 | **0.806** |
| | $P_{\text{BERT–Multi}}$ (idf) | 0.411 | 0.328 | 0.568 | 0.444 | 0.616 | 0.555 | 0.741 |
| | $R_{\text{BERT–Multi}}$ (idf) | 0.449 | 0.416 | **0.591** | **0.479** | **0.665** | 0.579 | 0.796 |
| | $F_{\text{BERT–Multi}}$ (idf) | 0.447 | 0.379 | 0.588 | 0.470 | **0.657** | 0.571 | 0.793 |
| | $P_{\text{XLM–100}}$ | 0.406 | 0.383 | 0.553 | 0.423 | 0.562 | 0.611 | 0.722 |
| | $R_{\text{XLM–100}}$ | 0.446 | **0.436** | 0.587 | 0.458 | 0.626 | **0.652** | 0.779 |
| | $F_{\text{XLM–100}}$ | 0.444 | 0.424 | 0.577 | 0.456 | 0.613 | 0.628 | 0.778 |
| | $P_{\text{XLM–100}}$ (idf) | 0.419 | 0.367 | 0.557 | 0.427 | 0.571 | 0.595 | 0.719 |
| | $R_{\text{XLM–100}}$ (idf) | 0.450 | 0.424 | **0.592** | 0.464 | 0.632 | **0.644** | 0.770 |
| | $F_{\text{XLM–100}}$ (idf) | **0.448** | 0.419 | 0.580 | 0.459 | 0.617 | **0.644** | 0.771 |

Table 14: Absolute Pearson correlation ($|r|$) and Kendall correlation ($\tau$) with segment-level human judgments on WMT17 from-English translations. Correlations of metrics not significantly outperformed by any other for that language pair are highlighted in bold. For each language pair, we specify the number of examples.

| Setting | Metric | cs-en 4 | de-en 11 | fi-en 6 | lv-en 9 | ru-en 9 | tr-en 10 | zh-en 16 |
|---|---|---|---|---|---|---|---|---|
| Unsupervised | BLEU | 0.971 | 0.923 | 0.903 | 0.979 | 0.912 | 0.976 | 0.864 |
| | CDER | 0.989 | 0.930 | 0.927 | 0.985 | 0.922 | 0.973 | 0.904 |
| | CHARACTER | 0.972 | 0.974 | 0.946 | 0.932 | **0.958** | 0.949 | 0.799 |
| | CHRF | 0.939 | 0.968 | 0.938 | 0.968 | 0.952 | 0.944 | 0.859 |
| | CHRF++ | 0.940 | 0.965 | 0.927 | 0.973 | 0.945 | 0.960 | 0.880 |
| | MEANT 2.0 | 0.926 | 0.950 | 0.941 | 0.970 | **0.962** | 0.932 | 0.838 |
| | MEANT 2.0-NOSRL | 0.902 | 0.936 | 0.933 | 0.963 | **0.960** | 0.896 | 0.800 |
| | NIST | **1.000** | 0.931 | 0.931 | 0.960 | 0.912 | 0.971 | 0.849 |
| | PER | 0.968 | 0.951 | 0.896 | 0.962 | 0.911 | 0.932 | 0.877 |
| | TER | 0.989 | 0.906 | 0.952 | 0.971 | 0.912 | 0.954 | 0.847 |
| | TREEAGGREG | 0.983 | 0.920 | 0.977 | 0.986 | 0.918 | 0.987 | 0.861 |
| | UHH_TSKM | 0.996 | 0.937 | 0.921 | 0.990 | 0.914 | 0.987 | 0.902 |
| | WER | 0.987 | 0.896 | 0.948 | 0.969 | 0.907 | 0.925 | 0.839 |
| Supervised | AUTODA | **0.438** | 0.959 | 0.925 | 0.973 | 0.907 | 0.916 | 0.734 |
| | BEER | 0.972 | 0.960 | 0.955 | 0.978 | 0.936 | 0.972 | 0.902 |
| | BLEND | 0.968 | 0.976 | 0.958 | 0.979 | **0.964** | 0.984 | 0.894 |
| | BLEU2VEC | 0.989 | 0.936 | 0.888 | 0.966 | 0.907 | 0.961 | 0.886 |
| | NGRAM2VEC | 0.984 | 0.935 | 0.890 | 0.963 | 0.907 | 0.955 | 0.880 |
| Pre-Trained | $P_{\text{BERT–Base}}$ | 0.975 | 0.936 | 0.991 | 0.993 | 0.918 | 0.981 | 0.892 |
| | $R_{\text{BERT–Base}}$ | **0.995** | 0.975 | 0.944 | 0.978 | 0.953 | 0.991 | 0.975 |
| | $F_{\text{BERT–Base}}$ | 0.987 | 0.961 | 0.979 | 0.991 | 0.937 | 0.991 | 0.953 |
| | $P_{\text{BERT–Base}}$ (idf) | 0.983 | 0.937 | **0.998** | **0.992** | 0.939 | 0.985 | 0.878 |
| | $R_{\text{BERT–Base}}$ (idf) | **0.997** | **0.981** | 0.962 | 0.968 | **0.977** | 0.985 | 0.949 |
| | $F_{\text{BERT–Base}}$ (idf) | 0.992 | 0.967 | **0.995** | 0.992 | 0.960 | 0.996 | 0.951 |
| | $P_{\text{BERT–Base–MRPC}}$ | 0.982 | 0.926 | 0.990 | 0.987 | 0.916 | 0.970 | 0.899 |
| | $R_{\text{BERT–Base–MRPC}}$ | **0.999** | 0.979 | 0.950 | 0.982 | 0.957 | 0.977 | **0.985** |
| | $F_{\text{BERT–Base–MRPC}}$ | 0.994 | 0.957 | 0.986 | 0.994 | 0.938 | 0.980 | 0.960 |
| | $P_{\text{BERT–Base–MRPC}}$ (idf) | 0.989 | 0.936 | 0.992 | 0.979 | 0.931 | 0.976 | 0.892 |
| | $R_{\text{BERT–Base–MRPC}}$ (idf) | 0.999 | **0.987** | 0.962 | 0.980 | **0.975** | 0.979 | 0.973 |
| | $F_{\text{BERT–Base–MRPC}}$ (idf) | 0.997 | 0.968 | **0.995** | **0.997** | 0.956 | 0.989 | 0.963 |
| | $P_{\text{BERT–Large}}$ | 0.981 | 0.937 | 0.991 | **0.996** | 0.921 | 0.987 | 0.905 |
| | $R_{\text{BERT–Large}}$ | **0.996** | 0.975 | 0.953 | 0.985 | 0.954 | 0.992 | 0.977 |
| | $F_{\text{BERT–Large}}$ | 0.990 | 0.960 | 0.981 | 0.995 | 0.938 | 0.992 | 0.957 |
| | $P_{\text{BERT–Large}}$ (idf) | 0.986 | 0.938 | **0.998** | 0.995 | 0.939 | 0.994 | 0.897 |
| | $R_{\text{BERT–Large}}$ (idf) | **0.997** | **0.982** | 0.967 | 0.979 | **0.974** | 0.992 | 0.966 |
| | $F_{\text{BERT–Large}}$ (idf) | 0.994 | 0.965 | **0.993** | 0.995 | 0.958 | **0.998** | 0.959 |
| | $P_{\text{RoBERTa–Base}}$ | 0.987 | 0.930 | 0.984 | 0.966 | 0.916 | 0.963 | 0.955 |
| | $R_{\text{RoBERTa–Base}}$ | **0.999** | 0.982 | 0.947 | 0.979 | 0.956 | 0.986 | **0.984** |
| | $F_{\text{RoBERTa–Base}}$ | 0.996 | 0.961 | 0.993 | 0.993 | 0.937 | 0.983 | 0.982 |
| | $P_{\text{RoBERTa–Base}}$ (idf) | 0.990 | 0.938 | 0.980 | 0.956 | 0.929 | 0.967 | 0.962 |
| | $R_{\text{RoBERTa–Base}}$ (idf) | 0.998 | 0.987 | 0.963 | 0.979 | 0.971 | 0.986 | 0.974 |
| | $F_{\text{RoBERTa–Base}}$ (idf) | 0.996 | 0.970 | **0.999** | **0.994** | 0.952 | 0.989 | **0.982** |
| | $P_{\text{RoBERTa–Large}}$ | 0.989 | 0.948 | 0.984 | 0.949 | 0.927 | 0.960 | 0.967 |
| | $R_{\text{RoBERTa–Large}}$ | 0.998 | **0.988** | 0.957 | 0.983 | 0.969 | 0.982 | **0.984** |
| | $F_{\text{RoBERTa–Large}}$ | 0.996 | 0.973 | **0.997** | 0.991 | 0.949 | 0.984 | **0.987** |
| | $P_{\text{RoBERTa–Large}}$ (idf) | 0.989 | 0.959 | 0.975 | 0.935 | 0.944 | 0.968 | 0.974 |
| | $R_{\text{RoBERTa–Large}}$ (idf) | 0.995 | **0.991** | 0.962 | 0.979 | **0.981** | 0.981 | 0.970 |
| | $F_{\text{RoBERTa–Large}}$ (idf) | 0.996 | 0.982 | **0.998** | 0.991 | 0.965 | 0.991 | **0.984** |
| | $P_{\text{RoBERTa–Large–MNLI}}$ | 0.994 | 0.963 | **0.995** | 0.990 | 0.944 | 0.981 | 0.974 |
| | $R_{\text{RoBERTa–Large–MNLI}}$ | 0.995 | **0.991** | 0.962 | 0.981 | 0.973 | 0.985 | **0.984** |
| | $F_{\text{RoBERTa–Large–MNLI}}$ | 0.999 | 0.982 | 0.992 | **0.996** | 0.961 | 0.988 | **0.989** |
| | $P_{\text{RoBERTa–Large–MNLI}}$ (idf) | 0.995 | 0.970 | **0.997** | 0.985 | 0.955 | 0.988 | 0.979 |
| | $R_{\text{RoBERTa–Large–MNLI}}$ (idf) | 0.994 | **0.992** | 0.967 | 0.977 | **0.983** | 0.988 | 0.972 |
| | $F_{\text{RoBERTa–Large–MNLI}}$ (idf) | 0.999 | **0.989** | 0.996 | **0.997** | 0.972 | 0.994 | **0.987** |
| | $P_{\text{XLNET–Base}}$ | 0.988 | 0.938 | 0.993 | 0.993 | 0.914 | 0.974 | 0.960 |
| | $R_{\text{XLNET–Base}}$ | **0.999** | 0.978 | 0.956 | 0.977 | 0.946 | 0.981 | **0.980** |
| | $F_{\text{XLNET–Base}}$ | 0.996 | 0.963 | 0.986 | 0.991 | 0.932 | 0.981 | 0.978 |
| | $P_{\text{XLNET–Base}}$ (idf) | 0.992 | 0.951 | **0.998** | **0.996** | 0.930 | 0.982 | 0.939 |
| | $R_{\text{XLNET–Base}}$ (idf) | 0.999 | 0.986 | 0.968 | 0.973 | 0.964 | 0.987 | 0.955 |
| | $F_{\text{XLNET–Base}}$ (idf) | 0.998 | 0.974 | **0.996** | 0.994 | 0.950 | 0.990 | 0.970 |
| | $P_{\text{XLNET–Large}}$ | 0.991 | 0.944 | 0.996 | 0.995 | 0.924 | 0.982 | 0.943 |
| | $R_{\text{XLNET–Large}}$ | 0.996 | 0.981 | 0.945 | 0.971 | 0.961 | 0.986 | 0.958 |
| | $F_{\text{XLNET–Large}}$ | **0.999** | 0.969 | 0.986 | 0.992 | 0.945 | 0.992 | 0.961 |
| | $P_{\text{XLNET–Large}}$ (idf) | 0.995 | 0.955 | **0.999** | **0.996** | 0.941 | 0.985 | 0.937 |
| | $R_{\text{XLNET–Large}}$ (idf) | 0.993 | **0.985** | 0.951 | 0.960 | **0.975** | 0.974 | 0.910 |
| | $F_{\text{XLNET–Large}}$ (idf) | **1.000** | 0.978 | **0.994** | 0.993 | 0.962 | 0.994 | 0.954 |
| | $P_{\text{XLM–En}}$ | 0.983 | 0.933 | 0.994 | 0.989 | 0.918 | 0.973 | 0.928 |
| | $R_{\text{XLM–En}}$ | **0.998** | 0.978 | 0.949 | 0.983 | 0.957 | 0.985 | 0.972 |
| | $F_{\text{XLM–En}}$ | 0.994 | 0.960 | 0.985 | **0.995** | 0.938 | 0.984 | 0.964 |
| | $P_{\text{XLM–En}}$ (idf) | 0.986 | 0.940 | **0.997** | **0.992** | 0.939 | 0.979 | 0.916 |
| | $R_{\text{XLM–En}}$ (idf) | 0.999 | **0.983** | 0.966 | 0.980 | **0.975** | 0.991 | 0.952 |
| | $F_{\text{XLM–En}}$ (idf) | 0.995 | 0.967 | **0.996** | **0.998** | 0.959 | 0.993 | 0.958 |

Table 15: Absolute Pearson correlations with system-level human judgments on WMT17 to-English translations. Correlations of metrics not significantly outperformed by any other for that language pair are highlighted in bold. For each language pair, we specify the number of systems.

| Setting | Metric | en-cs 14 | en-de 16 | en-lv 17 | en-ru 9 | en-tr 8 | en-zh 11 |
|---|---|---|---|---|---|---|---|
| Unsupervised | BLEU | 0.956 | 0.804 | 0.866 | 0.898 | 0.924 | – |
| | CDER | 0.968 | 0.813 | 0.930 | 0.924 | 0.957 | – |
| | CHARACTER | **0.981** | **0.938** | 0.897 | 0.939 | **0.975** | 0.933 |
| | CHRF | **0.976** | 0.863 | **0.955** | 0.950 | **0.991** | 0.976 |
| | CHRF++ | **0.974** | 0.852 | **0.956** | 0.945 | 0.986 | 0.976 |
| | MEANT 2.0 | – | 0.858 | – | – | – | 0.956 |
| | MEANT 2.0-NOSRL | **0.976** | 0.770 | **0.959** | 0.957 | **0.991** | 0.943 |
| | NIST | 0.962 | 0.769 | 0.935 | 0.920 | **0.986** | – |
| | PER | 0.954 | 0.687 | 0.851 | 0.887 | **0.963** | – |
| | TER | 0.955 | 0.796 | 0.909 | 0.933 | **0.967** | – |
| | TREEAGGREG | 0.947 | 0.773 | 0.927 | 0.921 | **0.983** | 0.938 |
| | UHH_TSKM | – | – | – | – | – | – |
| | WER | 0.954 | 0.802 | 0.906 | 0.934 | 0.956 | – |
| Supervised | AUTODA | **0.975** | 0.603 | 0.729 | 0.850 | 0.601 | 0.976 |
| | BEER | 0.970 | 0.842 | 0.930 | 0.944 | **0.980** | 0.914 |
| | BLEND | – | – | – | 0.953 | – | – |
| | BLEU2VEC | 0.963 | 0.810 | 0.859 | 0.903 | 0.911 | – |
| | NGRAM2VEC | – | – | 0.862 | – | – | – |
| Pre-Trained | $P_{\text{BERT–Multi}}$ | 0.959 | 0.798 | 0.960 | 0.946 | **0.981** | 0.970 |
| | $R_{\text{BERT–Multi}}$ | **0.982** | **0.909** | 0.957 | **0.980** | 0.979 | **0.994** |
| | $F_{\text{BERT–Multi}}$ | 0.976 | 0.859 | 0.959 | 0.966 | **0.980** | 0.992 |
| | $P_{\text{BERT–Multi}}$ (idf) | 0.963 | 0.760 | **0.960** | 0.947 | **0.984** | 0.971 |
| | $R_{\text{BERT–Multi}}$ (idf) | **0.985** | **0.907** | 0.955 | **0.981** | **0.984** | 0.982 |
| | $F_{\text{BERT–Multi}}$ (idf) | **0.979** | 0.841 | **0.958** | 0.968 | **0.984** | **0.991** |
| | $P_{\text{XLM–100}}$ | 0.967 | 0.825 | **0.965** | 0.953 | **0.974** | 0.977 |
| | $R_{\text{XLM–100}}$ | 0.980 | **0.902** | **0.965** | **0.982** | 0.977 | 0.979 |
| | $F_{\text{XLM–100}}$ | **0.979** | 0.868 | **0.969** | 0.971 | 0.976 | **0.986** |
| | $P_{\text{XLM–100}}$ (idf) | 0.968 | 0.809 | **0.965** | 0.955 | **0.980** | 0.975 |
| | $R_{\text{XLM–100}}$ (idf) | **0.981** | 0.894 | **0.964** | **0.984** | **0.983** | 0.968 |
| | $F_{\text{XLM–100}}$ (idf) | **0.979** | 0.856 | **0.966** | 0.973 | **0.982** | 0.979 |

Table 16: Absolute Pearson correlations with system-level human judgments on WMT17 from-English translations. Correlations of metrics not significantly outperformed by any other for that language pair are highlighted in bold. For each language pair, we specify the number of systems.

| Setting | Metric | cs-en 5K | de-en 78K | et-en 57K | fi-en 16K | ru-en 10K | tr-en 9K | zh-en 33K |
|---|---|---|---|---|---|---|---|---|
| Unsupervised | CHARACTER | 0.256 | 0.450 | 0.286 | 0.185 | 0.244 | 0.172 | 0.202 |
| | ITER | 0.198 | 0.396 | 0.235 | 0.128 | 0.139 | -0.029 | 0.144 |
| | METEOR++ | 0.270 | 0.457 | 0.329 | 0.207 | 0.253 | 0.204 | 0.179 |
| | SENTBLEU | 0.233 | 0.415 | 0.285 | 0.154 | 0.228 | 0.145 | 0.178 |
| | UHH_TSKM | 0.274 | 0.436 | 0.300 | 0.168 | 0.235 | 0.154 | 0.151 |
| | YISI-0 | 0.301 | 0.474 | 0.330 | 0.225 | 0.294 | 0.215 | 0.205 |
| | YISI-1 | 0.319 | 0.488 | 0.351 | 0.231 | 0.300 | 0.234 | 0.211 |
| | YISI-1 SRL | 0.317 | 0.483 | 0.345 | 0.237 | 0.306 | 0.233 | 0.209 |
| Supervised | BEER | 0.295 | 0.481 | 0.341 | 0.232 | 0.288 | 0.229 | 0.214 |
| | BLEND | 0.322 | 0.492 | 0.354 | 0.226 | 0.290 | 0.232 | 0.217 |
| | RUSE | 0.347 | 0.498 | 0.368 | 0.273 | 0.311 | 0.259 | 0.218 |
| Pre-Trained | $P_{\text{BERT–Base}}$ | 0.349 | 0.522 | 0.373 | 0.264 | 0.325 | 0.264 | 0.232 |
| | $R_{\text{BERT–Base}}$ | 0.370 | 0.528 | 0.378 | 0.291 | 0.333 | 0.257 | 0.244 |
| | $F_{\text{BERT–Base}}$ | 0.373 | 0.531 | 0.385 | 0.287 | 0.341 | 0.266 | 0.243 |
| | $P_{\text{BERT–Base}}$ (idf) | 0.352 | 0.524 | 0.382 | 0.27 | 0.326 | 0.277 | 0.235 |
| | $R_{\text{BERT–Base}}$ (idf) | 0.368 | 0.536 | 0.388 | 0.300 | 0.340 | 0.284 | 0.244 |
| | $F_{\text{BERT–Base}}$ (idf) | 0.375 | 0.535 | 0.393 | 0.294 | 0.339 | 0.289 | 0.243 |
| | $P_{\text{BERT–Base–MRPC}}$ | 0.343 | 0.520 | 0.365 | 0.247 | 0.333 | 0.25 | 0.227 |
| | $R_{\text{BERT–Base–MRPC}}$ | 0.370 | 0.524 | 0.373 | 0.277 | 0.34 | 0.261 | 0.244 |
| | $F_{\text{BERT–Base–MRPC}}$ | 0.366 | 0.529 | 0.377 | 0.271 | 0.342 | 0.263 | 0.242 |
| | $P_{\text{BERT–Base–MRPC}}$ (idf) | 0.348 | 0.522 | 0.371 | 0.25 | 0.318 | 0.256 | 0.224 |
| | $R_{\text{BERT–Base–MRPC}}$ (idf) | 0.379 | 0.531 | 0.383 | 0.285 | 0.339 | 0.266 | 0.242 |
| | $F_{\text{BERT–Base–MRPC}}$ (idf) | 0.373 | 0.534 | 0.383 | 0.274 | 0.342 | 0.275 | 0.242 |
| | $P_{\text{BERT–LARGE}}$ | 0.361 | 0.529 | 0.380 | 0.276 | 0.340 | 0.266 | 0.241 |
| | $R_{\text{BERT–LARGE}}$ | 0.386 | 0.532 | 0.386 | 0.297 | 0.347 | 0.268 | 0.247 |
| | $F_{\text{BERT–LARGE}}$ | 0.402 | 0.537 | 0.390 | 0.296 | 0.344 | 0.274 | 0.252 |
| | $P_{\text{BERT–LARGE}}$ (idf) | 0.377 | 0.532 | 0.390 | 0.287 | 0.342 | 0.292 | 0.246 |
| | $R_{\text{BERT–LARGE}}$ (idf) | 0.386 | 0.544 | 0.396 | 0.308 | 0.356 | 0.287 | 0.251 |
| | $F_{\text{BERT–LARGE}}$ (idf) | 0.388 | 0.545 | 0.399 | 0.309 | **0.358** | 0.300 | 0.257 |
| | $P_{\text{RoBERTa–Base}}$ | 0.368 | 0.53 | 0.371 | 0.274 | 0.318 | 0.265 | 0.235 |
| | $R_{\text{RoBERTa–Base}}$ | 0.383 | 0.536 | 0.376 | 0.283 | 0.336 | 0.253 | 0.245 |
| | $F_{\text{RoBERTa–Base}}$ | 0.391 | 0.540 | 0.383 | 0.273 | 0.339 | 0.270 | 0.249 |
| | $P_{\text{RoBERTa–Base}}$ (idf) | 0.379 | 0.528 | 0.372 | 0.261 | 0.314 | 0.265 | 0.232 |
| | $R_{\text{RoBERTa–Base}}$ (idf) | 0.389 | 0.539 | 0.384 | 0.288 | 0.332 | 0.267 | 0.245 |
| | $F_{\text{RoBERTa–Base}}$ (idf) | 0.400 | 0.540 | 0.385 | 0.274 | 0.337 | 0.277 | 0.247 |
| | $P_{\text{RoBERTa–LARGE}}$ | 0.387 | 0.541 | 0.389 | 0.283 | 0.345 | 0.280 | 0.248 |
| | $R_{\text{RoBERTa–LARGE}}$ | 0.388 | 0.546 | 0.391 | 0.304 | 0.343 | 0.290 | 0.255 |
| | $F_{\text{RoBERTa–LARGE}}$ | 0.404 | 0.550 | 0.397 | 0.296 | 0.353 | 0.292 | **0.264** |
| | $P_{\text{RoBERTa–LARGE}}$ (idf) | 0.391 | 0.540 | 0.387 | 0.280 | 0.334 | 0.284 | 0.252 |
| | $R_{\text{RoBERTa–LARGE}}$ (idf) | 0.386 | 0.548 | 0.394 | 0.305 | 0.338 | 0.295 | 0.252 |
| | $F_{\text{RoBERTa–LARGE}}$ (idf) | 0.408 | 0.550 | 0.395 | 0.293 | 0.346 | 0.296 | 0.260 |
| | $P_{\text{RoBERTa–Large–MNLI}}$ | 0.397 | 0.549 | 0.396 | 0.299 | 0.351 | 0.295 | 0.253 |
| | $R_{\text{RoBERTa–Large–MNLI}}$ | 0.404 | 0.553 | 0.393 | 0.313 | 0.351 | 0.279 | 0.253 |
| | $F_{\text{RoBERTa–Large–MNLI}}$ | **0.418** | 0.557 | 0.402 | 0.312 | 0.362 | 0.290 | 0.258 |
| | $P_{\text{RoBERTa–Large–MNLI}}$ (idf) | 0.414 | 0.552 | 0.399 | 0.301 | 0.349 | 0.306 | 0.249 |
| | $R_{\text{RoBERTa–Large–MNLI}}$ (idf) | 0.412 | 0.555 | 0.400 | **0.316** | 0.357 | 0.289 | 0.258 |
| | $F_{\text{RoBERTa–Large–MNLI}}$ (idf) | **0.417** | **0.559** | **0.403** | 0.309 | 0.357 | **0.307** | 0.258 |
| | $P_{\text{XLNet–Base}}$ | 0.335 | 0.514 | 0.359 | 0.243 | 0.308 | 0.247 | 0.232 |
| | $R_{\text{XLNet–Base}}$ | 0.351 | 0.515 | 0.362 | 0.261 | 0.311 | 0.227 | 0.232 |
| | $F_{\text{XLNet–Base}}$ | 0.351 | 0.517 | 0.365 | 0.257 | 0.315 | 0.25 | 0.237 |
| | $P_{\text{XLNet–Base}}$ (idf) | 0.339 | 0.516 | 0.366 | 0.258 | 0.307 | 0.261 | 0.236 |
| | $R_{\text{XLNet–Base}}$ (idf) | 0.364 | 0.521 | 0.371 | 0.268 | 0.317 | 0.242 | 0.238 |
| | $F_{\text{XLNet–Base}}$ (idf) | 0.355 | 0.524 | 0.374 | 0.265 | 0.320 | 0.261 | 0.241 |
| | $P_{\text{XL-NET–LARGE}}$ | 0.344 | 0.522 | 0.371 | 0.252 | 0.316 | 0.264 | 0.233 |
| | $R_{\text{XL-NET–LARGE}}$ | 0.358 | 0.524 | 0.374 | 0.275 | 0.332 | 0.249 | 0.239 |
| | $F_{\text{XL-NET–LARGE}}$ | 0.357 | **0.530** | 0.380 | 0.265 | **0.334** | 0.263 | 0.238 |
| | $P_{\text{XL-NET–LARGE}}$ (idf) | 0.348 | 0.520 | 0.373 | 0.260 | 0.319 | 0.265 | 0.235 |
| | $R_{\text{XL-NET–LARGE}}$ (idf) | 0.366 | 0.529 | 0.378 | **0.278** | 0.331 | 0.266 | **0.241** |
| | $F_{\text{XL-NET–LARGE}}$ (idf) | **0.375** | 0.530 | **0.382** | 0.274 | 0.332 | **0.274** | 0.240 |
| | $P_{\text{XLM–En}}$ | 0.349 | 0.516 | 0.366 | 0.244 | 0.310 | 0.259 | 0.233 |
| | $R_{\text{XLM–En}}$ | 0.358 | 0.518 | 0.364 | 0.264 | 0.320 | 0.244 | 0.237 |
| | $F_{\text{XLM–En}}$ | 0.358 | 0.525 | 0.373 | 0.259 | 0.322 | 0.258 | 0.238 |
| | $P_{\text{XLM–En}}$ (idf) | 0.355 | 0.527 | 0.374 | 0.254 | 0.311 | 0.28 | 0.238 |
| | $R_{\text{XLM–En}}$ (idf) | 0.362 | 0.528 | 0.376 | 0.274 | 0.333 | 0.26 | 0.24 |
| | $F_{\text{XLM–En}}$ (idf) | 0.367 | 0.531 | 0.382 | 0.273 | 0.330 | 0.275 | 0.246 |

Table 17: Kendall correlations with segment-level human judgments on WMT18 to-English translations. Correlations of metrics not significantly outperformed by any other for that language pair are highlighted in bold. For each language pair, we specify the number of examples.

| Setting | Metric | en-cs 5K | en-de 20K | en-et 32K | en-fi 10K | en-ru 22K | en-tr 1K | en-zh 29K |
|---|---|---|---|---|---|---|---|---|
| Unsupervised | CHARACTER | 0.414 | 0.604 | 0.464 | 0.403 | 0.352 | 0.404 | 0.313 |
| | ITER | 0.333 | 0.610 | 0.392 | 0.311 | 0.291 | 0.236 | - |
| | SENTBLEU | 0.389 | 0.620 | 0.414 | 0.355 | 0.330 | 0.261 | 0.311 |
| | YISI-0 | 0.471 | 0.661 | 0.531 | 0.464 | 0.394 | 0.376 | 0.318 |
| | YISI-1 | 0.496 | 0.691 | 0.546 | 0.504 | 0.407 | 0.418 | 0.323 |
| | YISI-1 SRL | - | 0.696 | - | - | - | - | 0.310 |
| Supervised | BEER | 0.518 | 0.686 | 0.558 | 0.511 | 0.403 | 0.374 | 0.302 |
| | BLEND | - | - | - | - | 0.394 | - | - |
| Pre-Trained | $P_{\text{BERT–Multi}}$ | 0.541 | 0.715 | 0.549 | 0.486 | 0.414 | 0.328 | 0.337 |
| | $R_{\text{BERT–Multi}}$ | **0.570** | **0.728** | 0.594 | 0.565 | 0.420 | 0.411 | 0.367 |
| | $F_{\text{BERT–Multi}}$ | 0.562 | **0.728** | 0.586 | 0.546 | 0.423 | 0.399 | 0.364 |
| | $P_{\text{BERT–Multi}}$ (idf) | 0.525 | 0.7 | 0.54 | 0.495 | 0.423 | 0.352 | 0.338 |
| | $R_{\text{BERT–Multi}}$ (idf) | 0.569 | 0.727 | 0.601 | 0.561 | 0.423 | 0.420 | **0.374** |
| | $F_{\text{BERT–Multi}}$ (idf) | 0.553 | 0.721 | 0.585 | 0.537 | **0.425** | 0.406 | 0.366 |
| | $P_{\text{XLM–100}}$ | 0.496 | 0.711 | 0.561 | 0.527 | 0.417 | 0.364 | 0.340 |
| | $R_{\text{XLM–100}}$ | 0.564 | 0.724 | **0.612** | 0.584 | 0.418 | 0.432 | 0.363 |
| | $F_{\text{XLM–100}}$ | 0.533 | 0.727 | 0.599 | 0.573 | 0.421 | 0.408 | 0.362 |
| | $P_{\text{XLM–100}}$ (idf) | 0.520 | 0.710 | 0.572 | 0.546 | 0.421 | 0.370 | 0.328 |
| | $R_{\text{XLM–100}}$ (idf) | 0.567 | 0.722 | 0.609 | **0.587** | 0.420 | **0.439** | 0.365 |
| | $F_{\text{XLM–100}}$ (idf) | 0.554 | 0.724 | 0.601 | 0.584 | 0.422 | 0.389 | 0.355 |

Table 18: Kendall correlations with segment-level human judgments on WMT18 from-English translations. Correlations of metrics not significantly outperformed by any other for that language pair are highlighted in bold. For each language pair, we specify the number of examples.

| Setting | Metric | cs-en 5 | de-en 16 | et-en 14 | fi-en 9 | ru-en 8 | tr-en 5 | zh-en 14 |
|---|---|---|---|---|---|---|---|---|
| Unsupervised | BLEU | 0.970 | 0.971 | **0.986** | 0.973 | 0.979 | 0.657 | 0.978 |
| | CDER | 0.972 | 0.980 | **0.990** | 0.984 | 0.980 | 0.664 | 0.982 |
| | CHARACTER | 0.970 | 0.993 | 0.979 | 0.989 | 0.991 | 0.782 | 0.950 |
| | ITER | 0.975 | 0.990 | 0.975 | **0.996** | 0.937 | **0.861** | 0.980 |
| | METEOR++ | 0.945 | 0.991 | 0.978 | 0.971 | **0.995** | 0.864 | 0.962 |
| | NIST | 0.954 | 0.984 | 0.983 | 0.975 | 0.973 | **0.970** | 0.968 |
| | PER | 0.970 | 0.985 | 0.983 | 0.993 | 0.967 | 0.159 | 0.931 |
| | TER | 0.950 | 0.970 | **0.990** | 0.968 | 0.970 | 0.533 | 0.975 |
| | UHH_TSKM | 0.952 | 0.980 | **0.989** | 0.982 | 0.980 | 0.547 | **0.981** |
| | WER | 0.951 | 0.961 | **0.991** | 0.961 | 0.968 | 0.041 | 0.975 |
| | YISI-0 | 0.956 | 0.994 | 0.975 | 0.978 | 0.988 | **0.954** | 0.957 |
| | YISI-1 | 0.950 | 0.992 | 0.979 | 0.973 | 0.991 | **0.958** | 0.951 |
| | YISI-1 SRL | 0.965 | 0.995 | 0.981 | 0.977 | 0.992 | **0.869** | 0.962 |
| Supervised | BEER | 0.958 | 0.994 | **0.985** | 0.991 | 0.982 | 0.870 | 0.976 |
| | BLEND | 0.973 | 0.991 | 0.985 | 0.994 | 0.993 | 0.801 | 0.976 |
| | RUSE | 0.981 | 0.997 | **0.990** | 0.991 | 0.988 | 0.853 | 0.981 |
| Pre-Trained | $P_{\text{BERT–Base}}$ | 0.965 | 0.995 | 0.986 | 0.973 | 0.976 | 0.941 | 0.974 |
| | $R_{\text{BERT–Base}}$ | 0.994 | 0.991 | 0.979 | 0.992 | 0.991 | 0.067 | 0.988 |
| | $F_{\text{BERT–Base}}$ | 0.982 | 0.994 | 0.983 | 0.986 | 0.985 | **0.949** | 0.984 |
| | $P_{\text{BERT–Base}}$ (idf) | 0.961 | 0.993 | 0.987 | 0.988 | 0.976 | **0.984** | 0.973 |
| | $R_{\text{BERT–Base}}$ (idf) | 0.996 | 0.994 | 0.977 | 0.995 | 0.995 | 0.874 | 0.983 |
| | $F_{\text{BERT–Base}}$ (idf) | 0.981 | 0.995 | 0.984 | 0.995 | 0.988 | **0.994** | 0.981 |
| | $P_{\text{BERT–Base–MRPC}}$ | 0.957 | 0.994 | 0.989 | 0.953 | 0.976 | 0.798 | 0.977 |
| | $R_{\text{BERT–Base–MRPC}}$ | 0.992 | 0.994 | 0.983 | 0.988 | 0.993 | 0.707 | **0.990** |
| | $F_{\text{BERT–Base–MRPC}}$ | 0.975 | 0.995 | 0.987 | 0.975 | 0.986 | 0.526 | 0.986 |
| | $P_{\text{BERT–Base–MRPC}}$ (idf) | 0.957 | 0.997 | 0.989 | 0.967 | 0.975 | 0.894 | 0.980 |
| | $R_{\text{BERT–Base–MRPC}}$ (idf) | 0.991 | 0.997 | 0.981 | 0.994 | 0.993 | 0.052 | 0.987 |
| | $F_{\text{BERT–Base–MRPC}}$ (idf) | 0.975 | **0.998** | 0.987 | 0.985 | 0.987 | 0.784 | 0.987 |
| | $P_{\text{BERT–Large}}$ | 0.978 | 0.992 | 0.987 | 0.971 | 0.977 | **0.920** | 0.978 |
| | $R_{\text{BERT–Large}}$ | 0.997 | 0.990 | 0.985 | 0.990 | 0.992 | 0.098 | **0.990** |
| | $F_{\text{BERT–Large}}$ | 0.989 | 0.992 | 0.987 | 0.983 | 0.985 | 0.784 | 0.986 |
| | $P_{\text{BERT–Large}}$ (idf) | 0.977 | 0.992 | 0.988 | 0.986 | 0.976 | **0.980** | 0.977 |
| | $R_{\text{BERT–Large}}$ (idf) | 0.998 | 0.993 | 0.983 | 0.996 | 0.995 | 0.809 | **0.986** |
| | $F_{\text{BERT–Large}}$ (idf) | 0.989 | 0.993 | 0.986 | 0.993 | 0.987 | **0.976** | 0.984 |
| | $P_{\text{RoBERTa–Base}}$ | 0.970 | 0.995 | 0.991 | **0.998** | 0.976 | 0.796 | 0.980 |
| | $R_{\text{RoBERTa–Base}}$ | 0.996 | 0.996 | 0.982 | **0.998** | 0.994 | 0.477 | **0.991** |
| | $F_{\text{RoBERTa–Base}}$ | 0.984 | 0.997 | 0.989 | **0.999** | 0.987 | 0.280 | **0.989** |
| | $P_{\text{RoBERTa–Base}}$ (idf) | 0.966 | 0.993 | 0.991 | 0.994 | 0.977 | 0.880 | **0.984** |
| | $R_{\text{RoBERTa–Base}}$ (idf) | 0.995 | **0.998** | 0.981 | **0.998** | 0.995 | 0.230 | **0.989** |
| | $F_{\text{RoBERTa–Base}}$ (idf) | 0.981 | 0.998 | 0.989 | **0.997** | 0.988 | 0.741 | **0.990** |
| | $P_{\text{RoBERTa–Large}}$ | 0.980 | **0.998** | 0.990 | 0.995 | 0.982 | 0.791 | 0.981 |
| | $R_{\text{RoBERTa–Large}}$ | 0.998 | 0.997 | 0.986 | **0.997** | **0.995** | 0.054 | **0.990** |
| | $F_{\text{RoBERTa–Large}}$ | 0.990 | **0.999** | 0.990 | **0.998** | 0.990 | 0.499 | **0.988** |
| | $P_{\text{RoBERTa–Large}}$ (idf) | 0.972 | 0.997 | **0.993** | 0.985 | 0.982 | 0.920 | 0.983 |
| | $R_{\text{RoBERTa–Large}}$ (idf) | 0.996 | **0.997** | 0.984 | **0.997** | **0.995** | 0.578 | **0.989** |
| | $F_{\text{RoBERTa–Large}}$ (idf) | 0.985 | **0.999** | 0.992 | 0.992 | 0.991 | 0.826 | **0.989** |
| | $P_{\text{RoBERTa–Large–MNLI}}$ | 0.989 | **0.998** | **0.994** | 0.998 | 0.985 | 0.908 | 0.982 |
| | $R_{\text{RoBERTa–Large–MNLI}}$ | **1.000** | 0.996 | 0.988 | 0.996 | **0.995** | 0.097 | **0.991** |
| | $F_{\text{RoBERTa–Large–MNLI}}$ | 0.996 | **0.998** | **0.992** | **0.998** | 0.992 | 0.665 | **0.989** |
| | $P_{\text{RoBERTa–Large–MNLI}}$ (idf) | 0.986 | 0.998 | **0.994** | 0.993 | 0.986 | **0.989** | 0.985 |
| | $R_{\text{RoBERTa–Large–MNLI}}$ (idf) | **0.999** | 0.997 | 0.986 | 0.997 | **0.993** | 0.633 | **0.990** |
| | $F_{\text{RoBERTa–Large–MNLI}}$ (idf) | 0.995 | **0.998** | **0.991** | 0.996 | 0.993 | 0.963 | **0.990** |
| | $P_{\text{XLNET–Base}}$ | 0.970 | 0.996 | 0.986 | 0.990 | 0.979 | 0.739 | 0.982 |
| | $R_{\text{XLNET–Base}}$ | 0.994 | 0.997 | 0.979 | 0.995 | 0.994 | 0.795 | **0.990** |
| | $F_{\text{XLNET–Base}}$ | 0.983 | 0.997 | 0.983 | 0.993 | 0.987 | 0.505 | **0.988** |
| | $P_{\text{XLNET–Base}}$ (idf) | 0.968 | **0.998** | 0.986 | 0.990 | 0.978 | 0.923 | 0.982 |
| | $R_{\text{XLNET–Base}}$ (idf) | 0.993 | **0.998** | 0.978 | **0.996** | 0.994 | 0.439 | **0.988** |
| | $F_{\text{XLNET–Base}}$ (idf) | 0.981 | **0.999** | 0.984 | 0.995 | 0.989 | 0.722 | **0.988** |
| | $P_{\text{XLNET–Large}}$ | 0.969 | **0.998** | 0.986 | 0.995 | 0.979 | 0.880 | 0.981 |
| | $R_{\text{XLNET–Large}}$ | 0.995 | 0.997 | 0.977 | **0.997** | 0.995 | 0.430 | **0.988** |
| | $F_{\text{XLNET–Large}}$ | 0.983 | **0.998** | 0.983 | **0.997** | 0.988 | 0.713 | **0.988** |
| | $P_{\text{XLNET–Large}}$ (idf) | 0.963 | 0.996 | 0.986 | 0.995 | 0.978 | **0.939** | 0.979 |
| | $R_{\text{XLNET–Large}}$ (idf) | 0.992 | 0.997 | 0.975 | 0.993 | **0.996** | 0.531 | 0.982 |
| | $F_{\text{XLNET–Large}}$ (idf) | 0.978 | 0.997 | 0.983 | **0.996** | 0.990 | 0.886 | 0.984 |
| | $P_{\text{XLM–En}}$ | 0.965 | 0.996 | 0.990 | 0.978 | 0.980 | 0.946 | 0.981 |
| | $R_{\text{XLM–En}}$ | 0.990 | 0.995 | 0.984 | **0.996** | **0.996** | 0.286 | **0.987** |
| | $F_{\text{XLM–En}}$ | 0.978 | 0.997 | 0.988 | 0.990 | 0.989 | 0.576 | 0.987 |
| | $P_{\text{XLM–En}}$ (idf) | 0.960 | 0.996 | 0.990 | 0.987 | 0.980 | **0.989** | 0.981 |
| | $R_{\text{XLM–En}}$ (idf) | 0.991 | 0.997 | 0.983 | **0.996** | **0.998** | 0.612 | 0.985 |
| | $F_{\text{XLM–En}}$ (idf) | 0.976 | **0.998** | 0.988 | 0.994 | 0.992 | **0.943** | 0.985 |

Table 19: Absolute Pearson correlations with system-level human judgments on WMT18 to-English translations. Correlations of metrics not significantly outperformed by any other for that language pair are highlighted in bold. For each language pair, we specify the number of systems.

| Setting | Metric | en-cs 5 | en-de 16 | en-et 14 | en-fi 12 | en-ru 9 | en-tr 8 | en-zh 14 |
|---|---|---|---|---|---|---|---|---|
| Unsupervised | BLEU | 0.995 | 0.981 | 0.975 | 0.962 | 0.983 | 0.826 | 0.947 |
| | CDER | 0.997 | 0.986 | 0.984 | 0.964 | 0.984 | 0.861 | 0.961 |
| | CHARACTER | **0.993** | **0.989** | 0.956 | **0.974** | 0.983 | 0.833 | **0.983** |
| | ITER | 0.915 | 0.984 | 0.981 | 0.973 | 0.975 | 0.865 | – |
| | METEOR++ | – | – | – | – | – | – | – |
| | NIST | **0.999** | 0.986 | **0.983** | 0.949 | **0.990** | 0.902 | 0.950 |
| | PER | 0.991 | 0.981 | 0.958 | 0.906 | **0.988** | 0.859 | 0.964 |
| | TER | **0.997** | 0.988 | 0.981 | 0.942 | **0.987** | 0.867 | 0.963 |
| | UHH_TSKM | – | – | – | – | – | – | – |
| | WER | **0.997** | 0.986 | 0.981 | 0.945 | 0.985 | 0.853 | 0.957 |
| | YISI-0 | 0.973 | 0.985 | 0.968 | 0.944 | **0.990** | **0.990** | 0.957 |
| | YISI-1 | **0.987** | 0.985 | 0.979 | 0.940 | **0.992** | **0.976** | 0.963 |
| | YISI-1 SRL | – | **0.990** | – | – | – | – | 0.952 |
| Supervised | BEER | **0.992** | **0.991** | 0.980 | 0.961 | **0.988** | **0.965** | 0.928 |
| | BLEND | – | – | – | – | **0.988** | – | – |
| | RUSE | – | – | – | – | – | – | – |
| Pre-Trained | $P_{\text{BERT–Multi}}$ | **0.994** | **0.988** | 0.981 | 0.957 | 0.990 | 0.935 | 0.954 |
| | $R_{\text{BERT–Multi}}$ | **0.997** | 0.990 | 0.980 | 0.980 | 0.989 | 0.879 | 0.976 |
| | $F_{\text{BERT–Multi}}$ | **0.997** | **0.989** | 0.982 | 0.972 | 0.990 | 0.908 | 0.967 |
| | $P_{\text{BERT–Multi}}$ (idf) | 0.992 | 0.986 | 0.974 | 0.954 | 0.991 | **0.969** | 0.954 |
| | $R_{\text{BERT–Multi}}$ (idf) | **0.997** | **0.993** | 0.982 | 0.982 | 0.992 | 0.901 | **0.984** |
| | $F_{\text{BERT–Multi}}$ (idf) | **0.995** | **0.990** | 0.981 | 0.972 | 0.991 | 0.941 | 0.973 |
| | $P_{\text{XLM–100}}$ | 0.984 | **0.992** | **0.993** | 0.972 | **0.993** | **0.962** | 0.965 |
| | $R_{\text{XLM–100}}$ | **0.991** | **0.992** | **0.992** | **0.989** | **0.992** | 0.895 | **0.983** |
| | $F_{\text{XLM–100}}$ | 0.988 | **0.993** | **0.993** | **0.986** | **0.993** | 0.935 | 0.976 |
| | $P_{\text{XLM–100}}$ (idf) | 0.982 | **0.992** | **0.994** | 0.975 | **0.993** | **0.968** | 0.964 |
| | $R_{\text{XLM–100}}$ (idf) | **0.993** | **0.993** | **0.991** | **0.989** | **0.993** | 0.911 | **0.986** |
| | $F_{\text{XLM–100}}$ (idf) | 0.989 | **0.993** | **0.994** | **0.985** | **0.993** | 0.945 | 0.979 |

Table 20: Absolute Pearson correlations with system-level human judgments on WMT18 from-English translations. Correlations of metrics not significantly outperformed by any other for that language pair are highlighted in bold. For each language pair, we specify the number of systems.

| Setting | Metric | cs-en 10K | de-en 10K | et-en 10K | fi-en 10K | ru-en 10K | tr-en 10K | zh-en 10K |
|---|---|---|---|---|---|---|---|---|
| Unsupervised | BLEU | 0.956 | 0.969 | 0.981 | 0.962 | 0.972 | 0.586 | 0.968 |
| | CDER | 0.964 | 0.980 | 0.988 | 0.976 | 0.974 | 0.577 | 0.973 |
| | CHARACTER | 0.960 | 0.992 | 0.975 | 0.979 | 0.984 | 0.680 | 0.942 |
| | ITER | 0.966 | 0.990 | 0.975 | 0.989 | 0.943 | 0.742 | 0.978 |
| | METEOR++ | 0.937 | 0.990 | 0.975 | 0.962 | 0.989 | 0.787 | 0.954 |
| | NIST | 0.942 | **0.982** | 0.980 | 0.965 | 0.965 | 0.862 | 0.959 |
| | PER | 0.937 | 0.982 | 0.978 | 0.983 | 0.955 | 0.043 | 0.923 |
| | TER | 0.942 | 0.970 | 0.988 | 0.960 | 0.963 | 0.450 | 0.967 |
| | UHH_TSKM | 0.943 | 0.979 | 0.987 | 0.974 | 0.973 | 0.443 | 0.972 |
| | WER | 0.942 | 0.961 | 0.989 | 0.953 | 0.962 | 0.072 | 0.967 |
| | YISI-0 | 0.947 | 0.992 | 0.972 | 0.969 | 0.982 | 0.863 | 0.950 |
| | YISI-1 | 0.942 | 0.991 | 0.976 | 0.964 | 0.985 | 0.881 | 0.943 |
| | YISI-1 SRL | 0.957 | 0.994 | 0.978 | 0.968 | 0.986 | 0.785 | **0.954** |
| Supervised | BEER | 0.950 | 0.993 | 0.983 | 0.982 | 0.976 | 0.723 | 0.968 |
| | BLEND | 0.965 | 0.990 | 0.982 | 0.985 | 0.986 | 0.724 | 0.969 |
| | RUSE | 0.974 | 0.996 | 0.988 | 0.983 | 0.982 | 0.780 | 0.973 |
| Pre-Trained | $P_{\text{BERT–Base}}$ | 0.954 | 0.992 | 0.984 | 0.980 | 0.970 | **0.917** | 0.965 |
| | $R_{\text{BERT–Base}}$ | 0.988 | 0.994 | 0.974 | 0.987 | 0.988 | 0.801 | 0.975 |
| | $F_{\text{BERT–Base}}$ | 0.973 | 0.994 | 0.981 | 0.987 | 0.982 | **0.924** | 0.973 |
| | $P_{\text{BERT–Base}}$ (idf) | 0.957 | 0.994 | 0.983 | 0.966 | 0.970 | 0.875 | 0.966 |
| | $R_{\text{BERT–Base}}$ (idf) | 0.986 | 0.990 | 0.976 | 0.984 | 0.984 | 0.019 | 0.980 |
| | $F_{\text{BERT–Base}}$ (idf) | **0.974** | 0.993 | 0.980 | 0.978 | 0.978 | 0.853 | 0.976 |
| | $P_{\text{BERT–Base–MRPC}}$ | 0.949 | 0.995 | 0.986 | 0.960 | 0.969 | 0.832 | 0.972 |
| | $R_{\text{BERT–Base–MRPC}}$ | 0.983 | 0.997 | 0.979 | 0.986 | 0.986 | 0.099 | 0.980 |
| | $F_{\text{BERT–Base–MRPC}}$ | 0.967 | 0.997 | 0.984 | 0.978 | 0.981 | 0.722 | 0.979 |
| | $P_{\text{BERT–Base–MRPC}}$ (idf) | 0.949 | 0.994 | 0.986 | 0.946 | 0.969 | 0.743 | 0.969 |
| | $R_{\text{BERT–Base–MRPC}}$ (idf) | 0.984 | 0.994 | 0.980 | 0.980 | 0.986 | 0.541 | 0.982 |
| | $F_{\text{BERT–Base–MRPC}}$ (idf) | 0.967 | 0.995 | 0.984 | 0.968 | 0.979 | 0.464 | 0.978 |
| | $P_{\text{BERT–Large}}$ | 0.969 | 0.991 | 0.985 | **0.979** | 0.970 | 0.915 | 0.969 |
| | $R_{\text{BERT–Large}}$ | 0.990 | 0.993 | 0.980 | 0.988 | 0.988 | 0.745 | 0.978 |
| | $F_{\text{BERT–Large}}$ | 0.982 | 0.993 | 0.984 | 0.986 | 0.981 | 0.909 | 0.976 |
| | $P_{\text{BERT–Large}}$ (idf) | 0.970 | 0.991 | 0.984 | 0.963 | 0.971 | 0.858 | 0.970 |
| | $R_{\text{BERT–Large}}$ (idf) | 0.989 | 0.990 | 0.982 | 0.982 | 0.985 | 0.047 | 0.982 |
| | $F_{\text{BERT–Large}}$ (idf) | 0.981 | 0.991 | 0.984 | 0.976 | 0.978 | 0.722 | 0.978 |
| | $P_{\text{RoBERTa–Base}}$ | 0.959 | 0.992 | 0.988 | 0.986 | 0.971 | 0.809 | 0.976 |
| | $R_{\text{RoBERTa–Base}}$ | 0.987 | 0.997 | 0.978 | 0.989 | 0.988 | 0.238 | 0.981 |
| | $F_{\text{RoBERTa–Base}}$ | 0.973 | 0.997 | 0.987 | 0.989 | 0.982 | 0.674 | 0.982 |
| | $P_{\text{RoBERTa–Base}}$ (idf) | 0.963 | 0.994 | 0.988 | 0.989 | 0.970 | 0.711 | 0.972 |
| | $R_{\text{RoBERTa–Base}}$ (idf) | 0.988 | 0.996 | 0.979 | 0.989 | 0.987 | 0.353 | **0.983** |
| | $F_{\text{RoBERTa–Base}}$ (idf) | 0.976 | 0.997 | 0.986 | **0.990** | 0.980 | 0.277 | 0.980 |
| | $P_{\text{RoBERTa–Large}}$ | 0.965 | 0.995 | 0.990 | 0.976 | 0.976 | 0.846 | 0.975 |
| | $R_{\text{RoBERTa–Large}}$ | **0.989** | 0.997 | 0.982 | 0.989 | 0.988 | 0.540 | 0.981 |
| | $F_{\text{RoBERTa–Large}}$ | 0.978 | 0.998 | 0.989 | 0.983 | 0.985 | 0.760 | 0.981 |
| | $P_{\text{RoBERTa–Large}}$ (idf) | 0.972 | 0.997 | 0.988 | 0.986 | 0.976 | 0.686 | 0.973 |
| | $R_{\text{RoBERTa–Large}}$ (idf) | 0.990 | 0.996 | 0.983 | 0.989 | 0.989 | 0.096 | 0.982 |
| | $F_{\text{RoBERTa–Large}}$ (idf) | 0.982 | **0.998** | 0.988 | 0.989 | 0.983 | 0.453 | 0.980 |
| | $P_{\text{RoBERTa–Large–MNLI}}$ | 0.978 | 0.997 | 0.991 | 0.984 | 0.980 | 0.914 | 0.977 |
| | $R_{\text{RoBERTa–Large–MNLI}}$ | 0.991 | 0.996 | 0.984 | 0.989 | 0.987 | 0.566 | 0.982 |
| | $F_{\text{RoBERTa–Large–MNLI}}$ | 0.987 | 0.998 | 0.989 | 0.988 | 0.986 | 0.873 | 0.982 |
| | $P_{\text{RoBERTa–Large–MNLI}}$ (idf) | 0.982 | 0.998 | **0.992** | 0.990 | 0.978 | 0.822 | 0.974 |
| | $R_{\text{RoBERTa–Large–MNLI}}$ (idf) | **0.992** | 0.996 | 0.985 | 0.988 | 0.988 | 0.022 | 0.983 |
| | $F_{\text{RoBERTa–Large–MNLI}}$ (idf) | **0.989** | 0.998 | 0.990 | 0.990 | 0.985 | 0.583 | 0.980 |
| | $P_{\text{XLNET–Base}}$ | 0.960 | 0.997 | 0.984 | 0.982 | 0.972 | 0.849 | **0.974** |
| | $R_{\text{XLNET–Base}}$ | 0.985 | 0.997 | 0.975 | 0.988 | 0.988 | 0.303 | 0.980 |
| | $F_{\text{XLNET–Base}}$ | 0.974 | 0.998 | 0.981 | 0.986 | 0.982 | 0.628 | 0.980 |
| | $P_{\text{XLNET–Base}}$ (idf) | 0.962 | 0.995 | 0.983 | 0.982 | 0.972 | 0.657 | 0.974 |
| | $R_{\text{XLNET–Base}}$ (idf) | 0.986 | 0.996 | 0.976 | 0.987 | 0.987 | 0.666 | 0.982 |
| | $F_{\text{XLNET–Base}}$ (idf) | 0.975 | 0.996 | 0.980 | 0.985 | 0.981 | **0.259** | 0.980 |
| | $P_{\text{XLNET–Large}}$ | 0.955 | 0.995 | 0.983 | 0.986 | 0.972 | 0.875 | 0.970 |
| | $R_{\text{XLNET–Large}}$ | 0.984 | 0.996 | 0.972 | 0.984 | **0.989** | 0.491 | 0.975 |
| | $F_{\text{XLNET–Large}}$ | 0.971 | 0.996 | 0.980 | 0.987 | 0.984 | 0.821 | 0.976 |
| | $P_{\text{XLNET–Large}}$ (idf) | 0.961 | 0.997 | 0.983 | 0.987 | 0.973 | 0.816 | 0.973 |
| | $R_{\text{XLNET–Large}}$ (idf) | **0.987** | 0.996 | 0.975 | 0.989 | 0.988 | 0.320 | 0.981 |
| | $F_{\text{XLNET–Large}}$ (idf) | 0.976 | 0.997 | 0.980 | 0.989 | 0.982 | 0.623 | 0.980 |
| | $P_{\text{XLM–En}}$ | 0.953 | 0.995 | 0.988 | 0.979 | 0.974 | 0.918 | 0.972 |
| | $R_{\text{XLM–En}}$ | 0.983 | 0.996 | 0.980 | 0.988 | **0.991** | 0.561 | 0.977 |
| | $F_{\text{XLM–En}}$ | 0.969 | 0.997 | 0.986 | 0.986 | 0.985 | 0.869 | 0.977 |
| | $P_{\text{XLM–En}}$ (idf) | 0.957 | 0.996 | 0.987 | 0.970 | 0.974 | 0.862 | 0.973 |
| | $R_{\text{XLM–En}}$ (idf) | 0.982 | 0.995 | 0.981 | 0.988 | 0.989 | 0.213 | 0.980 |
| | $F_{\text{XLM–En}}$ (idf) | 0.970 | 0.996 | 0.985 | 0.982 | 0.982 | 0.519 | 0.978 |

Table 21: Absolute Pearson correlations with human judgments on WMT18 to-English language pairs for 10K hybrid systems. Correlations of metrics not significantly outperformed by any other for that language pair are highlighted in bold. For each language pair, we specify the number of systems.

| Setting | Metric | en-cs 10K | en-de 10K | en-et 10K | en-fi 10K | en-ru 10K | en-tr 10K | en-zh 10K |
|---|---|---|---|---|---|---|---|---|
| Unsupervised | BLEU | 0.993 | 0.977 | 0.971 | 0.958 | 0.977 | 0.796 | 0.941 |
| | CDER | 0.995 | 0.984 | 0.981 | 0.961 | 0.982 | 0.832 | 0.956 |
| | CHARACTER | 0.990 | 0.986 | 0.950 | 0.963 | 0.981 | 0.775 | 0.978 |
| | ITER | 0.865 | 0.978 | 0.982 | 0.966 | 0.965 | 0.872 | – |
| | METEOR++ | – | – | – | – | – | – | – |
| | NIST | **0.997** | 0.984 | 0.980 | 0.944 | 0.988 | 0.870 | 0.944 |
| | PER | **0.987** | 0.979 | 0.954 | 0.904 | 0.986 | 0.829 | 0.950 |
| | TER | 0.995 | **0.986** | 0.977 | 0.939 | 0.985 | 0.837 | 0.959 |
| | UHH_TSKM | – | – | – | – | – | – | – |
| | WER | 0.994 | 0.984 | 0.977 | 0.942 | 0.983 | 0.824 | 0.954 |
| | YISI-0 | 0.971 | 0.983 | 0.965 | 0.942 | 0.988 | **0.953** | 0.951 |
| | YISI-1 | 0.985 | 0.983 | 0.976 | 0.938 | 0.989 | 0.942 | 0.957 |
| | YISI-1 SRL | – | 0.988 | – | – | – | – | 0.948 |
| Supervised | BEER | 0.990 | 0.989 | 0.978 | 0.959 | 0.986 | 0.933 | 0.925 |
| | BLEND | – | – | – | – | 0.986 | – | – |
| | RUSE | – | – | – | – | – | – | – |
| Pre-Trained | $P_{\text{BERT–Multi}}$ | 0.989 | 0.983 | **0.970** | 0.951 | 0.988 | 0.936 | 0.950 |
| | $R_{\text{BERT–Multi}}$ | 0.995 | **0.991** | 0.979 | 0.977 | 0.989 | **0.872** | 0.980 |
| | $F_{\text{BERT–Multi}}$ | **0.993** | 0.988 | 0.978 | 0.969 | 0.989 | 0.910 | 0.969 |
| | $P_{\text{BERT–Multi}}$ (idf) | 0.992 | 0.986 | 0.978 | 0.954 | 0.988 | 0.903 | 0.950 |
| | $R_{\text{BERT–Multi}}$ (idf) | 0.995 | 0.988 | 0.977 | 0.976 | 0.987 | 0.850 | 0.972 |
| | $F_{\text{BERT–Multi}}$ (idf) | 0.995 | 0.988 | 0.979 | 0.969 | 0.987 | 0.877 | 0.963 |
| | $P_{\text{XLM–100}}$ | 0.980 | 0.990 | **0.991** | 0.972 | 0.991 | 0.936 | **0.959** |
| | $R_{\text{XLM–100}}$ | 0.991 | 0.990 | 0.989 | 0.985 | 0.991 | 0.882 | **0.981** |
| | $F_{\text{XLM–100}}$ | 0.987 | 0.990 | **0.991** | 0.981 | 0.991 | 0.915 | 0.974 |
| | $P_{\text{XLM–100}}$ (idf) | 0.982 | 0.990 | 0.990 | **0.968** | **0.991** | 0.931 | 0.960 |
| | $R_{\text{XLM–100}}$ (idf) | 0.989 | 0.990 | 0.990 | **0.985** | 0.990 | 0.867 | 0.978 |
| | $F_{\text{XLM–100}}$ (idf) | 0.986 | **0.991** | 0.991 | 0.982 | 0.991 | 0.905 | **0.972** |

Table 22: Absolute Pearson correlations with human judgments on WMT18 from-English language pairs for 10K hybrid systems. Correlations of metrics not significantly outperformed by any other for that language pair are highlighted in bold. For each language pair, we specify the number of systems.

| Setting | Metric | cs-en | de-en | et-en | fi-en | ru-en | tr-en | zh-en |
|---|---|---|---|---|---|---|---|---|
| Unsupervised | BLEU | 0.135 | 0.804 | 0.757 | 0.460 | 0.230 | 0.096 | 0.661 |
| | CDER | 0.162 | 0.795 | 0.764 | 0.493 | 0.234 | 0.087 | 0.660 |
| | CHARACTER | 0.146 | 0.737 | 0.696 | 0.496 | 0.201 | 0.082 | 0.584 |
| | ITER | 0.152 | 0.814 | 0.746 | 0.474 | 0.234 | 0.100 | 0.673 |
| | METEOR++ | 0.172 | 0.804 | 0.646 | 0.456 | 0.253 | 0.052 | 0.597 |
| | NIST | 0.136 | 0.802 | 0.739 | 0.469 | 0.228 | 0.135 | 0.665 |
| | PER | 0.121 | 0.764 | 0.602 | 0.455 | 0.218 | 0.000 | 0.602 |
| | TER | 0.139 | 0.789 | 0.768 | 0.470 | 0.232 | 0.001 | 0.652 |
| | UHH_TSKM | 0.191 | 0.803 | 0.768 | 0.469 | 0.240 | 0.002 | 0.642 |
| | WER | 0.149 | 0.776 | 0.760 | 0.471 | 0.227 | 0.000 | 0.654 |
| | YISI-0 | 0.148 | 0.780 | 0.703 | 0.483 | 0.229 | 0.106 | 0.629 |
| | YISI-1 | 0.157 | 0.808 | 0.752 | 0.466 | 0.250 | 0.110 | 0.613 |
| | YISI-1 SRL | 0.159 | 0.814 | 0.763 | 0.484 | 0.243 | 0.008 | 0.620 |
| Supervised | BEER | 0.165 | 0.811 | 0.765 | 0.485 | 0.237 | 0.030 | 0.675 |
| | BLEND | 0.184 | 0.820 | 0.779 | 0.484 | 0.254 | 0.003 | 0.611 |
| | RUSE | **0.213** | 0.823 | **0.788** | 0.487 | 0.250 | 0.109 | 0.672 |
| Pre-Trained | $P_{\text{BERT–Base}}$ | 0.190 | 0.815 | 0.778 | 0.468 | 0.261 | 0.130 | 0.655 |
| | $R_{\text{BERT–Base}}$ | 0.189 | 0.813 | 0.775 | 0.481 | 0.266 | 0.014 | 0.663 |
| | $F_{\text{BERT–Base}}$ | 0.194 | 0.819 | 0.778 | 0.474 | 0.265 | 0.144 | 0.670 |
| | $P_{\text{BERT–Base}}$ (idf) | 0.189 | 0.817 | 0.775 | 0.477 | 0.255 | 0.131 | 0.650 |
| | $R_{\text{BERT–Base}}$ (idf) | 0.192 | 0.808 | 0.771 | 0.484 | 0.248 | 0.005 | 0.674 |
| | $F_{\text{BERT–Base}}$ (idf) | 0.193 | 0.817 | 0.774 | 0.483 | 0.262 | 0.081 | 0.669 |
| | $P_{\text{BERT–Base–MRPC}}$ | 0.190 | 0.701 | 0.766 | 0.487 | 0.254 | 0.126 | 0.653 |
| | $R_{\text{BERT–Base–MRPC}}$ | 0.199 | 0.826 | 0.765 | 0.493 | 0.258 | 0.000 | 0.671 |
| | $F_{\text{BERT–Base–MRPC}}$ | 0.197 | 0.824 | 0.767 | 0.491 | 0.260 | 0.147 | 0.668 |
| | $P_{\text{BERT–Base–MRPC}}$ (idf) | 0.186 | 0.806 | 0.765 | 0.492 | 0.247 | 0.125 | 0.661 |
| | $R_{\text{BERT–Base–MRPC}}$ (idf) | 0.200 | 0.823 | 0.760 | 0.495 | 0.258 | 0.000 | 0.680 |
| | $F_{\text{BERT–Base–MRPC}}$ (idf) | 0.196 | 0.821 | 0.763 | 0.497 | 0.254 | 0.031 | 0.676 |
| | $P_{\text{BERT–Large}}$ | 0.200 | 0.815 | 0.778 | 0.474 | 0.261 | 0.137 | 0.661 |
| | $R_{\text{BERT–Large}}$ | 0.194 | 0.809 | 0.779 | 0.493 | **0.270** | 0.006 | 0.672 |
| | $F_{\text{BERT–Large}}$ | 0.199 | 0.810 | 0.782 | 0.484 | 0.266 | 0.142 | 0.672 |
| | $P_{\text{BERT–Large}}$ (idf) | 0.200 | 0.813 | 0.772 | 0.485 | 0.256 | 0.136 | 0.657 |
| | $R_{\text{BERT–Large}}$ (idf) | 0.197 | 0.806 | 0.769 | 0.495 | 0.262 | 0.005 | 0.675 |
| | $F_{\text{BERT–Large}}$ (idf) | 0.199 | 0.811 | 0.772 | 0.494 | 0.262 | 0.006 | 0.673 |
| | $P_{\text{RoBERTa–Base}}$ | 0.173 | 0.675 | 0.757 | 0.502 | 0.258 | 0.126 | 0.654 |
| | $R_{\text{RoBERTa–Base}}$ | 0.165 | 0.816 | 0.764 | 0.483 | 0.266 | 0.000 | 0.674 |
| | $F_{\text{RoBERTa–Base}}$ | 0.173 | 0.820 | 0.764 | 0.498 | 0.262 | 0.090 | 0.669 |
| | $P_{\text{RoBERTa–Base}}$ (idf) | 0.172 | 0.691 | 0.755 | 0.503 | 0.252 | 0.123 | 0.661 |
| | $R_{\text{RoBERTa–Base}}$ (idf) | 0.172 | 0.809 | 0.758 | 0.490 | 0.268 | 0.000 | 0.678 |
| | $F_{\text{RoBERTa–Base}}$ (idf) | 0.178 | 0.820 | 0.758 | 0.501 | 0.260 | 0.001 | 0.674 |
| | $P_{\text{RoBERTa–Large}}$ | 0.174 | 0.704 | 0.765 | 0.497 | 0.255 | 0.140 | 0.663 |
| | $R_{\text{RoBERTa–Large}}$ | 0.163 | 0.805 | 0.770 | 0.491 | 0.263 | 0.005 | 0.679 |
| | $F_{\text{RoBERTa–Large}}$ | 0.175 | 0.825 | 0.770 | 0.499 | 0.262 | 0.143 | 0.675 |
| | $P_{\text{RoBERTa–Large}}$ (idf) | 0.181 | 0.821 | 0.758 | 0.500 | 0.256 | 0.089 | 0.669 |
| | $R_{\text{RoBERTa–Large}}$ (idf) | 0.165 | 0.787 | 0.763 | 0.495 | 0.270 | 0.000 | **0.684** |
| | $F_{\text{RoBERTa–Large}}$ (idf) | 0.179 | 0.824 | 0.761 | 0.502 | 0.265 | 0.004 | 0.679 |
| | $P_{\text{RoBERTa–Large–MNLI}}$ | 0.185 | **0.828** | 0.780 | 0.504 | 0.263 | 0.133 | 0.654 |
| | $R_{\text{RoBERTa–Large–MNLI}}$ | 0.179 | 0.779 | 0.775 | 0.494 | 0.266 | 0.004 | 0.670 |
| | $F_{\text{RoBERTa–Large–MNLI}}$ | 0.186 | 0.827 | 0.778 | 0.502 | 0.267 | 0.113 | 0.669 |
| | $P_{\text{RoBERTa–Large–MNLI}}$ (idf) | 0.190 | 0.820 | 0.771 | 0.504 | 0.261 | 0.102 | 0.661 |
| | $R_{\text{RoBERTa–Large–MNLI}}$ (idf) | 0.181 | 0.769 | 0.766 | 0.494 | 0.266 | 0.004 | 0.674 |
| | $F_{\text{RoBERTa–Large–MNLI}}$ (idf) | 0.188 | 0.822 | 0.768 | 0.501 | 0.265 | 0.004 | 0.671 |
| | $P_{\text{XLNET–Base}}$ | 0.186 | 0.771 | 0.762 | 0.496 | 0.247 | 0.153 | 0.658 |
| | $R_{\text{XLNET–Base}}$ | 0.182 | 0.823 | 0.764 | 0.496 | 0.256 | 0.000 | 0.671 |
| | $F_{\text{XLNET–Base}}$ | 0.186 | 0.824 | 0.765 | 0.499 | 0.253 | 0.049 | 0.673 |
| | $P_{\text{XLNET–Base}}$ (idf) | 0.178 | 0.819 | 0.756 | **0.506** | 0.241 | 0.130 | 0.656 |
| | $R_{\text{XLNET–Base}}$ (idf) | 0.183 | 0.817 | 0.754 | 0.501 | 0.256 | 0.000 | 0.673 |
| | $F_{\text{XLNET–Base}}$ (idf) | 0.182 | 0.821 | 0.755 | 0.505 | 0.250 | 0.000 | 0.670 |
| | $P_{\text{XLNET–Large}}$ | 0.195 | 0.721 | 0.767 | 0.493 | 0.152 | 0.144 | 0.661 |
| | $R_{\text{XLNET–Large}}$ | 0.192 | 0.821 | 0.766 | 0.494 | 0.260 | 0.001 | 0.659 |
| | $F_{\text{XLNET–Large}}$ | 0.196 | 0.824 | 0.773 | 0.496 | 0.261 | **0.155** | 0.675 |
| | $P_{\text{XLNET–Large}}$ (idf) | 0.191 | 0.811 | 0.765 | 0.500 | 0.167 | 0.144 | 0.657 |
| | $R_{\text{XLNET–Large}}$ (idf) | 0.196 | 0.815 | 0.762 | 0.495 | 0.259 | 0.000 | 0.673 |
| | $F_{\text{XLNET–Large}}$ (idf) | 0.195 | 0.822 | 0.764 | 0.499 | 0.256 | 0.046 | 0.674 |
| | $P_{\text{XLM–En}}$ | 0.192 | 0.796 | 0.779 | 0.486 | 0.255 | 0.131 | 0.665 |
| | $R_{\text{XLM–En}}$ | 0.202 | 0.818 | 0.772 | 0.495 | 0.261 | 0.005 | 0.662 |
| | $F_{\text{XLM–En}}$ | 0.199 | 0.827 | 0.778 | 0.491 | 0.262 | 0.086 | 0.674 |
| | $P_{\text{XLM–En}}$ (idf) | 0.189 | 0.818 | 0.770 | 0.485 | 0.259 | 0.116 | 0.662 |
| | $R_{\text{XLM–En}}$ (idf) | 0.202 | 0.812 | 0.761 | 0.490 | 0.250 | 0.003 | 0.668 |
| | $F_{\text{XLM–En}}$ (idf) | 0.196 | 0.821 | 0.766 | 0.490 | 0.263 | 0.003 | 0.672 |

Table 23: Model selection accuracies (Hits@1) on to-English WMT18 hybrid systems. We report the average of 100K samples and the 0.95 confidence intervals are below $10^{-3}$. We bold the highest numbers for each language pair and direction.

| Setting | Metric | cs-en | de-en | et-en | fi-en | ru-en | tr-en | zh-en |
|---|---|---|---|---|---|---|---|---|
| Unsupervised | BLEU | 0.338 | 0.894 | 0.866 | 0.666 | 0.447 | 0.265 | 0.799 |
| | CDER | 0.362 | 0.890 | 0.870 | 0.689 | 0.451 | 0.256 | 0.799 |
| | CHARACTER | 0.349 | 0.854 | 0.814 | 0.690 | 0.429 | 0.254 | 0.739 |
| | ITER | 0.356 | 0.901 | 0.856 | 0.676 | 0.454 | 0.278 | 0.811 |
| | METEOR++ | 0.369 | 0.895 | 0.798 | 0.662 | 0.470 | 0.174 | 0.757 |
| | NIST | 0.338 | 0.894 | 0.857 | 0.672 | 0.446 | 0.323 | 0.803 |
| | PER | 0.325 | 0.866 | 0.771 | 0.663 | 0.435 | 0.021 | 0.754 |
| | TER | 0.342 | 0.885 | 0.873 | 0.673 | 0.447 | 0.063 | 0.792 |
| | UHH_TSKM | 0.387 | 0.894 | 0.873 | 0.671 | 0.460 | 0.063 | 0.788 |
| | WER | 0.353 | 0.876 | 0.868 | 0.674 | 0.443 | 0.034 | 0.790 |
| | YiSi-0 | 0.344 | 0.881 | 0.834 | 0.681 | 0.452 | 0.275 | 0.776 |
| | YiSi-1 | 0.352 | 0.896 | 0.864 | 0.671 | 0.470 | 0.285 | 0.765 |
| | YiSi-1 SRL | 0.351 | 0.901 | 0.871 | 0.682 | 0.464 | 0.086 | 0.770 |
| Supervised | BEER | 0.364 | 0.899 | 0.871 | 0.684 | 0.460 | 0.125 | 0.811 |
| | BLEND | 0.382 | 0.904 | 0.880 | 0.681 | 0.473 | 0.077 | 0.767 |
| | RUSE | **0.417** | 0.906 | **0.885** | 0.686 | 0.468 | 0.273 | 0.809 |
| Pre-Trained | $P_{\text{BERT–Base}}$ | 0.386 | 0.901 | 0.880 | 0.674 | 0.481 | 0.318 | 0.793 |
| | $R_{\text{BERT–Base}}$ | 0.383 | 0.899 | 0.877 | 0.683 | 0.486 | 0.100 | 0.804 |
| | $F_{\text{BERT–Base}}$ | 0.388 | 0.903 | 0.879 | 0.678 | 0.484 | 0.331 | 0.808 |
| | $P_{\text{BERT–Base}}$ (idf) | 0.390 | 0.902 | 0.877 | 0.681 | 0.475 | 0.318 | 0.786 |
| | $R_{\text{BERT–Base}}$ (idf) | 0.390 | 0.896 | 0.874 | 0.686 | 0.475 | 0.077 | 0.811 |
| | $F_{\text{BERT–Base}}$ (idf) | 0.393 | 0.902 | 0.876 | 0.685 | 0.483 | 0.225 | 0.806 |
| | $P_{\text{BERT–Base–MRPC}}$ | 0.392 | 0.832 | 0.872 | 0.686 | 0.475 | 0.319 | 0.791 |
| | $R_{\text{BERT–Base–MRPC}}$ | 0.397 | 0.908 | 0.870 | 0.691 | 0.478 | 0.025 | 0.811 |
| | $F_{\text{BERT–Base–MRPC}}$ | 0.398 | 0.907 | 0.872 | 0.690 | 0.481 | 0.335 | 0.806 |
| | $P_{\text{BERT–Base–MRPC}}$ (idf) | 0.392 | 0.896 | 0.870 | 0.689 | 0.467 | 0.316 | 0.797 |
| | $R_{\text{BERT–Base–MRPC}}$ (idf) | 0.400 | 0.906 | 0.867 | 0.691 | 0.479 | 0.018 | 0.817 |
| | $F_{\text{BERT–Base–MRPC}}$ (idf) | 0.400 | 0.905 | 0.869 | 0.693 | 0.475 | 0.097 | 0.812 |
| | $P_{\text{BERT–Large}}$ | 0.398 | 0.901 | 0.880 | 0.678 | 0.481 | 0.327 | 0.799 |
| | $R_{\text{BERT–Large}}$ | 0.391 | 0.897 | 0.879 | 0.690 | **0.490** | 0.085 | 0.810 |
| | $F_{\text{BERT–Large}}$ | 0.397 | 0.898 | 0.882 | 0.684 | 0.486 | 0.328 | 0.810 |
| | $P_{\text{BERT–Large}}$ (idf) | 0.398 | 0.900 | 0.875 | 0.685 | 0.475 | 0.323 | 0.794 |
| | $R_{\text{BERT–Large}}$ (idf) | 0.395 | 0.895 | 0.873 | 0.692 | 0.488 | 0.080 | 0.813 |
| | $F_{\text{BERT–Large}}$ (idf) | 0.398 | 0.899 | 0.875 | 0.691 | 0.482 | 0.086 | 0.810 |
| | $P_{\text{RoBERTa–Base}}$ | 0.372 | 0.814 | 0.866 | 0.697 | 0.475 | 0.313 | 0.795 |
| | $R_{\text{RoBERTa–Base}}$ | 0.366 | 0.902 | 0.870 | 0.683 | 0.483 | 0.026 | 0.813 |
| | $F_{\text{RoBERTa–Base}}$ | 0.374 | 0.904 | 0.870 | 0.694 | 0.480 | 0.224 | 0.808 |
| | $P_{\text{RoBERTa–Base}}$ (idf) | 0.373 | 0.825 | 0.865 | 0.697 | 0.470 | 0.303 | 0.802 |
| | $R_{\text{RoBERTa–Base}}$ (idf) | 0.374 | 0.898 | 0.866 | 0.688 | 0.486 | 0.028 | 0.816 |
| | $F_{\text{RoBERTa–Base}}$ (idf) | 0.380 | 0.904 | 0.866 | 0.696 | 0.479 | 0.037 | 0.812 |
| | $P_{\text{RoBERTa–Large}}$ | 0.375 | 0.833 | 0.871 | 0.693 | 0.474 | 0.327 | 0.800 |
| | $R_{\text{RoBERTa–Large}}$ | 0.366 | 0.895 | 0.874 | 0.689 | 0.480 | 0.039 | 0.816 |
| | $F_{\text{RoBERTa–Large}}$ | 0.378 | 0.907 | 0.874 | 0.694 | 0.480 | 0.324 | 0.811 |
| | $P_{\text{RoBERTa–Large}}$ (idf) | 0.384 | 0.905 | 0.866 | 0.694 | 0.475 | 0.220 | 0.806 |
| | $R_{\text{RoBERTa–Large}}$ (idf) | 0.368 | 0.885 | 0.869 | 0.692 | 0.487 | 0.030 | **0.819** |
| | $F_{\text{RoBERTa–Large}}$ (idf) | 0.382 | 0.907 | 0.868 | 0.696 | 0.484 | 0.048 | 0.815 |
| | $P_{\text{RoBERTa–Large–MNLI}}$ | 0.383 | **0.909** | 0.880 | 0.698 | 0.480 | 0.323 | 0.795 |
| | $R_{\text{RoBERTa–Large–MNLI}}$ | 0.378 | 0.880 | 0.877 | 0.692 | 0.481 | 0.078 | 0.811 |
| | $F_{\text{RoBERTa–Large–MNLI}}$ | 0.385 | 0.909 | 0.879 | 0.697 | 0.484 | 0.286 | 0.809 |
| | $P_{\text{RoBERTa–Large–MNLI}}$ (idf) | 0.389 | 0.905 | 0.874 | 0.698 | 0.478 | 0.268 | 0.803 |
| | $R_{\text{RoBERTa–Large–MNLI}}$ (idf) | 0.380 | 0.874 | 0.870 | 0.691 | 0.483 | 0.079 | 0.814 |
| | $F_{\text{RoBERTa–Large–MNLI}}$ (idf) | 0.387 | 0.906 | 0.872 | 0.696 | 0.482 | 0.082 | 0.811 |
| | $P_{\text{XLNET–Base}}$ | 0.385 | 0.875 | 0.869 | 0.692 | 0.469 | 0.342 | 0.796 |
| | $R_{\text{XLNET–Base}}$ | 0.381 | 0.907 | 0.869 | 0.693 | 0.477 | 0.026 | 0.809 |
| | $F_{\text{XLNET–Base}}$ | 0.385 | 0.907 | 0.871 | 0.694 | 0.476 | 0.128 | 0.810 |
| | $P_{\text{XLNET–Base}}$ (idf) | 0.381 | 0.904 | 0.864 | **0.699** | 0.464 | 0.289 | 0.794 |
| | $R_{\text{XLNET–Base}}$ (idf) | 0.384 | 0.903 | 0.863 | 0.696 | 0.479 | 0.013 | 0.812 |
| | $F_{\text{XLNET–Base}}$ (idf) | 0.384 | 0.905 | 0.864 | 0.699 | 0.472 | 0.032 | 0.809 |
| | $P_{\text{XLNET–Large}}$ | 0.392 | 0.844 | 0.873 | 0.689 | 0.367 | 0.338 | 0.799 |
| | $R_{\text{XLNET–Large}}$ | 0.389 | 0.905 | 0.871 | 0.690 | 0.482 | 0.031 | 0.800 |
| | $F_{\text{XLNET–Large}}$ | 0.393 | 0.907 | 0.876 | 0.691 | 0.483 | **0.348** | 0.812 |
| | $P_{\text{XLNET–Large}}$ (idf) | 0.393 | 0.899 | 0.870 | 0.694 | 0.387 | 0.333 | 0.794 |
| | $R_{\text{XLNET–Large}}$ (idf) | 0.395 | 0.901 | 0.868 | 0.690 | 0.483 | 0.023 | 0.810 |
| | $F_{\text{XLNET–Large}}$ (idf) | 0.396 | 0.906 | 0.870 | 0.693 | 0.478 | 0.128 | 0.811 |
| | $P_{\text{XLM–En}}$ | 0.394 | 0.891 | 0.880 | 0.685 | 0.476 | 0.322 | 0.802 |
| | $R_{\text{XLM–En}}$ | 0.401 | 0.903 | 0.875 | 0.692 | 0.483 | 0.082 | 0.803 |
| | $F_{\text{XLM–En}}$ | 0.400 | 0.909 | 0.878 | 0.689 | 0.483 | 0.234 | 0.811 |
| | $P_{\text{XLM–En}}$ (idf) | 0.391 | 0.903 | 0.874 | 0.684 | 0.480 | 0.293 | 0.797 |
| | $R_{\text{XLM–En}}$ (idf) | 0.402 | 0.900 | 0.868 | 0.688 | 0.477 | 0.068 | 0.806 |
| | $F_{\text{XLM–En}}$ (idf) | 0.398 | 0.905 | 0.871 | 0.688 | 0.487 | 0.079 | 0.809 |

Table 24: Mean Reciprocal Rank (MRR) of the top metric-rated system on to-English WMT18 hybrid systems. We report the average of 100K samples and the 0.95 confidence intervals are below $10^{-3}$. We bold the highest numbers for each language pair and direction.

| Setting | Metric | cs-en | de-en | et-en | fi-en | ru-en | tr-en | zh-en |
|---------|--------|-------|-------|-------|-------|-------|-------|-------|
| Unsupervised | BLEU | 3.85 | 0.45 | 1.01 | 2.17 | 2.34 | 4.48 | 3.19 |
| | CDER | 3.88 | 0.43 | 0.87 | 1.33 | 2.30 | 4.58 | 3.43 |
| | CHARACTER | 3.77 | 0.49 | 0.94 | 2.07 | 2.25 | 4.07 | 3.37 |
| | ITER | 3.55 | 0.46 | 1.25 | 1.43 | 4.65 | 3.11 | 2.92 |
| | METEOR++ | 3.70 | 0.41 | 0.69 | 1.13 | 2.28 | 1.40 | 3.50 |
| | NIST | 3.93 | 0.49 | 1.10 | 1.19 | 2.36 | 1.42 | 3.92 |
| | PER | 2.02 | 0.46 | 1.71 | 1.49 | 2.25 | 4.22 | 3.20 |
| | TER | 3.86 | 0.43 | 1.14 | 1.14 | 4.34 | 5.18 | 3.82 |
| | UHH_TSKM | 3.98 | 0.40 | 1.27 | 1.10 | 2.23 | 4.26 | 3.47 |
| | WER | 3.85 | 0.44 | 1.48 | 1.18 | 4.87 | 5.96 | 3.72 |
| | YISI-0 | 3.81 | 0.48 | 0.72 | 1.20 | 1.75 | 1.40 | 3.44 |
| | YISI-1 | 3.88 | 0.44 | 0.65 | 1.13 | 2.17 | 1.32 | 3.40 |
| | YISI-1 SRL | 3.67 | 0.41 | 0.64 | 1.20 | 2.15 | 1.31 | 3.55 |
| Supervised | BEER | 3.82 | 0.41 | 0.79 | 1.08 | 1.92 | 1.96 | 3.43 |
| | BLEND | 3.77 | 0.41 | 0.66 | 1.09 | 2.21 | 1.28 | 3.46 |
| | RUSE | 3.13 | **0.32** | 0.64 | 1.03 | 1.51 | 1.94 | 3.15 |
| Pre-Trained | $P_{\text{BERT–Base}}$ | 3.97 | 0.36 | 0.72 | 1.16 | 2.20 | 1.25 | 3.26 |
| | $R_{\text{BERT–Base}}$ | 1.51 | 0.43 | 0.60 | 1.65 | 1.33 | 1.34 | 3.50 |
| | $F_{\text{BERT–Base}}$ | 3.70 | 0.36 | 0.59 | 1.08 | 1.92 | 1.27 | 3.38 |
| | $P_{\text{BERT–Base}}$ (idf) | 3.94 | 0.36 | 0.64 | 1.18 | 2.06 | 2.55 | 3.54 |
| | $R_{\text{BERT–Base}}$ (idf) | 1.54 | 0.43 | 0.63 | 1.87 | 1.12 | 5.96 | 3.38 |
| | $F_{\text{BERT–Base}}$ (idf) | 2.75 | 0.39 | 0.60 | 1.10 | 1.38 | 1.26 | 3.51 |
| | $P_{\text{BERT–Base–MRPC}}$ | 4.02 | 0.35 | 0.74 | 1.15 | 1.09 | 3.33 | 3.06 |
| | $R_{\text{BERT–Base–MRPC}}$ | 2.66 | 0.43 | 0.62 | 1.75 | 1.10 | 5.64 | 3.34 |
| | $F_{\text{BERT–Base–MRPC}}$ | 3.89 | 0.36 | 0.60 | 1.09 | 1.08 | 3.82 | 3.23 |
| | $P_{\text{BERT–Base–MRPC}}$ (idf) | 4.02 | 0.35 | 0.67 | 1.18 | 1.48 | 3.30 | 3.49 |
| | $R_{\text{BERT–Base–MRPC}}$ (idf) | 1.63 | 0.43 | 0.65 | 1.93 | 1.13 | 7.26 | 3.13 |
| | $F_{\text{BERT–Base–MRPC}}$ (idf) | 3.86 | 0.38 | 0.61 | 1.11 | 1.14 | 4.24 | 3.28 |
| | $P_{\text{BERT–Large}}$ | 3.82 | 0.34 | 0.66 | 1.12 | 2.10 | 1.31 | 3.60 |
| | $R_{\text{BERT–Large}}$ | 1.49 | 0.40 | 0.59 | 1.56 | 1.17 | 1.35 | 3.61 |
| | $F_{\text{BERT–Large}}$ | 1.71 | 0.35 | **0.58** | 1.08 | 1.65 | 1.29 | 3.60 |
| | $P_{\text{BERT–Large}}$ (idf) | 3.74 | 0.35 | 0.65 | 1.12 | 1.90 | 1.98 | 3.77 |
| | $R_{\text{BERT–Large}}$ (idf) | 1.51 | 0.42 | 0.62 | 1.86 | 1.10 | 5.84 | 3.21 |
| | $F_{\text{BERT–Large}}$ (idf) | 1.49 | 0.38 | 0.60 | 1.17 | 1.24 | 1.96 | 3.53 |
| | $P_{\text{RoBERTa–Base}}$ | 3.89 | 0.37 | 0.75 | 1.18 | 1.07 | 3.45 | 2.62 |
| | $R_{\text{RoBERTa–Base}}$ | 1.92 | 0.39 | 0.64 | 1.57 | 1.11 | 5.75 | 3.13 |
| | $F_{\text{RoBERTa–Base}}$ | 3.56 | 0.37 | 0.59 | 1.10 | 1.08 | 3.79 | 2.90 |
| | $P_{\text{RoBERTa–Base}}$ (idf) | 3.89 | 0.38 | 0.67 | 1.20 | 1.30 | 3.27 | 3.47 |
| | $R_{\text{RoBERTa–Base}}$ (idf) | 1.61 | 0.42 | 0.67 | 1.65 | 1.14 | 6.55 | 2.95 |
| | $F_{\text{RoBERTa–Base}}$ (idf) | 3.18 | 0.38 | 0.60 | 1.11 | 1.13 | 6.54 | 3.11 |
| | $P_{\text{RoBERTa–Large}}$ | 3.64 | 0.36 | 0.71 | 1.10 | **1.03** | 2.69 | **2.57** |
| | $R_{\text{RoBERTa–Large}}$ | 1.60 | 0.37 | 0.64 | 1.51 | 1.09 | 3.91 | 3.27 |
| | $F_{\text{RoBERTa–Large}}$ | 2.38 | 0.35 | 0.58 | 1.06 | 1.05 | 3.57 | 2.95 |
| | $P_{\text{RoBERTa–Large}}$ (idf) | 2.70 | 0.36 | 0.69 | 1.13 | 1.08 | 3.18 | 2.89 |
| | $R_{\text{RoBERTa–Large}}$ (idf) | 1.55 | 0.39 | 0.66 | 1.59 | 1.10 | 6.66 | 3.18 |
| | $F_{\text{RoBERTa–Large}}$ (idf) | 1.68 | 0.37 | 0.59 | 1.08 | 1.08 | 5.58 | 2.91 |
| | $P_{\text{RoBERTa–Large–MNLI}}$ | 2.14 | 0.35 | 0.61 | 1.07 | 1.09 | **1.21** | 3.35 |
| | $R_{\text{RoBERTa–Large–MNLI}}$ | 1.45 | 0.37 | 0.64 | 1.49 | 1.10 | 4.42 | 3.55 |
| | $F_{\text{RoBERTa–Large–MNLI}}$ | **1.42** | 0.35 | 0.59 | 1.07 | 1.07 | 1.27 | 3.41 |
| | $P_{\text{RoBERTa–Large–MNLI}}$ (idf) | 1.55 | 0.35 | 0.60 | 1.08 | 1.12 | 1.54 | 3.87 |
| | $R_{\text{RoBERTa–Large–MNLI}}$ (idf) | 1.45 | 0.39 | 0.64 | 1.65 | 1.09 | 5.89 | 3.32 |
| | $F_{\text{RoBERTa–Large–MNLI}}$ (idf) | 1.42 | 0.36 | 0.60 | 1.10 | 1.08 | 3.80 | 3.45 |
| | $P_{\text{XLNET–Base}}$ | 3.90 | 0.37 | 0.68 | 1.07 | 1.16 | 2.47 | 2.91 |
| | $R_{\text{XLNET–Base}}$ | 1.71 | 0.45 | 0.72 | 1.58 | 1.07 | 6.29 | 3.36 |
| | $F_{\text{XLNET–Base}}$ | 3.78 | 0.39 | 0.62 | 1.05 | 1.07 | 3.60 | 3.20 |
| | $P_{\text{XLNET–Base}}$ (idf) | 3.90 | 0.46 | 0.65 | 1.08 | 2.93 | 3.30 | 3.39 |
| | $R_{\text{XLNET–Base}}$ (idf) | 1.51 | 0.45 | 0.82 | 1.78 | 1.12 | 10.77 | 3.13 |
| | $F_{\text{XLNET–Base}}$ (idf) | 3.67 | 0.42 | 0.66 | 1.11 | 1.22 | 7.13 | 3.23 |
| | $P_{\text{XLNET–Large}}$ | 3.94 | 0.37 | 0.71 | 1.10 | 21.10 | 1.85 | 2.90 |
| | $R_{\text{XLNET–Large}}$ | 2.23 | 0.41 | 0.69 | 1.34 | 1.07 | 4.46 | 3.40 |
| | $F_{\text{XLNET–Large}}$ | 3.84 | 0.36 | 0.60 | **1.03** | 1.07 | 3.38 | 3.22 |
| | $P_{\text{XLNET–Large}}$ (idf) | 3.92 | 0.41 | 0.64 | 1.12 | 21.10 | 3.24 | 3.37 |
| | $R_{\text{XLNET–Large}}$ (idf) | 1.60 | 0.43 | 0.78 | 1.70 | 1.09 | 6.13 | 3.20 |
| | $F_{\text{XLNET–Large}}$ (idf) | 3.80 | 0.38 | 0.63 | 1.06 | 1.09 | 3.72 | 3.25 |
| | $P_{\text{XLM–En}}$ | 3.88 | 0.33 | 0.75 | 1.16 | 2.16 | 1.28 | 3.29 |
| | $R_{\text{XLM–En}}$ | 1.98 | 0.41 | 0.60 | 1.41 | 1.21 | 3.30 | 3.47 |
| | $F_{\text{XLM–En}}$ | 3.78 | 0.36 | 0.61 | 1.09 | 1.71 | 1.30 | 3.40 |
| | $P_{\text{XLM–En}}$ (idf) | 3.84 | 0.36 | 0.69 | 1.17 | 1.86 | 1.33 | 3.47 |
| | $R_{\text{XLM–En}}$ (idf) | 1.70 | 0.42 | 0.63 | 1.55 | 1.11 | 5.87 | 3.36 |
| | $F_{\text{XLM–En}}$ (idf) | 3.72 | 0.40 | 0.62 | 1.14 | 1.32 | 4.15 | 3.43 |

Table 25: Absolute Difference ($\times 100$) of the top metric-rated and the top human-rated system on to-English WMT18 hybrid systems. Smaller difference signify higher agreement with human scores. We report the average of 100K samples and the 0.95 confidence intervals are below $10^{-3}$. We bold the lowest numbers for each language pair and direction.

| Setting | Metric | en-cs | en-de | en-et | en-fi | en-ru | en-tr | en-zh |
|---|---|---|---|---|---|---|---|---|
| | BLEU | 0.151 | 0.611 | 0.617 | 0.087 | 0.519 | 0.029 | 0.515 |
| | CDER | 0.163 | 0.663 | 0.731 | 0.081 | 0.541 | 0.032 | 0.552 |
| | CHARACTER | 0.135 | **0.737** | 0.639 | **0.492** | 0.543 | 0.027 | **0.667** |
| | ITER | 0.000 | 0.691 | 0.734 | 0.112 | 0.534 | 0.031 | – |
| | METEOR++ | – | – | – | – | – | – | – |
| Unsupervised | NIST | 0.182 | 0.662 | 0.549 | 0.083 | 0.537 | 0.033 | 0.553 |
| | PER | 0.179 | 0.555 | 0.454 | 0.062 | 0.535 | 0.032 | 0.539 |
| | TER | 0.175 | 0.657 | 0.550 | 0.065 | 0.545 | 0.029 | 0.551 |
| | UHH_TSKM | – | – | – | – | – | – | – |
| | WER | 0.155 | 0.643 | 0.552 | 0.067 | 0.538 | 0.029 | 0.546 |
| | YISI-0 | 0.154 | 0.674 | 0.622 | 0.356 | 0.523 | 0.383 | 0.600 |
| | YISI-1 | 0.178 | 0.670 | 0.674 | 0.230 | 0.548 | **0.396** | 0.595 |
| | YISI-1 SRL | – | 0.708 | – | – | – | – | 0.537 |
| | BEER | 0.174 | 0.670 | 0.662 | 0.113 | 0.555 | 0.296 | 0.531 |
| Supervised | BLEND | – | – | – | – | **0.559** | – | – |
| | RUSE | – | – | – | – | – | – | – |
| | $P_{\text{BERT–Multi}}$ | 0.181 | 0.665 | 0.771 | 0.077 | 0.550 | 0.373 | 0.550 |
| | $R_{\text{BERT–Multi}}$ | 0.184 | 0.728 | 0.722 | 0.146 | 0.544 | 0.031 | 0.657 |
| | $F_{\text{BERT–Multi}}$ | 0.185 | 0.703 | 0.764 | 0.081 | 0.548 | 0.032 | 0.629 |
| | $P_{\text{BERT–Multi}}$ (idf) | 0.175 | 0.713 | 0.769 | 0.080 | 0.542 | 0.031 | 0.549 |
| | $R_{\text{BERT–Multi}}$ (idf) | 0.177 | 0.725 | 0.752 | 0.178 | 0.538 | 0.031 | 0.628 |
| | $F_{\text{BERT–Multi}}$ (idf) | 0.178 | 0.721 | 0.766 | 0.081 | 0.543 | 0.030 | 0.594 |
| Pre-Trained | $P_{\text{XLM–100}}$ | 0.175 | 0.669 | 0.748 | 0.079 | 0.550 | 0.314 | 0.582 |
| | $R_{\text{XLM–100}}$ | **0.195** | 0.671 | 0.770 | 0.222 | 0.555 | 0.034 | 0.658 |
| | $F_{\text{XLM–100}}$ | 0.187 | 0.670 | **0.775** | 0.099 | 0.552 | 0.034 | 0.615 |
| | $P_{\text{XLM–100}}$ (idf) | 0.163 | 0.664 | 0.750 | 0.091 | 0.550 | 0.288 | 0.578 |
| | $R_{\text{XLM–100}}$ (idf) | 0.191 | 0.681 | 0.770 | 0.231 | 0.548 | 0.033 | 0.645 |
| | $F_{\text{XLM–100}}$ (idf) | 0.180 | 0.672 | 0.774 | 0.127 | 0.550 | 0.033 | 0.616 |

Table 26: Model selection accuracies (Hits@1) on to-English WMT18 hybrid systems. We report the average of 100K samples and the 0.95 confidence intervals are below $10^{-3}$. We bold the highest numbers for each language pair and direction.

| Setting | Metric | en-cs | en-de | en-et | en-fi | en-ru | en-tr | en-zh |
|---------|--------|-------|-------|-------|-------|-------|-------|-------|
| Unsupervised | BLEU | 0.363 | 0.764 | 0.766 | 0.323 | 0.714 | 0.205 | 0.666 |
| | CDER | 0.371 | 0.803 | 0.851 | 0.319 | 0.729 | 0.210 | 0.700 |
| | CHARACTER | 0.346 | **0.853** | 0.781 | **0.667** | 0.732 | 0.205 | **0.809** |
| | ITER | 0.044 | 0.825 | 0.853 | 0.365 | 0.717 | 0.210 | – |
| | METEOR++ | – | – | – | – | – | – | – |
| | NIST | 0.393 | 0.803 | 0.710 | 0.326 | 0.726 | 0.211 | 0.698 |
| | PER | 0.387 | 0.719 | 0.624 | 0.301 | 0.725 | 0.211 | 0.678 |
| | TER | 0.384 | 0.798 | 0.708 | 0.305 | 0.733 | 0.209 | 0.695 |
| | UHH_TSKM | – | – | – | – | – | – | – |
| | WER | 0.367 | 0.787 | 0.710 | 0.308 | 0.728 | 0.209 | 0.696 |
| | YISI-0 | 0.370 | 0.811 | 0.775 | 0.553 | 0.715 | 0.602 | 0.753 |
| | YISI-1 | 0.390 | 0.808 | 0.811 | 0.439 | 0.735 | **0.612** | 0.750 |
| | YISI-1 SRL | – | 0.835 | – | – | – | – | 0.691 |
| Supervised | BEER | 0.388 | 0.808 | 0.804 | 0.353 | 0.739 | 0.507 | 0.683 |
| | BLEND | – | – | – | – | **0.742** | – | – |
| | RUSE | – | – | – | – | – | – | – |
| Pre-Trained | $P_{\text{BERT–Multi}}$ | 0.395 | 0.805 | 0.876 | 0.314 | 0.736 | 0.586 | 0.694 |
| | $R_{\text{BERT–Multi}}$ | 0.401 | 0.849 | 0.844 | 0.368 | 0.732 | 0.212 | 0.802 |
| | $F_{\text{BERT–Multi}}$ | 0.400 | 0.832 | 0.872 | 0.317 | 0.735 | 0.214 | 0.775 |
| | $P_{\text{BERT–Multi}}$ (idf) | 0.390 | 0.839 | 0.875 | 0.320 | 0.730 | 0.213 | 0.691 |
| | $R_{\text{BERT–Multi}}$ (idf) | 0.395 | 0.847 | 0.864 | 0.398 | 0.727 | 0.212 | 0.776 |
| | $F_{\text{BERT–Multi}}$ (idf) | 0.395 | 0.844 | 0.873 | 0.319 | 0.730 | 0.212 | 0.739 |
| | $P_{\text{XLM–100}}$ | 0.391 | 0.808 | 0.862 | 0.316 | 0.735 | 0.522 | 0.733 |
| | $R_{\text{XLM–100}}$ | **0.413** | 0.809 | 0.876 | 0.435 | 0.738 | 0.216 | 0.803 |
| | $F_{\text{XLM–100}}$ | 0.404 | 0.809 | **0.878** | 0.333 | 0.737 | 0.216 | 0.767 |
| | $P_{\text{XLM–100}}$ (idf) | 0.377 | 0.805 | 0.863 | 0.326 | 0.735 | 0.497 | 0.729 |
| | $R_{\text{XLM–100}}$ (idf) | 0.409 | 0.816 | 0.876 | 0.444 | 0.733 | 0.214 | 0.793 |
| | $F_{\text{XLM–100}}$ (idf) | 0.396 | 0.810 | 0.878 | 0.355 | 0.735 | 0.214 | 0.767 |

Table 27: Mean Reciprocal Rank (MRR) of the top metric-rated system on to-English WMT18 hybrid systems. We report the average of 100K samples and the 0.95 confidence intervals are below $10^{-3}$. We bold the highest numbers for each language pair and direction.

| Setting | Metric | en-cs | en-de | en-et | en-fi | en-ru | en-tr | en-zh |
|---|---|---|---|---|---|---|---|---|
| Unsupervised | BLEU | 1.26 | 6.36 | 2.59 | 0.92 | 0.76 | 9.40 | 3.01 |
| | CDER | 1.25 | 6.70 | 1.90 | 1.41 | 0.87 | 9.37 | 1.75 |
| | CHARACTER | 1.23 | 6.90 | 2.19 | 4.35 | 0.93 | 5.22 | 1.64 |
| | ITER | 1.25 | 9.14 | 2.52 | 1.52 | 1.35 | 7.33 | – |
| | METEOR++ | – | – | – | – | – | – | – |
| | NIST | 1.24 | 5.28 | 2.55 | 1.02 | 0.75 | 8.82 | 3.34 |
| | PER | 1.25 | 6.62 | 4.92 | 7.43 | 0.68 | 9.76 | 2.31 |
| | TER | 1.21 | 6.02 | 4.34 | 2.17 | 0.73 | 8.80 | 1.43 |
| | UHH_TSKM | – | – | – | – | – | – | – |
| | WER | 1.22 | 6.15 | 4.19 | 2.43 | 0.72 | 9.28 | 1.49 |
| | YISI-0 | 1.25 | 6.62 | 1.53 | 1.46 | 0.75 | 3.47 | 2.87 |
| | YISI-1 | 1.22 | 6.27 | 1.21 | 1.13 | 0.71 | 3.51 | 3.33 |
| | YISI-1 SRL | – | 6.57 | – | – | – | – | 3.71 |
| Supervised | BEER | 1.21 | 5.96 | 1.84 | 0.77 | 0.74 | 3.36 | 1.96 |
| | BLEND | – | – | – | – | 0.71 | – | – |
| | RUSE | – | – | – | – | – | – | – |
| Pre-Trained | $P_{\text{BERT–Multi}}$ | 1.17 | 3.27 | 1.38 | 1.24 | 0.75 | 4.14 | 2.08 |
| | $R_{\text{BERT–Multi}}$ | 1.16 | 6.68 | **0.77** | 0.94 | 0.68 | 3.22 | **1.31** |
| | $F_{\text{BERT–Multi}}$ | 1.15 | 5.17 | 0.90 | 0.98 | 0.71 | 3.26 | 1.62 |
| | $P_{\text{BERT–Multi}}$ (idf) | 1.14 | **3.82** | 1.66 | 1.27 | 0.76 | 4.57 | 2.04 |
| | $R_{\text{BERT–Multi}}$ (idf) | 1.15 | 6.97 | 0.83 | 3.65 | 0.68 | 3.32 | 1.37 |
| | $F_{\text{BERT–Multi}}$ (idf) | **1.14** | 5.63 | 1.13 | 1.19 | 0.71 | 3.38 | 1.58 |
| | $P_{\text{XLM–100}}$ | 1.22 | 6.30 | 1.14 | 0.79 | 0.74 | 3.73 | 2.21 |
| | $R_{\text{XLM–100}}$ | 1.18 | 6.89 | 0.76 | 0.77 | 0.66 | 3.26 | 1.68 |
| | $F_{\text{XLM–100}}$ | 1.19 | 6.44 | 0.82 | 0.76 | 0.69 | **3.21** | 1.57 |
| | $P_{\text{XLM–100}}$ (idf) | 1.21 | 6.61 | 1.07 | 0.78 | 0.72 | 5.59 | 2.02 |
| | $R_{\text{XLM–100}}$ (idf) | 1.19 | 7.07 | 0.77 | 0.77 | **0.66** | 3.33 | 1.60 |
| | $F_{\text{XLM–100}}$ (idf) | 1.20 | 6.57 | 0.86 | **0.76** | 0.68 | 3.28 | 1.56 |

Table 28: Absolute Difference ($\times 100$) of the top metric-rated and the top human-rated system on to-English WMT18 hybrid systems. Smaller difference indicate higher agreement with human scores. We report the average of 100K samples and the 0.95 confidence intervals are below $10^{-3}$. We bold the lowest numbers for each language pair and direction.

| Metric | M1 | M2 |
|---|---|---|
| BLEU-1 | $0.124^*$ | $0.135^*$ |
| BLEU-2 | $0.037^*$ | $0.048^*$ |
| BLEU-3 | $0.004^*$ | $0.016^*$ |
| BLEU-4 | $-0.019^*$ | $-0.005^*$ |
| METEOR | $0.606^*$ | $0.594^*$ |
| ROUGE-L | $0.090^*$ | $0.096^*$ |
| CIDER | $0.438^*$ | $0.440^*$ |
| SPICE | $0.759^*$ | $0.750^*$ |
| LEIC | $\mathbf{0.939}^*$ | $\mathbf{0.949}^*$ |
| BEER | 0.491 | 0.562 |
| EED | 0.545 | 0.599 |
| CHRF++ | 0.702 | 0.729 |
| CHARACTER | 0.800 | 0.801 |
| $P_{\text{BERT-Base}}$ | 0.313 | 0.344 |
| $R_{\text{BERT-Base}}$ | 0.679 | 0.622 |
| $F_{\text{BERT-Base}}$ | 0.531 | 0.519 |
| $P_{\text{BERT-Base}}$ (idf) | 0.243 | 0.286 |
| $R_{\text{BERT-Base}}$ (idf) | 0.834 | 0.783 |
| $F_{\text{BERT-Base}}$ (idf) | 0.579 | 0.581 |
| $P_{\text{BERT-Base-MRPC}}$ | 0.252 | 0.331 |
| $R_{\text{BERT-Base-MRPC}}$ | 0.644 | 0.641 |
| $F_{\text{BERT-Base-MRPC}}$ | 0.470 | 0.512 |
| $P_{\text{BERT-Base-MRPC}}$ (idf) | 0.264 | 0.300 |
| $R_{\text{BERT-Base-MRPC}}$ (idf) | 0.794 | 0.767 |
| $F_{\text{BERT-Base-MRPC}}$ (idf) | 0.575 | 0.583 |
| $P_{\text{BERT-Large}}$ | 0.454 | 0.486 |
| $R_{\text{BERT-Large}}$ | 0.756 | 0.697 |
| $F_{\text{BERT-Large}}$ | 0.649 | 0.634 |
| $P_{\text{BERT-Large}}$ (idf) | 0.327 | 0.372 |
| $R_{\text{BERT-Large}}$ (idf) | 0.873 | 0.821 |
| $F_{\text{BERT-Large}}$ (idf) | 0.645 | 0.647 |
| $P_{\text{RoBERTa-Base}}$ | -0.223 | -0.179 |
| $R_{\text{RoBERTa-Base}}$ | 0.827 | 0.800 |
| $F_{\text{RoBERTa-Base}}$ | 0.176 | 0.191 |
| $P_{\text{RoBERTa-Base}}$ (idf) | -0.256 | -0.267 |
| $R_{\text{RoBERTa-Base}}$ (idf) | 0.901 | 0.869 |
| $F_{\text{RoBERTa-Base}}$ (idf) | 0.188 | 0.157 |
| $P_{\text{RoBERTa-Large}}$ | -0.105 | -0.041 |
| $R_{\text{RoBERTa-Large}}$ | 0.888 | 0.863 |
| $F_{\text{RoBERTa-Large}}$ | 0.322 | 0.350 |
| $P_{\text{RoBERTa-Large}}$ (idf) | 0.063 | -0.011 |
| $R_{\text{RoBERTa-Large}}$ (idf) | $\mathbf{0.917}$ | $\mathbf{0.889}$ |
| $F_{\text{RoBERTa-Large}}$ (idf) | 0.519 | 0.453 |
| $P_{\text{RoBERTa-Large-MNLI}}$ | 0.129 | 0.208 |
| $R_{\text{RoBERTa-Large-MNLI}}$ | 0.820 | 0.823 |
| $F_{\text{RoBERTa-Large-MNLI}}$ | 0.546 | 0.592 |
| $P_{\text{RoBERTa-Large-MNLI}}$ (idf) | 0.081 | 0.099 |
| $R_{\text{RoBERTa-Large-MNLI}}$ (idf) | 0.906 | 0.875 |
| $F_{\text{RoBERTa-Large-MNLI}}$ (idf) | 0.605 | 0.596 |
| $P_{\text{XLNet-Base}}$ | -0.046 | 0.080 |
| $R_{\text{XLNet-Base}}$ | 0.409 | 0.506 |
| $F_{\text{XLNet-Base}}$ | 0.146 | 0.265 |
| $P_{\text{XLNet-Base}}$ (idf) | 0.006 | 0.145 |
| $R_{\text{XLNet-Base}}$ (idf) | 0.655 | 0.720 |
| $F_{\text{XLNet-Base}}$ (idf) | 0.270 | 0.391 |
| $P_{\text{XLNet-Large}}$ | -0.188 | -0.115 |
| $R_{\text{XLNet-Large}}$ | 0.178 | 0.195 |
| $F_{\text{XLNet-Large}}$ | -0.014 | 0.036 |
| $P_{\text{XLNet-Large}}$ (idf) | -0.186 | -0.072 |
| $R_{\text{XLNet-Large}}$ (idf) | 0.554 | 0.555 |
| $F_{\text{XLNet-Large}}$ (idf) | 0.151 | 0.234 |
| $P_{\text{XLM-En}}$ | 0.230 | 0.220 |
| $R_{\text{XLM-En}}$ | 0.333 | 0.263 |
| $F_{\text{XLM-En}}$ | 0.297 | 0.243 |
| $P_{\text{XLM-En}}$ (idf) | 0.266 | 0.275 |
| $R_{\text{XLM-En}}$ (idf) | 0.700 | 0.640 |
| $F_{\text{XLM-En}}$ (idf) | 0.499 | 0.470 |

Table 29: Pearson correlation on the 2015 COCO Captioning Challenge. The M1 and M2 measures are described in Section 4. We bold the best correlating task-specific and task-agnostic metrics in each setting LEIC uses images as additional inputs. Numbers with $^*$ are cited from Cui et al. (2018).

| Type | Method | QQP | PAWS$_{\text{QQP}}$ |
|---|---|---|---|
| Trained on QQP (supervised) | DecAtt | 0.939* | 0.263 |
| | DIIN | 0.952* | 0.324 |
| | BERT | **0.963*** | **0.351** |
| Trained on QQP + PAWS$_{\text{QQP}}$ (supervised) | DecAtt | - | 0.511 |
| | DIIN | - | 0.778 |
| | BERT | - | **0.831** |
| Metric (Not trained on QQP or PAWS$_{\text{QQP}}$) | BLEU-1 | 0.737 | 0.402 |
| | BLEU-2 | 0.720 | 0.548 |
| | BLEU-3 | 0.712 | 0.527 |
| | BLEU-4 | 0.707 | 0.527 |
| | METEOR | 0.755 | 0.532 |
| | ROUGE-L | 0.740 | 0.536 |
| | CHRF++ | 0.577 | 0.608 |
| | BEER | 0.741 | 0.564 |
| | EED | 0.743 | 0.611 |
| | CHARACTER | 0.698 | 0.650 |
| | $P_{\text{BERT–Base}}$ | 0.750 | 0.654 |
| | $R_{\text{BERT–Base}}$ | 0.739 | 0.655 |
| | $F_{\text{BERT–Base}}$ | 0.755 | 0.654 |
| | $P_{\text{BERT–Base}}$ (idf) | 0.766 | 0.665 |
| | $R_{\text{BERT–Base}}$ (idf) | 0.752 | 0.665 |
| | $F_{\text{BERT–Base}}$ (idf) | 0.770 | 0.664 |
| | $P_{\text{BERT–Base–MRPC}}$ | 0.742 | 0.615 |
| | $R_{\text{BERT–Base–MRPC}}$ | 0.729 | 0.617 |
| | $F_{\text{BERT–Base–MRPC}}$ | 0.746 | 0.614 |
| | $P_{\text{BERT–Base–MRPC}}$ (idf) | 0.752 | 0.618 |
| | $R_{\text{BERT–Base–MRPC}}$ (idf) | 0.737 | 0.619 |
| | $F_{\text{BERT–Base–MRPC}}$ (idf) | 0.756 | 0.617 |
| | $P_{\text{BERT–Large}}$ | 0.752 | 0.706 |
| | $R_{\text{BERT–Large}}$ | 0.740 | 0.710 |
| | $F_{\text{BERT–Large}}$ | 0.756 | 0.707 |
| | $P_{\text{BERT–Large}}$ (idf) | 0.766 | 0.713 |
| | $R_{\text{BERT–Large}}$ (idf) | 0.751 | 0.718 |
| | $F_{\text{BERT–Large}}$ (idf) | 0.769 | 0.714 |
| | $P_{\text{RoBERTa–Base}}$ | 0.746 | 0.657 |
| | $R_{\text{RoBERTa–Base}}$ | 0.736 | 0.656 |
| | $F_{\text{RoBERTa–Base}}$ | 0.751 | 0.654 |
| | $P_{\text{RoBERTa–Base}}$ (idf) | 0.760 | 0.666 |
| | $R_{\text{RoBERTa–Base}}$ (idf) | 0.745 | 0.666 |
| | $F_{\text{RoBERTa–Base}}$ (idf) | 0.765 | 0.664 |
| | $P_{\text{RoBERTa–Large}}$ | 0.757 | 0.687 |
| | $R_{\text{RoBERTa–Large}}$ | 0.744 | 0.685 |
| | $F_{\text{RoBERTa–Large}}$ | 0.761 | 0.685 |
| | $P_{\text{RoBERTa–Large}}$ (idf) | 0.773 | 0.691 |
| | $R_{\text{RoBERTa–Large}}$ (idf) | 0.757 | 0.697 |
| | $F_{\text{RoBERTa–Large}}$ (idf) | 0.777 | 0.693 |
| | $P_{\text{RoBERTa–Large–MNLI}}$ | 0.763 | 0.767 |
| | $R_{\text{RoBERTa–Large–MNLI}}$ | 0.750 | 0.772 |
| | $F_{\text{RoBERTa–Large–MNLI}}$ | 0.766 | **0.770** |
| | $P_{\text{RoBERTa–Large–MNLI}}$ (idf) | 0.783 | 0.756 |
| | $R_{\text{RoBERTa–Large–MNLI}}$ (idf) | 0.767 | 0.764 |
| | $F_{\text{RoBERTa–Large–MNLI}}$ (idf) | **0.784** | 0.759 |
| | $P_{\text{XLNet–Base}}$ | 0.737 | 0.603 |
| | $R_{\text{XLNet–Base}}$ | 0.731 | 0.607 |
| | $F_{\text{XLNet–Base}}$ | 0.739 | 0.605 |
| | $P_{\text{XLNet–Base}}$ (idf) | 0.751 | 0.625 |
| | $R_{\text{XLNet–Base}}$ (idf) | 0.743 | 0.630 |
| | $F_{\text{XLNet–Base}}$ (idf) | 0.751 | 0.626 |
| | $P_{\text{XLNet–Large}}$ | 0.742 | 0.593 |
| | $R_{\text{XLNet–Large}}$ | 0.734 | 0.598 |
| | $F_{\text{XLNet–Large}}$ | 0.744 | 0.596 |
| | $P_{\text{XLNet–Large}}$ (idf) | 0.759 | 0.604 |
| | $R_{\text{XLNet–Large}}$ (idf) | 0.749 | 0.610 |
| | $F_{\text{XLNet–Large}}$ (idf) | 0.760 | 0.606 |
| | $P_{\text{XLM–En}}$ | 0.734 | 0.600 |
| | $R_{\text{XLM–En}}$ | 0.725 | 0.604 |
| | $F_{\text{XLM–En}}$ | 0.737 | 0.602 |
| | $P_{\text{XLM–En}}$ (idf) | 0.757 | 0.596 |
| | $R_{\text{XLM–En}}$ (idf) | 0.745 | 0.603 |
| | $F_{\text{XLM–En}}$ (idf) | 0.759 | 0.600 |

Table 30: Area under ROC curve (AUC) on QQP and PAWS$_{\text{QQP}}$ datasets. The scores of trained DecATT (Parikh et al., 2016), DIIN (Gong et al., 2018), and fine-tuned BERT are reported by Zhang et al. (2019). We bold the best task-specific and task-agnostic metrics. Numbers with * are scores on the held-out test set of QQP.

