# OpenReview forum: "BERTScore: Evaluating Text Generation with BERT"
_ICLR.cc/2020/Conference — Accept (Poster)_

### Official Review · AnonReviewer1 · 2019-10-21
**Official Blind Review #1**

**Rating:** 8

**Review:**


*** Update ***
I'd like to thank the authors for answering my questions, and I am satisfied with their response. I have read the other reviews for this paper as well, and I am keeping my score.


This paper proposes BERTScore, a method for automatic evaluation of text. Their method uses BERT to produce contextualized word representations for the words in the reference and hypothesis. Then they compute the precision, recall, and F1 by greedily matching up words between the hypothesis and reference. To be more specific, for say recall they take each word in the reference and compute the cosine with all words in the hypothesis. Then they add up the largest cosine similarity for each word and average them together. Precision is defined similarly but with the roles of hypothesis and reference switched. F1 is then the harmonic mean of these two scores. They also experiment with using idf to weight importance.

Their method is simple, but achieves very strong results and there are a ton of experiments in this paper (it is 41 pages). The focus is largely on metrics for MT, but they also evaluate on image captioning. The paper is also very thorough and many of the questions I had when reading it are answered (like effect of optimal matching, running time, etc.). The latter (running time) being one of the downsides of the method if it was to be used for fine-tuning MT systems. 40 times slower than BLEU, but I think this increased cost would be worth it and could be engineered around.

Overall, I like the paper - it is simple and effective on its goal task of automatic evaluation for text generation. I think we are moving that way as a field and this paper proposes a useful method and is additionally a good study on the subject.

A question I have is why the method doesn't perform well in certain cases. For instance, in Table 2 and 3 - some of the evaluations with tr and fi fall well below relative performance for other language pairs. Does this have to do with the quality of the representations in multilingual BERT? What is YiSi-1 doing, for instance for model selection of en-fi and en-tr that makes it have so much better performance?

Edit: I also wonder if incorporating idf would be better if the values were computed by a larger corpus. I think it would make the most sense to compute these from the training data for the underlying BERT models. Since BERT itself is a function of this training data, it seems appropriate that these values would be as well (or perhaps at least a subset of this data).

Missing citations:
A citation to "Beyond BLEU:Training Neural Machine Translation with Semantic Similarity" from ACL 2019 should be incorporated into the related work. They use semantic similarity to fine-tune NMT systems with their own embedding-based (semantic similarity) metric and they found some nice properties from training in this way. Have you tried BERTScore on sentence similarity tasks? It's possible BERTScore could have strong performance and some readers may wonder this. There are evaluations on PAWS for paraphrase detection which I appreciated, but that is a little different.

A citation to "Deep Reinforcement Learning with Distributional Semantic Rewards for Abstractive Summarization" from EMNLP 2019 should also be incorporated. This paper is a big boon to BERT score showing that it is a very helpful metric for fine-tuning summarization systems. They don't even need a cross-entropy term since BERTScore captures fluency so well. I'd like to see it for MT as well, but perhaps that is the next paper.

Typos:
The word "language" is misspelled twice in Appendix E.

**Experience Assessment:**

I have published in this field for several years.

**Review Assessment: Checking Correctness Of Derivations And Theory:**

N/A

**Review Assessment: Checking Correctness Of Experiments:**

I carefully checked the experiments.

**Review Assessment: Thoroughness In Paper Reading:**

I read the paper thoroughly.

---

> ### Author Response · Authors · 2019-11-07
> **Thank you for your comments!**
>
> Thank you for your comments! We appreciate the very detailed review. We have included the missing citations and fixed the typos in the revised version.
>
> Similar to your hypothesis, we suspect that multilingual BERT cannot produce high-quality representations for Turkish and Finnish. This can lead to worse performance of BERTScore. Based on [1] and [2], YiSi-1 trains word2vec embeddings on the monolingual data provided as part of the WMT translation task, which may explain its comparably higher performance on these languages. We believe it is an important future direction to improve the performance of multilingual BERT on low-resource language, but this requires a broader study of training BERT in low-resource regimes.
>
> We have studied computing the idf on a larger corpus. We computed idf scores using the monolingual English corpus released by WMT18, a much larger amount of data then we used before. The importance-weighted version of BERTScore using these idf scores performs worse than the original importance-weighted version. We hypothesize this is due to the domain shift between the corpora.
>
> Beyond paraphrase detection, we didn’t try using BERTScore for text similarity tasks. The results on paraphrase are definitely promising. Given the number of experiments we conducted, we decided to consider this an important direction for future work. Indeed, several groups are already following the direction of using BERTScore for other tasks, including [3] and one of the papers R1 points to ("Deep Reinforcement Learning with Distributional Semantic Rewards for Abstractive Summarization"). We discuss these follow up works in Section 7.
>
> [1] Chi-kiu Lo. 2017. Meant 2.0: Accurate semantic mt evaluation for any output language. In Proceedings of the Second Conference on Machine Translation, Volume 2: Shared Tasks Papers, Copenhagen, Denmark, September. Association for Computational Linguistics.
> [2] Chi-kiu Lo. 2018. The NRC metric submission to the WMT18 metric and parallel corpus filtering shared task. Proceedings of the Third Conference on Machine Translation: Shared Task Papers, Belgium, Brussels, October. Association for Computational Linguistics.
> [3] Qin, L., Bosselut, A., Holtzman, A., Bhagavatula, C., Clark, E., & Choi, Y. (2019). Counterfactual Story Reasoning and Generation. Proceedings of the 2019 Conference on Empirical Methods in Natural Language Processing and the 9th International Joint Conference on Natural Language Processing (EMNLP-IJCNLP). Hong Kong, China, November. Association for Computational Linguistics.

---

### Official Review · AnonReviewer2 · 2019-10-24
**Official Blind Review #2**

**Rating:** 3

**Review:**

This paper presents a simple application of BERT-based contextual embeddings for evaluation of text generation models such as machine translation and image caption generation. An extensive set of experiments have been carried out to show that the proposed BERTScore metric achieves better correlation with human judgments than the existing metrics. Overall, the paper is well-written and the motivations are clear. However, I am not sure about the technical novelty of the paper as the proposed approach is a natural application of BERT along with traditional cosine similarity measures and precision, recall, F1-based computations, and simple IDF-based importance weighting.

Other comments:

- It would be interesting to see how the proposed metric performs to evaluate paraphrase generation and text simplification models as the models need to follow specific constraints such as semantic equivalence, novelty, simplicity etc. with respect to source and reference sentences.

- Another limitation of the proposed metric is memory and time complexity as it takes relatively more time to evaluate the sentences compared to BLEU, as authors acknowledged in Section 5.

**Experience Assessment:**

I have published in this field for several years.

**Review Assessment: Checking Correctness Of Derivations And Theory:**

N/A

**Review Assessment: Checking Correctness Of Experiments:**

I carefully checked the experiments.

**Review Assessment: Thoroughness In Paper Reading:**

I read the paper thoroughly.

---

> ### Author Response · Authors · 2019-11-07
> **Thank you for your comments!**
>
> Thank you for your comments!
>
> We agree that paraphrase generation and text simplification pose interesting challenges for evaluation methods. We consider this outside the scope of this work. In general, this work was enabled by data from multiple WMT competitions, generated over multiple years, and with huge efforts. Similar data, including system outputs and human judgments, don’t exist for paraphrase generation and text simplification. We hope our extensive experiments further demonstrate the utility of such competitions, and potentially will encourage them for other text generation tasks.
>
> While human evaluation for text simplification is not available, we conduct additional experiments on a similar task, abstractive text compression, using the MSR Abstractive Text Compression Dataset [1]. Human labelers are asked to score each compressed text in two aspects: meaning and grammar. A metric is evaluated based on its Pearson correlation with human judgment. The details are explained in their paper. We will add the results to the paper in the next few days. In the meantime, the results are:
> ————————————————————————————————————
> ————————————————————————————————————
> Metric                                                 Meaning     Grammar      Combined (M+G)
> ————————————————————————————————————
> BERTScore using RoBERTa-large:
> P_BERT                                                        0.36            *0.47*              0.46
> R_BERT                                                      *0.64*            0.29                0.52
> F_BERT                                                         0.58              0.41              *0.56*
> ————————————————————————————————————
> Common Metrics:
> BLEU                                                             0.46              0.13                0.33
> METEOR                                                       0.53              0.11                0.36
> ROUGE_L                                                      0.51              0.16                0.38
> SARI                                                              0.50              0.15                0.37
> ————————————————————————————————————
> Best metrics found by Toutanova et al.:
> SKIP-2+Recall+MULT-PROB                      0.59               n/a                 0.51
> PARSE-2+Recall+MULT-MAX                      n/a              0.35                0.52
> PARSE-2+Recall+MULT-PROB                   0.57              0.35                0.52
> ————————————————————————————————————
> ————————————————————————————————————
>
> We observe that R_BERT has the highest correlation with human judgments on whether the meaning is preserved, and P_BERT correlates highly with human judgments on whether the compressed text is grammatically correct. F_BERT lies at a sweet spot when we want to balance these two aspects.
>
>
>
> We wish to provide more context regarding the time complexity of BERTScore. On the same test set we used in Section 5, a standard transformer model takes about 58s for inference to generate the test translations. BERTScore takes ~15s, and BLEU takes ~5s. Overall, using BERTScore adds 25% of overhead to the evaluation process (inference+testing). As Reviewer 1 pointed out, this computational cost is relatively small and definitely worth it. During training, if using BERTScore to score the validation set (usually scored at specific intervals), the addition of time to the overall learning process is marginal.
>
>
> In our design, we were data-driven, and this led us to a relatively simple method. The advantage of the simplicity is how easy it is to compute it (which helps to make the time impact marginal), and the lack of additional parameters and design choices that further complicate the metric search space. In its simplicity, BERTScore is similar to BLEU and provides insight into the scoring process by allowing for the visualization of the matching. We hope this will help its adoption in the community. We experimented with several novel technical improvements in Appendix C, where we compare a recent approach that uses word mover’s distance and other technical enhancements. We find the benefit of these enhancements marginal (Table 9).
>
> [1] Toutanova et al. "A dataset and evaluation metrics for abstractive compression of sentences and short paragraphs.” Proceedings of the 2016 Conference on Empirical Methods in Natural Language Processing

---

### Official Review · AnonReviewer3 · 2019-10-24
**Official Blind Review #3**

**Rating:** 6

**Review:**

Paper Contributions

This paper introduces a new text generation scoring approach using BERT, called BERTScore. Using BERT embeddings and optionally idf scores, a greedy matching is performed between all reference and candidate words, with cosine similarity between vector representations as the scoring. From this, a precision, recall and F1 score can be derived. This notably outperforms BLEU, as well as other metrics, most but not all of the time. The paper offers a broad range of comparisons and analysis.

Decision

I'm leaning towards accepting the paper on the basis of the following.

Strong points taken in consideration:
- Simple, well-motivated metric that uses powerful BERT-style models, without being slow to compute either.
- Good performance empirically on WMT. I'm less convinced on COCO since using the image is fair game there.
- Code is provided, and it is simple and adaptable for future work.
- Experimentation is detailed and reproducible.

Weaker points taken in consideration:
- Work conducted in parallel matches or exceeds the performance of BERTScore. This shouldn't necessarily be a reason to choose not to publish this work in my opinion, but it should be taken into consideration. I like that the authors were open and clear regarding this in their discussion.
- The authors haven't come up with a recommendation for a single configuration of their approach. In one place they recommend F-BERT without idf, in another they argue for picking and choosing based on context, with little help about how to choose. I think practitioners are only going to be willing to switch away from BLEU, for example, if a single one-size-fits-all metric is proposed instead. I identify this ambiguity between BERTScore versions as an important weakness of the paper.
- It's unclear throughout whether words or wordpieces are the main token being considered. Most discussion and definitions use "words", but in section 3, subsection Token Representation, it appears to be clearly stated that BERTScore uses a BERT model based on word pieces. I recommend adjust the language to be more consistent throughout. Also, scoring examples with word pieces would be more consistent with this as well, imo. Notably, I'm actually unsure whether you compute IDF over words or word pieces, and how this is applied.
- Finally, I found some weaknesses in the Importance Weighting section (though this isn't too important since IDF isn't part of the recommended BERTScore I believe). The IDF scores would be stronger if they were computed on a bigger in-domain corpus than the gold test set. This would add extra steps to using BERTScore though and make things more complicated in practice, but this should nevertheless probably be tried, or at least discussed in the paper. Also, the plus-one smoothing handles unknown words (or word piece?) and I'm not sure why. If we're using the test set to compute IDF, and the sentences we're looking *are* in the test set, then there shouldn't be unknown words and no smoothing is required.

So overall, I still think this deserves publication because it's valuable information for researchers, and the metric itself could be immediately useful to some as well. However, the weaknesses mentioned make me hesitate to fully endorse the work.


**Experience Assessment:**

I have published one or two papers in this area.

**Review Assessment: Checking Correctness Of Derivations And Theory:**

I carefully checked the derivations and theory.

**Review Assessment: Checking Correctness Of Experiments:**

I assessed the sensibility of the experiments.

**Review Assessment: Thoroughness In Paper Reading:**

I read the paper at least twice and used my best judgement in assessing the paper.

---

> ### Author Response · Authors · 2019-11-07
> **Thank you for your comments!**
>
> Thank you for your comments!
>
> We agree with R3 that it would be ideal to have a one-size-fits-all metric. Unfortunately, the complex landscape of the problem doesn’t permit a single recommendation. We did our best to conduct a detailed and honest study. We believe our experiments to be some of the most extensive in this area, and we hope they will contribute to researchers’ understanding of the problem. It’s important to note, though, that BERTScore is an improvement over the commonly used Bleu across the board. Our recommendation to use F1, while potentially not optimal in specific cases, generally performs very well and much better than Bleu. There are largely two sets of options, (1) Among P, R, F; and  (2) What model to use. For (1), as we specify, F-BERT is a reliable metric for MT. For (2), Roberta-Large performs consistently well for to-English language pairs. The results are less conclusive for from-English language pairs. BERTScore computed with Multilingual-BERT is better than most existing metrics except on few low-resource languages. We have updated the paper with these recommendations in Section 7.
>
> We are using word pieces in all experiments, and we compute IDF using word pieces. We updated the paper to make this clear in Section 3, under Importance Weighting.
>
> Regarding unknown words handling, we computed the IDF on the reference sentences in the test set. This ensures that the IDF is the same for all MT systems that are tested. The candidate sentences generated by MT systems may contain words that never appear in the test set. We apply plus-one smoothing to handle such words.
> Following your suggestion, we further studied idf scoring. We computed idf scores on the monolingual English corpus released by WMT18 and experimented with BERTScore computed with the Roberta-large model. We have found that this leads to worse performance, likely because of the domain shift.

---

### Decision · Program_Chairs · 2019-12-19

**Decision:**

Accept (Poster)

**Comment:**

Thanks for an interesting discussion. The authors present a supposedly task-independent evaluation metric for generation tasks with references that relies on BERT or similar pretrained language models and a BERT-internal alignment. Reviewers are moderately positive. I encourage the authors to think about a) whether their approach scales to language pairs where wordpieces are less comparable; b) whether second order similarly, e.g., using RSA, would be better than alignment-based similarity; c) whether this metric works in the extremes, e.g., can it distinguish between bad output and super-bad output (where in both cases alignment may be impossible), and can it distinguish between good output and super-good output (where BERT scores may be too biased by BERT's training objective).